# Estimating Continuous Treatment Effects with Two-Stage Kernel Ridge Regression

**Seok-Jin Kim**[1]   **Kaizheng Wang**[1,2]

## Abstract

We study the problem of estimating the effect function for a continuous treatment, which maps each treatment value to a population-averaged outcome. A central challenge in this setting is confounding: treatment assignment often depends on covariates, creating selection bias that makes direct regression of the response on treatment unreliable. To address this issue, we propose a two-stage kernel ridge regression method. In the first stage, we learn a model for the response as a function of both treatment and covariates; in the second stage, we use this model to construct pseudo-outcomes that correct for distribution shift, and then fit a second model to estimate the treatment effect. Although the response varies with both treatment and covariates, the induced effect function obtained by averaging over covariates is typically much simpler, and our estimator adapts to this structure. Our optimal learning bounds are achieved without estimating the conditional treatment density, thereby bypassing a major bottleneck in existing methods. Furthermore, we introduce a fully data-driven model selection procedure that achieves provable adaptivity to both the unknown degree of overlap and the spectral decay of the underlying kernel.

## 1. Introduction

Estimating the causal effects of continuous treatments—such as drug dosages, training intensity, or varying policy levels—is a fundamental challenge in causal inference (Ishak et al., 2015; Kennedy et al., 2017; Colangelo & Lee, 2026; Zhang & Chen, 2025). The primary estimand of interest is the Treatment Effect Function (TEF), which maps each treatment value to the population-averaged outcome. A central difficulty in observational settings is confounding (or distribution shift): treatment assignment often depends on complex, high-dimensional covariates. Consequently, direct regression of the outcome on the treatment yields biased estimates of the TEF. Because the TEF depends only on the treatment, a central goal is to derive learning bounds that adapt to its typically lower complexity rather than to that of the nuisance component.

State-of-the-art nonparametric approaches typically address this via two-stage "debiasing" procedures, which estimate nuisance components—most notably the conditional outcome mean and the treatment density, or generalized propensity score—to construct pseudo-outcomes (Kennedy et al., 2017; Colangelo & Lee, 2026; Bonvini & Kennedy, 2022). A key bottleneck is the conditional treatment density: existing guarantees typically require estimating it, together with the outcome nuisance, in stronger norms such as $L^4$ or $L^\infty$, rather than only $L^2$. Providing such convergence is especially challenging when covariates are high-dimensional, discrete, or sparse. Moreover, estimating the conditional treatment density is often challenging and highly sensitive to weak overlap and model misspecification. Consequently, continued efforts have been made to avoid this difficulty (Tübbicke, 2022; Huling et al., 2024; Fong et al., 2018).

In this work, we propose a kernel-based framework that overcomes these limitations. By leveraging the flexibility of Reproducing Kernel Hilbert Spaces (RKHS) (Schölkopf & Smola, 2002), our method accommodates rich nonlinear interactions while naturally capturing the smoothness structures inherent in causal problems. We establish convergence rates that adapt to the intrinsic complexity of the TEF. Unlike prior debiasing methods, our approach avoids the instability associated with explicit conditional density estimation in high dimensions. Furthermore, this adaptivity requires only that the nuisance outcome model be correctly specified, with no explicit convergence-rate assumptions on nuisance estimation. These features are especially valuable when the covariate distribution is poorly behaved, when conditional-density estimation is difficult, and when the

[1]Department of Industrial Engineering and Operations Research, Columbia University, New York, NY, United States [2]Data Science Institute, Columbia University, New York, NY, United States. Correspondence to: Kaizheng Wang < kaizheng.wang@columbia.edu>.

*Proceedings of the $43^{rd}$ International Conference on Machine Learning*, Seoul, South Korea. PMLR 306, 2026. Copyright 2026 by the author(s).

problem lies in a weak-overlap regime. Below are our main contributions:

- **Methodology:** We propose a two-stage kernel ridge regression (KRR) estimator. The first stage learns the conditional outcome surface, and the second stage regresses the estimated pseudo-outcomes on the treatment variable. To ensure practical applicability, we develop a fully data-driven model selection procedure that eliminates the need for manual tuning of regularizers.

- **Theory:** We establish minimax-optimal error bounds for our estimator. Our learning bounds adapt to the complexity of the TEF while remaining decoupled from nuisance complexity, so the leading rate is not driven by the nuisance component. We also prove adaptation to unknown structural parameters, specifically the degree of treatment overlap and the spectral decay of the kernel.

**Related Work.** Early studies on continuous treatment effects focused primarily on parametric models (Robins, 2000; van der Laan & Robins, 2003; Díaz & van der Laan, 2013; Imbens, 2000), while recent advances have shifted toward flexible nonparametric estimation (Kennedy et al., 2017; Semenova & Chernozhukov, 2021; Bonvini & Kennedy, 2022; Colangelo & Lee, 2026; Takatsu & Westling, 2025). The dominant paradigm in this literature combines orthogonalization (debiasing) with Hölder function classes. However, these methods rely on estimating the conditional treatment density and typically require convergence guarantees for the nuisance estimators in norms stronger than $L^2$, such as $L^4$ or $L^\infty$, which become increasingly difficult to obtain as the covariate dimension grows.

RKHS-based methods offer a powerful alternative for structural adaptation in causal inference (Nie & Wager, 2021; Mou et al., 2023; Singh et al., 2024; Hirshberg & Wager, 2021; Kim et al., 2025). Mou et al. (2023) study kernel methods for policy evaluation in contextual bandits, but their setting is essentially confounding-free rather than observational. Most relevant to our work, Singh et al. (2024) analyze continuous treatment effect estimators using tensor-product kernels under source conditions on the response function. Their guarantees are formulated via assumptions on the joint tensor-product space, and it remains unclear whether those rates adapt to the spectrum of the treatment effect space alone. In contrast, we consider general kernels and dispense with the source condition requirement, thereby offering broader applicability.

Our upper bound is in fact faster than the standard double machine learning (DML) rate, which helps explain why conditional-density estimation can be avoided in our framework. There is a line of work that likewise avoids explicit conditional treatment-density estimation by achieving faster rates than standard DML, including Bonvini & Kennedy (2022) and Huling et al. (2024). These results, however, are restricted to Sobolev- or Hölder-type function classes, require strong assumptions on the nuisance structure, or rely on computationally expensive procedures. We return to a more detailed comparison with these works after presenting our main results.

Finally, the concept of two-stage regression has appeared in semi-supervised learning with auxiliary covariates (Liu et al., 2023; Xia & Wainwright, 2024). However, these works address fundamentally different problems and do not analyze TEF estimation or the adaptation phenomena that arise in continuous treatment settings.

**Notation.** The constants $c_1, c_2, C_1, C_2, \ldots$ may differ from line to line. We use the symbol $[n]$ as a shorthand for $\{1, 2, \ldots, n\}$. For nonnegative sequences $\{a_n\}_{n=1}^\infty$ and $\{b_n\}_{n=1}^\infty$, we write $a_n \lesssim b_n$ or $a_n = \mathcal{O}(b_n)$ if there exists a positive constant $C$ such that $a_n \leq C b_n$ for all $n$. We use $\widetilde{\mathcal{O}}$ to hide polylogarithmic factors. In addition, we write $a_n \asymp b_n$ if $a_n \lesssim b_n$ and $b_n \lesssim a_n$. For a bounded linear operator $\boldsymbol{A}$, we use $\|\boldsymbol{A}\|_{\mathrm{op}}$ to refer to its operator norm. For any $\boldsymbol{u}$ and $\boldsymbol{v}$ in a Hilbert space $\mathbb{H}$, their inner and outer products are denoted by $\langle \boldsymbol{u}, \boldsymbol{v} \rangle_{\mathbb{H}}$ and $\boldsymbol{u} \otimes \boldsymbol{v}$, respectively. We use $\mathcal{N}(\boldsymbol{\mu}, \boldsymbol{\Sigma})$ to represent the Gaussian distribution with mean $\boldsymbol{\mu}$ and covariance matrix $\boldsymbol{\Sigma}$, and $\mathcal{U}(S)$ for the uniform distribution on a set $S$.

## 2. Problem Setup

### 2.1. Formulation and Objective

Let $\mathcal{X}$ denote the covariate space and $\mathcal{A}$ the domain of continuous treatments (e.g., drug dosage). Let $X \in \mathcal{X}$ and $A \in \mathcal{A}$ denote generic random covariates and treatment, respectively, and write $\mathcal{P}_{X,A}$ for the joint law of $(X, A)$, with marginal $\mathcal{P}_X$. We set $\mathcal{Z} := \mathcal{X} \times \mathcal{A}$. Under the potential outcomes framework (Robins, 2000), a generic unit is endowed with the collection $\{Y(a)\}_{a \in \mathcal{A}}$, where $Y(a)$ is the potential outcome under treatment level $a$. The observed outcome is $Y := Y(A)$. We observe an i.i.d. dataset $\mathcal{D} = \{(x_i, a_i, y_i)\}_{i=1}^n$ drawn from the law of $(X, A, Y)$, and write $z_i := (x_i, a_i)$.

We define the conditional mean response by

$$f^\star(x, a) := \mathbb{E}[Y \mid X = x, A = a].$$

Our target estimand is the *Treatment Effect Function (TEF)*,

$$h^\star(a) := \mathbb{E}[Y(a)]$$

which maps each treatment value to the population-averaged outcome. Under the identification conditions, which we define formally in Section 2.3, we have

$$h^\star(a) = \mathbb{E}_{X \sim \mathcal{P}_X}[f^\star(X, a)]$$

for every $a \in \mathcal{A}$.

A fundamental challenge in this setting is confounding (or selection bias). In observational data, treatment assignment typically depends on covariates; consequently, the conditional distribution of covariates given treatment, denoted by $\mathcal{P}_{X|A=a}$, varies with $a$ and generally differs from the marginal distribution $\mathcal{P}_X$. Consider the function obtained by directly regressing the observed outcome $Y$ on the treatment $A$:

$$g(a) = \mathbb{E}[Y \mid A = a] = \mathbb{E}_{x \sim \mathcal{P}_{X|A=a}}[f^\star(x, a)].$$

Because the integration measure $\mathcal{P}_{X|A=a}$ changes with $a$, the function $g(a)$ conflates the true treatment effect with the effect of the shifting covariate distribution. As a result, $g(a)$ generally differs from $h^\star(a)$, and recovering $h^\star$ necessitates adjusting for this distribution shift.

Our objective is to construct an estimator $\hat{h}$ that minimizes the Mean Integrated Squared Error (MISE) with respect to a reference distribution $\mathcal{P}_{\mathrm{ref}}$ on $\mathcal{A}$:

$$\mathcal{E}(\hat{h}) := \int_{\mathcal{A}} \left( \hat{h}(a) - h^\star(a) \right)^2 \mathrm{d}\mathcal{P}_{\mathrm{ref}}(a). \qquad (2.1)$$

The reference distribution $\mathcal{P}_{\mathrm{ref}}$ is user-specified and reflects the region of interest for evaluation. In nonparametric function estimation, MISE is a standard performance metric; see Tsybakov (2009). It captures the overall quality of the estimated function. The same criterion is also widely used in continuous-treatment settings: Schwab et al. (2020) use an integrated-error criterion for dose-response estimation, and Kennedy et al. (2017) evaluate the estimated curve using integrated error in their experiments. $L^2$-type criteria are also standard in binary-treatment settings; see Nie & Wager (2021) and Foster & Syrgkanis (2023).

## 2.2. RKHS Framework

We assume that both the conditional response function $f^\star$ and the treatment effect function $h^\star$ reside in RKHSs, denoted by $\mathcal{F}$ and $\mathcal{H}$, defined on domains $\mathcal{Z}$ and $\mathcal{A}$, respectively. Let $K_{\mathcal{F}}$ and $K_{\mathcal{H}}$ be their associated positive semidefinite kernels. By standard theory (Aronszajn, 1950), there exist Hilbert spaces $\mathbb{F}$ and $\mathbb{H}$, along with feature maps $\psi : \mathcal{Z} \to \mathbb{F}$ and $\phi : \mathcal{A} \to \mathbb{H}$, satisfying the reproducing properties: $\langle \psi(z), \psi(z') \rangle_{\mathbb{F}} = K_{\mathcal{F}}(z, z')$ and $\langle \phi(a), \phi(a') \rangle_{\mathbb{H}} = K_{\mathcal{H}}(a, a')$. Accordingly, the hypothesis spaces are defined as:

$$\mathcal{F} = \{ f_\theta(\cdot) = \langle \psi(\cdot), \theta \rangle_{\mathbb{F}} \mid \theta \in \mathbb{F} \},$$
$$\mathcal{H} = \{ h_\eta(\cdot) = \langle \phi(\cdot), \eta \rangle_{\mathbb{H}} \mid \eta \in \mathbb{H} \}.$$

Here, $\mathcal{F}$ and $\mathcal{H}$ denote RKHSs of functions, whereas $\mathbb{F}$ and $\mathbb{H}$ denote the coefficient Hilbert spaces associated with the feature maps $\psi$ and $\phi$.

A central motivation for our framework is the observation that $h^\star$, being a marginal average of $f^\star$ over the covariate distribution $\mathcal{P}_X$, typically exhibits lower structural complexity than the nuisance component $f^\star$ since it is a function of lower dimension. Our method leverages this distinction through flexible modeling. To illustrate, consider the case where $\mathcal{Z} \subset \mathbb{R}^d$ and $\mathcal{A} \subset \mathbb{R}^p$ are bounded regular domains.

- *Sobolev smoothness.* The standard Sobolev space $H^\beta(\mathcal{Z})$ (with $\beta > d/2$) and the mixed Sobolev space $H^\beta_{\mathrm{mix}}(\mathcal{Z})$ (with $\beta > 1/2$) are known to admit RKHS structures (Doumèche et al., 2025; Zhang et al., 2023; Dick et al., 2007; Kühn et al., 2015; Suzuki, 2019). [1] If $f^\star \in H^\beta(\mathcal{Z})$ or $f^\star \in H^\beta_{\mathrm{mix}}(\mathcal{Z})$, then the target function $h^\star$ inherits the same degree of $\beta$-smoothness on $\mathcal{A}$; that is, $h^\star \in H^\beta(\mathcal{A})$ or $h^\star \in H^\beta_{\mathrm{mix}}(\mathcal{A})$, respectively. Since the treatment dimension $p$ is typically much smaller than the dimension $d$ of the joint covariate-treatment space $\mathcal{Z}$ ($p \ll d$), estimating $h^\star$ directly in $\mathcal{H}$ can yield statistically more efficient rates than estimating $f^\star$ in $\mathcal{F}$.

- *High-dimensional covariates.* Our framework allows for modeling $\mathcal{F}$ using kernels designed for high-dimensional data, such as Neural Tangent Kernels (NTK) (Ghorbani et al., 2021; Bietti & Bach, 2021) or inner product kernels (Mei et al., 2022; Liang & Rakhlin, 2020; Mei et al., 2021), to capture complex nuisance interactions. Simultaneously, one can model the simpler dose-response curve in $\mathcal{H}$ using a standard smooth kernel (e.g., Matérn), thereby decoupling the complexity of the nuisance from the target.

### 2.3. Assumptions

We posit the following standard assumptions to facilitate our theoretical analysis.

**Assumption 2.1** (Identification). (i) Consistency: $Y = Y(A)$ almost surely. (ii) No unmeasured confounding: $\{Y(a)\}_{a \in \mathcal{A}} \perp A \mid X$.

**Assumption 2.2** (Sub-Gaussian Noise). The noise $\varepsilon := Y - f^\star(X, A)$ is $\sigma$-sub-Gaussian given $(X, A)$, that is, $\mathbb{E}[e^{t\varepsilon} \mid X, A] \le e^{t^2 \sigma^2 / 2}$ for all $t \in \mathbb{R}$. We assume $\sigma$ is bounded by an absolute constant.

**Assumption 2.3** (Boundedness). The kernels are uniformly bounded by a constant $\xi$: $\sup_{z \in \mathcal{Z}} K_{\mathcal{F}}(z, z) \le \xi$ and $\sup_{a \in \mathcal{A}} K_{\mathcal{H}}(a, a) \le \xi$. Additionally, we assume $\|h^\star\|_{\mathcal{H}}$ is bounded by a constant.

Assumption 2.1 is standard in the causal inference literature (Bonvini & Kennedy, 2022; Kennedy et al., 2017;

---

[1]Compared with standard Sobolev spaces, mixed Sobolev spaces require only $\beta > 1/2$, which is substantially weaker than $\beta > d/2$.

Künzel et al., 2019; Curth & Van der Schaar, 2021). Under Assumption 2.1, we have $f^\star(x, a) = \mathbb{E}[Y(a) \mid X = x]$, and therefore $h^\star(a) = \mathbb{E}_{x \sim \mathcal{P}_X}[f^\star(x, a)]$. Assumption 2.2 is also standard. For example, Bonvini & Kennedy (2022); Kennedy et al. (2017); Nie & Wager (2021) assume bounded outcomes, which imply sub-Gaussian noise and are therefore stronger than our assumption. The boundedness of the kernels in Assumption 2.3 is common in the analysis of kernel methods (Wainwright, 2019; Singh et al., 2024). Notably, unlike many existing results in nonparametric regression, we *do not* assume that the norm of the full conditional response, $\|f^\star\|_{\mathcal{F}}$, is bounded by a constant. Instead, we allow it to grow with $n$, and our error bounds explicitly characterize the dependence on this quantity. This relaxes the requirement that the complex nuisance function $f^\star$ must lie strictly within a fixed ball of the RKHS.

## 3. Methodology

This section introduces the two-stage KRR estimator and the associated data-driven model selection procedure. Section 3.1 presents the estimation algorithm, and Section 3.2 describes how we select the second-stage regularizer.

### 3.1. Estimation Procedure

We propose a two-stage KRR framework designed to decouple the estimation of the complex conditional response surface $f^\star$ from the smoothing of the simpler treatment effect function $h^\star$.

The procedure consists of three steps: (1) learning the full conditional mean $f^\star$ via KRR; (2) constructing "pseudo-outcomes" by empirically marginalizing the estimated nuisance function over the covariate distribution; and (3) regressing these pseudo-outcomes on treatment values to recover $h^\star$. A summary is provided in Algorithm 1.

**Step 1: Nuisance Estimation.** We first estimate the conditional mean function $f^\star : \mathcal{Z} \to \mathbb{R}$ using KRR on the joint space $\mathcal{Z} = \mathcal{X} \times \mathcal{A}$. Recall that $\mathcal{D} = \{(x_i, a_i, y_i)\}_{i=1}^n$ denotes the observed data. We solve the following optimization problem over the hypothesis space $\mathcal{F}$:

$$\hat{f} := \underset{f \in \mathcal{F}}{\operatorname{argmin}} \left\{ \frac{1}{n} \sum_{i=1}^n (y_i - f(x_i, a_i))^2 + \lambda_0 \|f\|_{\mathcal{F}}^2 \right\}. \quad (3.1)$$

The solution admits the usual kernel-trick representation in terms of the kernel similarities $\{K_{\mathcal{F}}(z_i, z_j)\}_{1 \le i, j \le n}$. We use a small *nuisance regularizer* $\lambda_0 \asymp n^{-1} \log n$ because the first stage is intended primarily to control bias and recover the conditional surface accurately; the final bias-variance trade-off is handled in the second stage.

**Step 2: Empirical Marginalization.** To recover the target $h^\star(a) = \mathbb{E}_{X \sim \mathcal{P}_X}[f^\star(X, a)]$, we construct a synthetic

dataset. We sample $n$ treatment values $\{a'_j\}_{j=1}^n$ i.i.d. from a user-specified sampling distribution $\mathcal{P}_{\text{samp}}$ (e.g., the uniform distribution on $\mathcal{A}$). For each $a'_j$, we compute the *pseudo-outcome* $m_j$ by averaging the fitted nuisance function $\hat{f}$ over the empirical distribution of covariates:

$$m_j := \frac{1}{n} \sum_{i=1}^n \hat{f}(x_i, a'_j). \quad (3.2)$$

Here, $m_j$ serves as a noisy proxy for $h^\star(a'_j)$. The sampling distribution $\mathcal{P}_{\text{samp}}$ allows us to probe the treatment effect function across the entire domain $\mathcal{A}$, independent of the observational treatment distribution $\mathcal{P}_A$.

**Step 3: Pseudo-Outcome Regression.** Finally, we treat the constructed set $\mathcal{D}' = \{(a'_j, m_j)\}_{j=1}^n$ as our dataset for the target function. We perform a second KRR in the simpler space $\mathcal{H}$ to smooth the pseudo-outcomes:

$$\hat{h}_\lambda := \underset{h \in \mathcal{H}}{\operatorname{argmin}} \left\{ \frac{1}{n} \sum_{j=1}^n (m_j - h(a'_j))^2 + \lambda \|h\|_{\mathcal{H}}^2 \right\}. \quad (3.3)$$

The *main regularizer* $\lambda$ controls the final smoothness of the TEF. We propose a data-driven procedure to select the optimal $\lambda$ in Section 3.2.

---

**Algorithm 1** TEF Estimation via Two-Stage KRR
***
**Require:** Dataset $\mathcal{D} = \{(x_i, a_i, y_i)\}_{i=1}^n$, sampling distribution $\mathcal{P}_{\text{samp}}$, nuisance regularizer $\lambda_0$, main regularizer $\lambda$.
1: **Stage 1:** Compute $\hat{f}$ by solving Equation (3.1).
2: **Stage 2:** Sample query points $a'_1, \ldots, a'_n \overset{\text{i.i.d.}}{\sim} \mathcal{P}_{\text{samp}}$.
3: Compute pseudo-outcomes: $m_j \leftarrow \frac{1}{n} \sum_{i=1}^n \hat{f}(x_i, a'_j)$ for $j = 1, \ldots, n$.
4: **Stage 3:** Compute $\hat{h}_\lambda$ by solving Equation (3.3) on $\{(a'_j, m_j)\}_{j=1}^n$.
   **Output** Estimator $\hat{h}_\lambda$.

---

### 3.2. Model Selection Procedure

While the nuisance regularizer $\lambda_0$ in Algorithm 1 follows a concrete theoretical guideline ($\lambda_0 \asymp n^{-1} \log n$), the main regularizer $\lambda$ requires careful tuning. Standard cross-validation is not directly applicable in this setting because the target labels $h^\star(a)$ are never observed directly. Instead, we only observe noisy outcomes $y_i$ that are confounded by covariates. To address this challenge, we introduce a fully data-driven model selection procedure based on *proxy validation*. The core idea is to construct a nearly unbiased (albeit noisy) "proxy" for the ground truth via data splitting, allowing us to estimate the generalization error of the candidate models. Crucially, this procedure provably adapts to

unknown structural parameters without requiring manual tuning.

We randomly split the dataset $\mathcal{D}$ into a training set $\mathcal{D}_{\text{train}}$ and a validation set $\mathcal{D}_{\text{val}}$, each of size $n/2$.

1. **Candidate Training ($\mathcal{D}_{\text{train}}$):** On the training set, we run Algorithm 1 for a grid of main regularizers $\Lambda$, generating a set of candidate estimators $\mathcal{M} = \{\hat{h}_\lambda \mid \lambda \in \Lambda\}$.

2. **Proxy Construction ($\mathcal{D}_{\text{val}}$):** On the validation set, we run Algorithm 1 using a fixed, small main regularizer $\tilde{\lambda} \asymp n^{-1} \log n$. Let $\tilde{h}$ denote this proxy estimator.

3. **Selection:** We select the candidate $\hat{h}_\lambda \in \mathcal{M}$ that is closest to the proxy $\tilde{h}$ in terms of the $L^2(\mathcal{P}_{\text{ref}})$ distance. In practice, we approximate this distance by Monte Carlo integration, sampling query points from the reference distribution $\mathcal{P}_{\text{ref}}$.

The formal procedure is detailed in Algorithm 2.

---

**Algorithm 2** Data-Driven Model Selection for TEF

---

**Require:** Dataset $\mathcal{D}$, reference distribution $\mathcal{P}_{\text{ref}}$, candidate grid $\Lambda$, nuisance regularizer $\lambda_0$, number of Monte Carlo samples $M$.
1: Randomly split $\mathcal{D}$ into $\mathcal{D}_{\text{train}}$ and $\mathcal{D}_{\text{val}}$ (sizes $n/2$).
2: **Train Candidates:** Run Algorithm 1 on $\mathcal{D}_{\text{train}}$ with each main regularizer $\lambda \in \Lambda$ to obtain $\{\hat{h}_\lambda\}_{\lambda \in \Lambda}$.
3: **Train Proxy:** Run Algorithm 1 on $\mathcal{D}_{\text{val}}$ with main regularizer $\tilde{\lambda} \asymp n^{-1} \log n$ to obtain $\tilde{h}$.
4: **Evaluate:** Sample test points $\tilde{a}_1, \ldots, \tilde{a}_M \overset{\text{i.i.d.}}{\sim} \mathcal{P}_{\text{ref}}$.
5: Select $\hat{\lambda}$ by minimizing the distance to the proxy:

$$\hat{\lambda} := \operatorname*{argmin}_{\lambda \in \Lambda} \sum_{k=1}^{M} \left( \hat{h}_\lambda(\tilde{a}_k) - \tilde{h}(\tilde{a}_k) \right)^2.$$

**Output** Selected estimator $\hat{h}_{\hat{\lambda}}$.

---

## 4. Theoretical Analysis

This section establishes theoretical guarantees for Algorithms 1 and 2. We first introduce a generalized notion of overlap tailored to our setting. We then derive upper bounds on the MISE that adapt to the spectral decay of the target kernel $K_\mathcal{H}$. Finally, we prove minimax lower bounds showing that our estimator is optimal with respect to both the sample size $n$ and the overlap parameter $\gamma$.

### 4.1. Upper Bound Analysis under Relative Overlap

Standard causal inference frameworks typically assume "positivity" (or "overlap"), requiring the conditional treatment density to be uniformly bounded away from zero.

We introduce a more flexible condition, *Relative Overlap*, which quantifies overlap specifically with respect to the user-specified reference distribution $\mathcal{P}_{\text{ref}}$.

**Definition 4.1** (Relative Overlap)**.** Let $\mathcal{P}_{A|X=x}$ denote the conditional distribution of treatment given covariates $x$. We say the data distribution satisfies *relative overlap of degree* $\gamma \in (0, 1]$ with respect to a reference measure $\mathcal{P}_{\text{ref}}$ if the following bound holds for all $x \in \mathcal{X}$:

$$\frac{\mathrm{d}\mathcal{P}_{\text{ref}}}{\mathrm{d}\mathcal{P}_{A|X=x}} \le \frac{1}{\gamma} \quad \text{almost everywhere on } \mathcal{A}.$$

Intuitively, $\gamma$ represents the minimum density of the observed treatments relative to the target distribution. A smaller $\gamma$ implies weaker overlap. Based on this, we define the *effective sample size* as:

$$n_{\text{eff}} := \gamma n.$$

This quantity reflects the effective number of samples available for learning the target function $h^\star$ after accounting for the distribution shift.

Under this overlap condition, we now characterize the complexity of the target RKHS $\mathcal{H}$ via the spectral decay of its kernel integral operator in order to state our convergence rates. Define the second moment operator associated with kernel $K_\mathcal{H}$ and reference distribution $\mathcal{P}_{\text{ref}}$ as

$$\mathbf{\Sigma}_{\text{ref}} := \mathbb{E}_{a \sim \mathcal{P}_{\text{ref}}}[\phi(a) \otimes \phi(a)].$$

The following definition summarizes the spectral decay regimes considered in our analysis.

**Definition 4.2** (Spectral Decay)**.** Let $\rho_1 \ge \rho_2 \ge \ldots$ denote the eigenvalues of $\mathbf{\Sigma}_{\text{ref}}$. We consider three commonly studied decay regimes:

(a) **Polynomial:** $\rho_j \lesssim j^{-2\ell}$ for some $\ell > 1/2$ (e.g., Sobolev spaces).

(b) **Exponential:** $\rho_j \lesssim \exp(-cj)$ for some $c > 0$ (e.g., Gaussian kernel).

(c) **Finite Rank:** $\rho_j = 0$ for all $j > D$ (e.g., Linear kernels).

We further define the *effective dimension* at scale $\lambda$ as

$$\Gamma(\lambda) := \text{Tr}[(\mathbf{\Sigma}_{\text{ref}} + \lambda \mathbf{I})^{-1} \mathbf{\Sigma}_{\text{ref}}] = \sum_{j=1}^{\infty} \frac{\rho_j}{\rho_j + \lambda}.$$

Our main result establishes a finite-sample bound for the proposed estimator.

**Theorem 4.3** (MISE Upper Bound). *Suppose we run Algorithm 1 with nuisance regularizer $\lambda_0 \asymp n^{-1} \log n$, main regularizer $\lambda \gtrsim n^{-1} \log n$, and sampling distribution $\mathcal{P}_{\mathrm{samp}} = \mathcal{P}_{\mathrm{ref}}$. Under Assumptions 2.1 to 2.3 and Relative Overlap of degree $\gamma$, with probability at least $1 - n^{-10}$, the estimator $\hat{h}_\lambda$ satisfies:*

$$\mathcal{E}(\hat{h}_\lambda) \lesssim \underbrace{\lambda \|h^\star\|_{\mathcal{H}}^2}_{Bias} + \underbrace{\frac{\Gamma(\lambda)}{\gamma n}}_{Variance} + \underbrace{\frac{\|f^\star\|_{\mathcal{F}}^2}{\gamma n}}_{Nuisance\ Error} . \qquad (4.1)$$

*Consequently, optimizing $\lambda$ yields the following convergence rates in terms of the effective sample size $n_{\mathrm{eff}} = \gamma n$:*

- ***Case (a) (Polynomial):*** *$\lambda \asymp n_{\mathrm{eff}}^{-\frac{2\ell}{1+2\ell}} \implies \mathcal{E}(\hat{h}_\lambda) \lesssim n_{\mathrm{eff}}^{-\frac{2\ell}{1+2\ell}} + \frac{\|f^\star\|_{\mathcal{F}}^2}{n_{\mathrm{eff}}}$.*

- ***Case (b) (Exponential):*** *$\lambda \asymp \frac{1}{n_{\mathrm{eff}}} \implies \mathcal{E}(\hat{h}_\lambda) \lesssim \frac{1}{n_{\mathrm{eff}}} + \frac{\|f^\star\|_{\mathcal{F}}^2}{n_{\mathrm{eff}}}$.*

- ***Case (c) (Finite Rank):*** *$\lambda \asymp \frac{D}{n_{\mathrm{eff}}} \implies \mathcal{E}(\hat{h}_\lambda) \lesssim \frac{D}{n_{\mathrm{eff}}} + \frac{\|f^\star\|_{\mathcal{F}}^2}{n_{\mathrm{eff}}}$.*

*The notation $\lesssim$ hides constants and logarithmic factors.*

We defer the proof to Appendix A. The bound in Equation (4.1) highlights the main strength of our method: the leading bias-variance tradeoff is governed by the complexity of the *simpler* target space $\mathcal{H}$ through $\Gamma(\lambda)$, rather than by the potentially much more complex nuisance space $\mathcal{F}$. The nuisance function $f^\star$ enters only through the parametric-order term $\|f^\star\|_{\mathcal{F}}^2/(\gamma n)$. In particular, under polynomial eigendecay, the target-space term remains dominant whenever $\|f^\star\|_{\mathcal{F}}^2 = \mathcal{O}((\gamma n)^{1/(1+2\ell)})$. Moreover, our upper bound scales explicitly with the weak-overlap parameter through $n_{\mathrm{eff}} = \gamma n$, and the lower bound in Theorem 4.6 shows that this dependence is unimprovable in the Sobolev benchmark considered there. This structural decoupling yields fast rates in the following examples:

**Example 4.1** (Sobolev Nuisance). *If $\mathcal{F} = H^\beta(\mathcal{Z})$ and $\mathcal{H} = H^\beta(\mathcal{A})$ for $\mathcal{Z} \subset \mathbb{R}^d$, $\mathcal{A} \subset \mathbb{R}$ with $\beta > d/2$, then Case (a) implies a rate of $\tilde{\mathcal{O}}((\gamma n)^{-\frac{2\beta}{1+2\beta}})$. This is significantly faster than the rate for learning $f^\star$ directly, which would be $\mathcal{O}(n^{-\frac{2\beta}{d+2\beta}})$.*

**Example 4.2** (NTK Nuisance). *If $\mathcal{F}$ corresponds to a Neural Tangent Kernel (modeling an overparameterized neural network) but $h^\star \in H^1(\mathcal{A})$ for some $\mathcal{A} \subset \mathbb{R}$ (a smooth dose-response), our method achieves the fast rate $\tilde{\mathcal{O}}((\gamma n)^{-2/3})$ for the TEF, bypassing the slow learning rate associated with the high-dimensional NTK.*

**Comparison with Other Works.** Compared with double machine learning (DML) methods, our approach entails a clear tradeoff. In addition to avoiding conditional-density estimation, our analysis imposes no explicit estimation-error requirement on the nuisance outcome model $f^\star$. As a result, the theory continues to apply even in irregular covariate settings where standard guarantees for outcome regression may be unavailable.

DML methods, by contrast, offer robustness to misspecification, which is an advantage over our approach. However, their guarantees typically require quantitative rates for the nuisance estimators. In particular, the second-order remainder is driven by the product of the estimation errors for $f^\star$ and the conditional treatment density $P_{A|X=x}(\cdot)$, together with an additional weak-overlap penalty that often scales as $1/\gamma^2$. These nuisance requirements are also commonly stated in stronger norms such as $L^4$ or $L^\infty$, rather than only $L^2$, making them substantially more demanding. The extra $1/\gamma^2$ factor can itself become a serious burden for nuisance estimation under weak overlap. By contrast, our method dispenses entirely with conditional-density estimation and requires no learning bound for $f^\star$ beyond correct specification.

Another closely related RKHS-based approach is Singh et al. (2024). Their method is essentially plug-in: it first estimates a response function in a tensor-product RKHS over $(X, A)$, and then obtains the target curve by marginalizing over covariates. Due to the tensor-product structure, the resulting marginalized curve may indeed be represented as an element of the treatment RKHS $\mathcal{H}$. However, their error bounds remain governed by the eigendecay and source conditions of the larger nuisance space $\mathcal{F}$, and therefore do not adapt to the potentially simpler structure of $h^\star$ in $\mathcal{H}$. By contrast, our analysis allows broad RKHS choices beyond tensor-product kernels and yields rates whose leading term is controlled by the complexity of $\mathcal{H}$ itself.

There is also a line of work showing that one can avoid explicit conditional-density estimation and still achieve fast rates, but only under more specialized conditions. Robins et al. (2008) and Bonvini & Kennedy (2022) use higher-order influence functions (HOIFs) to obtain fast rates for binary and continuous treatments, respectively. In the continuous-treatment case, Bonvini & Kennedy (2022) avoid conditional-density estimation by replacing it with joint density estimation for $(X, A)$, and their theory requires this joint density to be sufficiently regular. Their guarantees are also limited to Hölder classes $\mathcal{F} = C^\beta(\mathcal{Z})$ with $\beta > d/2$, and the HOIF construction is computationally demanding, involving projections onto finite-dimensional subspaces and the evaluation of U-statistics. The independence-weighting approach of Huling et al. (2024) is closer in spirit, but its guarantees are tied to a product-type smooth-

ness class, roughly $\mathcal{F} \approx H^{d/2}(\mathcal{X}) \otimes H^1(\mathcal{A})$ together with $\mathcal{H} \approx C^2(\mathcal{A})$, which is a special case of our framework. Their method also requires solving a large-scale non-convex quadratic program, which is NP-hard in general. Their bound is not optimal in its dependence on the degree of overlap $\gamma$. By contrast, our method is computationally practical and provides optimal dependence on both $n$ and $\gamma$ for broad RKHS choices of $\mathcal{F}$ and $\mathcal{H}$, beyond classical Sobolev and Hölder classes. For example, our framework includes $\mathcal{F} = H_{\mathrm{mix}}^{\frac{1}{2}+\varepsilon}(\mathcal{Z})$ for any $\varepsilon > 0$, which is broader than the class treated in Huling et al. (2024).

**Proof Idea and Technical Novelty.** Our analysis differs from standard DML-type and plug-in arguments in that we do not collapse the first-stage estimator $\hat{f}$ into a deterministic nuisance-error term. Instead, we carry the randomness of $\hat{f}$, induced by the outcome noise, through the pseudo-outcomes and track how it propagates into the second stage. This yields a decomposition into propagated bias and variance: first-stage undersmoothing keeps the bias mild, while the second-stage regularizer shrinks the variance enough for it to be absorbed into the usual KRR variance term in $\mathcal{H}$. This is what allows the leading rate to be governed by the simpler target space $\mathcal{H}$, with the complexity of $\mathcal{F}$ entering only through a parametric-order term. This proof strategy is a key distinction from prior kernel and two-stage analyses such as Singh et al. (2024), Liu et al. (2023), and Xia & Wainwright (2024).

**Growing Nuisance Norms.** Another useful feature of Equation (4.1) is that it remains informative even when the nuisance norm $\|f^\star\|_{\mathcal{F}}$ is allowed to grow with $n$. In many RKHS analyses, the theorem is stated only over a fixed-radius ball, with the radius absorbed into the constant. Our bound makes this dependence explicit. More generally, under polynomial eigendecay, the optimal rate continues to hold whenever $\|f^\star\|_{\mathcal{F}}^2 = \mathcal{O}((\gamma n)^{1/(1+2\ell)})$. For example, when $\mathcal{H} = H^1(\mathcal{A})$, we still achieve the optimal rate for estimating $h^\star$ as long as $\|f^\star\|_{\mathcal{F}}^2 \lesssim n_{\mathrm{eff}}^{1/3}$. This clarifies the precise sense in which our result relaxes the usual fixed-radius RKHS-ball assumption.

**$L^\infty$ Results Under a Source Condition.** If one is willing to impose a source condition directly on the *target* RKHS $\mathcal{H}$, our estimator also admits an $L^\infty$ guarantee. The key point is that both the source condition and the eigendecay assumption are imposed only on $\mathcal{H}$, whereas in Singh et al. (2024) the corresponding assumptions are formulated on a substantially more complex joint tensor-product RKHS over $\mathcal{Z}$. This distinction can be important in practice, since the treatment-only target space $\mathcal{H}$ is often expected to exhibit substantially faster eigendecay than the full nuisance-side space.

*Remark* 4.4 ($L^\infty$ Bound under Source Condition). Assume $h^\star \in \mathcal{H}^s$ for some $s \in (1, 2]$, and suppose $\boldsymbol{\Sigma}_{\mathrm{ref}}$ has polynomial eigendecay $\rho_j \lesssim j^{-2\ell}$. Then, under the assumptions of Theorem 4.3, for $\lambda \asymp n_{\mathrm{eff}}^{-1/(s+1/(2\ell))}$, we have

$$\|\hat{h}_\lambda - h^\star\|_{L^\infty(\mathcal{A})} = \widetilde{\mathcal{O}}\left( n_{\mathrm{eff}}^{-\frac{s-1}{2(s+\frac{1}{2\ell})}} \right)$$

with probability at least $1 - n^{-10}$, where the hidden constant may depend on $\|h^\star\|_{\mathcal{H}^s}$ and $\|f^\star\|_{\mathcal{F}}$. A proof is given in Appendix D.

**Challenge of Regularizer Selection.** Achieving the theoretical rates in Theorem 4.3 requires an optimal choice of the regularizer $\lambda$, which balances bias and variance. In practice, however, the optimal $\lambda$ depends on unknown structural properties: the smoothness of the true TEF $h^\star$, the spectral decay of the kernel, and the degree of overlap $\gamma$. To address this, we introduce a data-driven model selection procedure (Algorithm 2) in Section 3.2. In the subsequent section, we establish the adaptivity guarantees of this procedure, demonstrating that it automatically achieves minimax-optimal rates without requiring prior knowledge of these structural parameters.

### 4.2. Adaptivity and Matching Minimax Lower Bounds

We now present the adaptivity guarantee for Algorithm 2.

**Theorem 4.5** (Adaptivity of Model Selection). *Suppose we run Algorithm 2 with $M = n$, $\lambda_0 \asymp \log n/n$, $\mathcal{P}_{\mathrm{samp}} = \mathcal{P}_{\mathrm{ref}}$ and a geometric grid $\Lambda := \{\frac{2^{k-1} \log n}{n} : k = 1, \ldots, L\}$, where $L = \lceil \log_2 n \rceil + 1$. Under the assumptions of Theorem 4.3, with probability at least $1 - 2n^{-10}$, the selected estimator $\hat{h}_{\hat{\lambda}}$ satisfies the following:*

- *Case (a):*   $\mathcal{E}(\hat{h}_{\hat{\lambda}}) \lesssim (n_{\mathrm{eff}})^{-\frac{2\ell}{1+2\ell}} + \frac{1}{n_{\mathrm{eff}}}\|f^\star\|_{\mathcal{F}}^2$;

- *Case (b):*   $\mathcal{E}(\hat{h}_{\hat{\lambda}}) \lesssim \frac{1}{n_{\mathrm{eff}}}(1 + \|f^\star\|_{\mathcal{F}}^2)$;

- *Case (c):*   $\mathcal{E}(\hat{h}_{\hat{\lambda}}) \lesssim \frac{D}{n_{\mathrm{eff}}} + \frac{\|f^\star\|_{\mathcal{F}}^2}{n_{\mathrm{eff}}}$.

*The notation $\lesssim$ hides constants and logarithmic factors.*

The proof is provided in Appendix B. Our analysis demonstrates that the proposed procedure automatically selects the optimal main regularizer for the second-stage KRR, achieving the minimax-optimal rates established in Theorem 4.3. Crucially, the estimator $\hat{h}_{\hat{\lambda}}$ adapts to the unknown structural parameters—specifically the degree of overlap $\gamma$ and the spectral decay of the kernel integral operator—without requiring prior knowledge.

Our model-selection procedure shares the same high-level held-out validation idea as Wang (2026), but the technical

setting is substantially different. Their analysis studies regularizer tuning for classical single-stage KRR under covariate shift, whereas our first contribution is an error analysis for a two-stage KRR estimator of the treatment effect function, followed by model selection built specifically for this two-stage structure.

To complete the adaptivity picture, we now establish a matching minimax lower bound, showing both that the lower bound is governed by the complexity of the target RKHS $\mathcal{H}$ and that the dependence on the effective sample size $n_{\text{eff}} = \gamma n$ is fundamental to the problem and cannot be improved. We derive the minimax lower bound for the Sobolev class $H^\beta([0,1])$.

**Theorem 4.6** (Minimax Lower Bound). *Let $\mathcal{A} = [0,1]$ and $\mathcal{Z} = [0,1]^d$. Let $\mathcal{P}(\gamma)$ denote the set of distributions satisfying Relative Overlap with degree $\gamma$ with respect to the uniform measure. For Sobolev classes $\mathcal{H} = H^\beta(\mathcal{A})$ and $\mathcal{F} = H^\beta(\mathcal{Z})$, the minimax MISE with respect to the uniform measure satisfies, for all sufficiently large $n$,*

$$\inf_{\hat{h}} \sup_{\substack{\|f^\star\|_{\mathcal{F}} \leq 1 \\ \mathcal{P} \in \mathcal{P}(\gamma)}} \mathbb{E}[\mathcal{E}(\hat{h})] \gtrsim (\gamma n)^{-\frac{2\beta}{1+2\beta}} = n_{\text{eff}}^{-\frac{2\beta}{1+2\beta}}. \quad (4.2)$$

The proof can be found in Appendix C. This lower bound matches our upper bound in Theorem 4.3 (Case (a)), confirming that our estimator is minimax-optimal with respect to both the sample size $n$ and the overlap parameter $\gamma$.

## 5. Experiments

We evaluate the proposed estimator on both synthetic and semi-real benchmarks. Code and data-processing scripts are available at https://github.com/seokjinkim0428/CTE. Additional implementation details and hyperparameter grids are deferred to Appendix E.

### 5.1. Synthetic Data

We first study a confounded synthetic benchmark with a known ground-truth TEF, which allows us to isolate the contribution of the second-stage smoothing step relative to standard plug-in estimation.

**Data-Generating Process.** Let $x = (x^1, \ldots, x^q) \in \mathbb{R}^q$ with $q = 10$, and draw $x_i \overset{\text{i.i.d.}}{\sim} \mathcal{U}([-1,1]^q)$. Define $s(x) := \frac{1}{q} \sum_{i=1}^q \sin(x^i)$ and set the true conditional mean to

$$f^\star(x,a) = \sin(a) + 4s(x)(\sin(a) + 1).$$

This yields the target TEF $h^\star(a) = \sin(a)$. To induce confounding, let $r(x) = \sigma(2\sum_{i=1}^q x^i)$, where $\sigma(\cdot)$ denotes the sigmoid function. Conditional on $x$, we sample

$u \sim \text{Beta}(20r(x), 20(1-r(x)))$ and map it to the treatment domain $\mathcal{A} = [-\pi, \pi]$ via $a = \pi(2u - 1)$. Finally, outcomes are generated as $y_i = f^\star(x_i, a_i) + \varepsilon_i$ with $\varepsilon_i \sim \mathcal{N}(0,1)$. We report MISE for $n \in \{500, 1000\}$.

**Baselines.** We compare with two natural baselines: *Plug-in KRR*, which estimates $f^\star$ and directly marginalizes it without a second smoothing stage, and *Direct Regression*, which regresses $y$ on $a$ alone and therefore ignores confounding. All methods use comparable data-driven tuning; the exact grids are reported in Appendix E.

**Results.** Table 1 presents the MISE with respect to the uniform distribution on $\mathcal{A} = [-\pi, \pi]$, averaged over 100 independent runs. In implementation, the integral is approximated by averaging squared errors over a dense uniform grid on the treatment domain. Our proposed method consistently outperforms both the Plug-in baseline and Direct Regression across both sample sizes, confirming the efficacy of the second-stage smoothing procedure.

*Table 1.* Comparison of MISE over 100 runs (standard error in parentheses) on synthetic data. Reported values are scaled by 100.

| Method | $n = 1000$ | $n = 500$ |
|---|---|---|
| **Ours** | 6.22 (0.32) | 8.40 (0.35) |
| Plug-in | 9.87 (0.31) | 11.81 (0.28) |
| Direct Regression | 14.52 (0.31) | 14.84 (0.36) |

### 5.2. Semi-real Data

We further evaluate our methodology on a semi-real benchmark derived from the Job Corps dataset, following the same empirical benchmark specification as Colangelo & Lee (2026). The Job Corps dataset is a well-established benchmark in the continuous treatment effect literature (Singh et al., 2024; Colangelo & Lee, 2026; Huber et al., 2020; Hsu et al., 2026; Lee, 2018; Flores et al., 2012). Funded by the U.S. Department of Labor, the Job Corps program is the largest publicly funded job training program for disadvantaged youth in the United States. The data come from the National Job Corps Study, conducted during the program's randomized evaluation period (November 1994–February 1996). While access to the program was randomized among eligible applicants, the intensity of participation was self-selected. This makes it a canonical benchmark for continuous treatment effect estimation under a selection-on-observables assumption. The relevant variables are defined as follows:

- **Treatment ($a$):** Total hours spent in academic or vocational training classes during the first year after randomization.

- **Outcome** ($y$)**:** The proportion of weeks employed during the second year after randomization.

- **Covariates** ($x$)**:** A 40-dimensional vector ($x \in \mathbb{R}^{40}$) of baseline characteristics, including age, gender, ethnicity, language ability, education, marital status, household size, household income, prior receipt of public assistance, family background, and health-related behaviors.

**Benchmark Construction.** To obtain a benchmark with known ground truth while preserving the empirical covariate-treatment structure, we adopt a semi-synthetic construction. Let the original Job Corps sample be $\{x_{\text{job},i}, a_{\text{job},i}, y_{\text{job},i}\}_{i=1}^n$ with $n = 4024$. We first fit a Generalized Random Forest (GRF) regressor (Athey et al., 2019), denoted by $\hat{f}_{\text{semi}}$, to the original covariates, treatments, and outcomes, and then compute residuals

$$g_i := y_{\text{job},i} - \hat{f}_{\text{semi}}(x_{\text{job},i}, a_{\text{job},i}).$$

The semi-real dataset $\tilde{\mathcal{D}} = \{x_{\text{job},i}, a_{\text{job},i}, \tilde{y}_i\}_{i=1}^n$ is obtained by injecting randomized noise into the fitted response,

$$\tilde{y}_i = \hat{f}_{\text{semi}}(x_{\text{job},i}, a_{\text{job},i}) + \tilde{e}_i g_i,$$

where $\{\tilde{e}_i\}_{i=1}^n$ are i.i.d. Rademacher random variables. We follow Colangelo & Lee (2026) and evaluate MISE on the treatment grid $\{160, 200, \ldots, 2000\}$. The ground-truth marginal dose-response curve is defined by averaging the fitted response surface over the empirical covariate distribution:

$$h^\star(a) := \frac{1}{n} \sum_{i=1}^n \hat{f}_{\text{semi}}(x_{\text{job},i}, a),$$

for $a \in \mathcal{A}$. The reported MISE therefore measures the discrepancy between the estimated treatment effects and this ground-truth curve on the evaluation grid.

**Baselines and Implementation.** We compare against Plug-in KRR and Direct Regression, both tuned by LOOCV, as well as the DML estimators of Colangelo & Lee (2026) with neural-network (NN), kernel-neural-network (KNN), GRF, and LASSO nuisance learners. For DML, we use the tuning parameters and bandwidths reported in Colangelo & Lee (2026). For the kernel-based methods, we use Laplace/Matérn kernels with data-driven selection of length scales and regularization. Full preprocessing and hyperparameter details are reported in Appendix E.

**Results.** Table 2 summarizes the results over 100 independent runs. Our method achieves the lowest MISE among all baselines. Notably, although the ground-truth curve is constructed from the GRF-fitted response surface $\hat{f}_{\text{semi}}$, our kernel-based approach still outperforms DML (GRF) (1.24 vs. 2.42). This highlights the ability of our two-stage smoothing procedure to recover the structure of the treatment effect function even under model misspecification.

*Table 2.* MISE (standard errors in parentheses) for the semi-real benchmark, averaged over 100 independent runs. Our proposed method yields the lowest error among all compared baselines.

| Method | Mean MISE (SE) |
|---|---|
| **Ours** | **1.2466 (0.1209)** |
| Plug-in LOOCV | 1.6197 (0.1146) |
| Direct Regression | 1.6970 (0.1264) |
| DML (GRF) | 2.4230 (0.1837) |
| DML (NN) | 2.1065 (0.1454) |
| DML (LASSO) | 2.8732 (0.2391) |
| DML (KNN) | 2.9742 (0.2165) |

## 6. Conclusion

In this work, we introduce a two-stage kernel ridge regression framework for estimating continuous treatment effects under confounding. Our primary theoretical contribution is to show that the statistical complexity of estimating the TEF is governed by the complexity of the target function space $\mathcal{H}$, rather than the potentially high-dimensional nuisance space $\mathcal{F}$. By constructing pseudo-outcomes and then applying a second-stage smoother, our estimator achieves minimax-optimal convergence rates that adapt to both the intrinsic regularity of the TEF and the degree of overlap. Furthermore, we develop a fully data-driven model selection procedure that attains these optimal rates without requiring prior knowledge of the structural parameters.

Several promising directions remain for future research. First, our current approach is predicated on the assumption of a well-specified outcome model. To improve robustness against model misspecification, future work could investigate incorporating inverse propensity weighting or developing doubly-robust kernel estimators that guarantee consistency if either the outcome or treatment density model is correct. Second, while our analysis leverages the tractability of RKHS theory, extending this "decoupling" strategy to general function classes, such as neural networks beyond the NTK regime, represents an important step toward broader practical applicability.

## Acknowledgement

We thank the reviewers for their thoughtful and constructive feedback, which helped improve the presentation of this work. The research is supported by NSF grant DMS-2515679.

## Impact Statement

This paper presents work whose goal is to advance the field of Machine Learning. There are many potential societal consequences of our work, none of which we feel must be

specifically highlighted here.

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

# A. Proof of Theorem 4.3

## A.1. Groundwork for the Proof

**Proof Roadmap.** We first define high-probability events controlling empirical covariance operators and Monte Carlo fluctuations. On this good event, we derive an error decomposition for $\hat{h}_\lambda - h^\star$, and bound each term separately: pseudo-outcome bias ($\mathscr{B}$), second-stage ridge bias ($\mathscr{C}$), and propagated first-stage error ($\mathscr{P}$). This yields Equation (A.18), and the theorem follows by optimizing $\lambda$ under each eigendecay regime.

**Linear Model in RKHS.** We recall the key definitions. For the function classes $\mathcal{F}$ and $\mathcal{H}$, we denote the associated Hilbert spaces by $\mathbb{F}$ and $\mathbb{H}$, respectively. For $\mathbb{F}$, the feature map is $\psi(\cdot) : \mathcal{Z} \to \mathbb{F}$. For $\mathbb{H}$, the feature map is $\phi(\cdot) : \mathcal{A} \to \mathbb{H}$. Recall $\mathcal{Z} = \mathcal{X} \times \mathcal{A}$. For any $f_\theta \in \mathcal{F}$, we refer to $\theta$ as its Hilbertian element; similarly, for any $h_\eta \in \mathcal{H}$, $\eta$ denotes its Hilbertian element.

We recast our response model in Hilbertian form. Let $\mathcal{D} = \{(x_i, a_i, y_i)\}_{i=1}^n$ with

$$y_i = f^\star(x_i, a_i) + \varepsilon_i$$

where

$$f^\star(x, a) := \psi(x, a)^\top \theta^\star$$

and $\theta^\star$ is the Hilbertian element corresponding to $f^\star$. For the TEF, for any $a \in \mathcal{A}$, we write

$$h^\star(a) = \mathbb{E}[Y(a)] = \phi(a)^\top \eta^\star$$

where $\eta^\star$ is the Hilbertian element of $h^\star$ in $\mathbb{H}$.

**Design Operators.** Consider $N$ generic elements $\{v_1, \ldots, v_N\}$ in a Hilbert space $\mathbb{X}$, with $N > 1$. We define the *design operator* of $\{v_1, \ldots, v_N\}$ as $\mathbf{V} : \mathbb{X} \to \mathbb{R}^N$, which, for all $\theta \in \mathbb{X}$, satisfies:

$$\mathbf{V}\theta = (\langle v_1, \theta \rangle, \langle v_2, \theta \rangle, \ldots, \langle v_N, \theta \rangle)^\top.$$

Similarly, we define the adjoint of $\mathbf{V}$, denoted by $\mathbf{V}^\top : \mathbb{R}^N \to \mathbb{X}$, as the operator such that for all $\mathbf{a} = (a_1, \ldots, a_N) \in \mathbb{R}^N$,

$$\mathbf{V}^\top \mathbf{a} = \sum_{i=1}^N a_i v_i \in \mathbb{X}.$$

Recall that we defined the second moment of $\mathcal{P}_{\text{ref}}$ as

$$\boldsymbol{\Sigma}_{\text{ref}} := \mathbb{E}_{a \sim \mathcal{P}_{\text{ref}}}[\phi(a) \otimes \phi(a)].$$

We also define

$$\boldsymbol{\Sigma}_{\text{samp}} := \mathbb{E}_{a \sim \mathcal{P}_{\text{samp}}}[\phi(a) \otimes \phi(a)].$$

When we take $\mathcal{P}_{\text{samp}} = \mathcal{P}_{\text{ref}}$ (in Theorem 4.3), we have

$$\boldsymbol{\Sigma}_{\text{samp}} = \boldsymbol{\Sigma}_{\text{ref}}.$$

**Summary of Notation.** We define the design operators of $\{\phi(a_j')\}_{j=1}^n$ and $\{\psi(x_i, a_i)\}_{i=1}^n$ as $\mathbf{A}$ and $\mathbf{Z}$, respectively. Also, we define

$$\bar{\psi}(a) := \mathbb{E}_{x \sim \mathcal{P}_X}[\psi(x, a)]$$

and the empirical feature mean as

$$\hat{\bar{\psi}}(a) := \frac{1}{n} \sum_{i=1}^n \psi(x_i, a).$$

We define the design operator of $\{\hat{\bar{\psi}}(a'_j)\}_{j=1}^n$ as $\mathbf{W}$. We summarize the key notation below in table 3.

*Table 3.* Summary of notation

| Notation | Description |
|---|---|
| $\mathbf{Z}$ | Design operator of $\{\psi(x_i, a_i)\}_{i=1}^n$ |
| $\mathbf{A}$ | Design operator of $\{\phi(a'_j)\}_{j=1}^n$ |
| $\bar{\psi}(a)$ | Feature mean, defined as $\mathbb{E}_{x \sim \mathcal{P}_X}[\psi(x, a)]$ |
| $\hat{\bar{\psi}}(a)$ | Empirical feature mean, defined as $\frac{1}{n} \sum_{i=1}^n \psi(x_i, a)$ |
| $\mathbf{W}$ | Design operator of $\{\hat{\bar{\psi}}(a'_j)\}_{j=1}^n$ |
| $\mathbf{Y}$ | Vector of responses $y_1, \ldots, y_n$ ($\mathbf{Y} \in \mathbb{R}^n$) |
| $\theta^\star$ | Hilbertian element of $f^\star$ in $\mathbb{F}$ |
| $\eta^\star$ | Hilbertian element of $h^\star$ in $\mathbb{H}$ |
| $\mathbf{\Sigma}_{\text{ref}}$ | $\mathbb{E}_{a \sim \mathcal{P}_{\text{ref}}}[\phi(a) \otimes \phi(a)]$ |
| $\mathbf{\Sigma}_{\text{samp}}$ | $\mathbb{E}_{a \sim \mathcal{P}_{\text{samp}}}[\phi(a) \otimes \phi(a)]$ |
| $\mathbf{S}_\lambda$ | Regularized second-moment operator $\mathbf{\Sigma}_{\text{ref}}^{1/2}(\mathbf{\Sigma}_{\text{ref}} + \lambda\mathbf{I})^{-1}\mathbf{\Sigma}_{\text{ref}}^{1/2}$ |
| $\hat{\theta}$ | Corresponding Hilbertian element of $\hat{f}$ in Algorithm 1 |
| $\hat{\eta}_\lambda$ | Corresponding Hilbertian element of $\hat{h}_\lambda$ in Algorithm 1 |

**First-Stage Regression: Hilbertian Formulation.** Recall that we define the design operator of $\{\psi(x_i, a_i)\}_{i=1}^n$ as $\mathbf{Z}$. We run kernel ridge regression to obtain the estimator $\hat{\theta}$:

$$\hat{\theta} = (\mathbf{Z}^\top\mathbf{Z} + n\lambda_0\mathbf{I})^{-1}\mathbf{Z}^\top\mathbf{Y}.$$

Then $\hat{\theta}$ is the corresponding Hilbertian element of $\hat{f} \in \mathcal{F}$ in Algorithm 1.

**Second-Stage Regression: Hilbertian Formulation.** In the second stage of Algorithm 1, we regress on the pseudo-outcomes $m_j := m(a'_j)$. We express this process in a closed form using the language of elements and operators in a Hilbert space. We define $\hat{\eta}_\lambda$ as the corresponding Hilbertian element of $\hat{h}_\lambda$.

1. Sample $\{a'_j\}_{j=1}^n$ from $\mathcal{P}_{\text{samp}}$ and create pseudo-outcomes $\{m_j := m(a'_j)\}_{j=1}^n$.

   Explicitly, the pseudo-outcome for $a'_j$ is:

   $$m(a'_j) := \frac{1}{n} \sum_{i=1}^n \psi(x_i, a'_j)^\top \hat{\theta} = \hat{\bar{\psi}}(a'_j)^\top \hat{\theta}.$$

2. Define the design operator of $\{\phi(a'_j)\}_{j=1}^n$ as $\mathbf{A}$. Obtain the final estimator

   $$\hat{\eta}_\lambda = (\mathbf{A}^\top\mathbf{A} + n\lambda\mathbf{I})^{-1} \sum_{j=1}^n \phi(a'_j)m(a'_j)$$

   $$= (\mathbf{A}^\top\mathbf{A} + n\lambda\mathbf{I})^{-1}\mathbf{A}^\top\mathbf{W}\hat{\theta}.$$

   for a regularizer $\lambda > 0$.

For a generic $a$, we define

$$m(a) := \frac{1}{n} \sum_{i=1}^n \hat{f}(x_i, a).$$

## A.2. Good Events for Proof

We now define the high-probability events used in the proof. Specifically, we define $\mathscr{E}_1, \mathscr{E}_2, \mathscr{E}_3$, and finally $\mathscr{E}_{\text{good}} := \mathscr{E}_1 \cap \mathscr{E}_2 \cap \mathscr{E}_3$.

**Good Event 1: $\mathscr{E}_1$** We define the good event $\mathscr{E}_1$ as follows. First, sample $a_1', \ldots, a_n'$ from $\mathcal{P}_{\text{samp}}$, independently of $\{x_1, \ldots, x_n\}$. Then, with probability at least $1 - \frac{1}{n^{11}}$, by Hoeffding's inequality and a union bound over $j \in [n]$, the following holds simultaneously for all $j \in [n]$:

$$\left| \frac{1}{n} \sum_{i=1}^n f^\star(x_i, a_j') - \mathbb{E}_{x \sim \mathcal{P}_X}[f^\star(x, a_j')] \right| \lesssim \frac{\sqrt{\xi} \|\theta^\star\|_{\mathbb{F}} \sqrt{\log n}}{\sqrt{n}}. \tag{A.1}$$

We can apply Hoeffding's inequality since for all $x \in \mathcal{X}, a \in \mathcal{A}$,

$$|f^\star(x, a)| = |\psi(x, a)^\top \theta^\star| \leq \|\psi(x, a)\|_{\mathbb{F}} \|\theta^\star\|_{\mathbb{F}} \leq \sqrt{\xi} \|\theta^\star\|_{\mathbb{F}}.$$

**Definition A.1** (Good event $\mathscr{E}_1$). We define the good event $\mathscr{E}_1$ as the event where Equation (A.1) holds for all $j \in [n]$. From the above observation, we have $\mathbb{P}[\mathscr{E}_1] \geq 1 - n^{-11}$.

**Good Event 2: $\mathscr{E}_2$** Next, we define the second good event $\mathscr{E}_2$. For a fixed $a$, we define

$$\widehat{\mathbf{Q}}(a) := \frac{1}{n} \sum_{i=1}^n \psi(x_i, a) \psi(x_i, a)^\top.$$

Also, we define

$$\mathbf{Q}(a) = \mathbb{E}_{x \sim \mathcal{P}_X}[\psi(x, a) \psi(x, a)^\top].$$

Using Lemma F.2, with probability at least $1 - \frac{1}{n^{11}}$, we have

$$\frac{1}{c}\left(\widehat{\mathbf{Q}}(a_j') + \frac{\log n}{n} \mathbf{I}\right) \preceq \mathbf{Q}(a_j') + \frac{\log n}{n} \mathbf{I} \preceq c\left(\widehat{\mathbf{Q}}(a_j') + \frac{\log n}{n} \mathbf{I}\right) \tag{A.2}$$

for all $j \in [n]$ for some absolute constant $c > 1$. We set

$$\mathbf{Q} := \mathbb{E}_{a_j' \sim \mathcal{P}_{\text{samp}}}[\mathbf{Q}(a_j')] = \mathbb{E}_{\mathcal{P}_X \otimes \mathcal{P}_{\text{samp}}}[\psi(x, a) \psi(x, a)^\top]$$

and

$$\bar{\mathbf{Q}} := \frac{1}{n} \sum_{j=1}^n \mathbf{Q}(a_j').$$

**Lemma A.2.** *There exists an absolute constant $c > 1$ such that, with probability at least $1 - n^{-11}$,*

$$\bar{\mathbf{Q}} \preceq c\left(\mathbf{Q} + \frac{\log n}{n} \mathbf{I}\right) \tag{A.3}$$

*holds.*

*Proof.* For $\lambda' > 0$, define

$$\mathbf{T}_j := (\mathbf{Q} + \lambda' \mathbf{I})^{-\frac{1}{2}} (\mathbf{Q}(a_j') - \mathbf{Q})(\mathbf{Q} + \lambda' \mathbf{I})^{-\frac{1}{2}}, \qquad j \in [n].$$

Then $\mathbb{E}[\mathbf{T}_j] = 0$. By bounded features, $\|\mathbf{Q}(a)\|_{\text{op}} \leq \xi$, so in Lemma D.3 of Wang (2026) we can take $U \asymp \frac{\xi}{\lambda'}$ since

$$\|\mathbf{T}_j\|_{\text{op}} \lesssim \frac{\xi}{\lambda'}.$$

For the variance proxy, let

$$X_j := (\mathbf{Q} + \lambda'\mathbf{I})^{-\frac{1}{2}}\mathbf{Q}(a'_j)(\mathbf{Q} + \lambda'\mathbf{I})^{-\frac{1}{2}}, \qquad M := \mathbb{E}[X_j] = (\mathbf{Q} + \lambda'\mathbf{I})^{-\frac{1}{2}}\mathbf{Q}(\mathbf{Q} + \lambda'\mathbf{I})^{-\frac{1}{2}}.$$

Then $\mathbf{T}_j = X_j - M$, and

$$\mathbb{E}[\mathbf{T}_j^2] = \mathbb{E}[(X_j - M)^2] = \mathbb{E}[X_j^2] - M^2 \preceq \mathbb{E}[X_j^2]$$

since $M^2 \succeq 0$. By monotonicity of $\|\cdot\|_{\mathrm{op}}$ on positive semidefinite operators, we get

$$
\begin{aligned}
\|\mathbb{E}[\mathbf{T}_j^2]\|_{\mathrm{op}} &\leq \|\mathbb{E}[(\mathbf{Q}+\lambda'\mathbf{I})^{-\frac{1}{2}}\mathbf{Q}(a'_j)(\mathbf{Q}+\lambda'\mathbf{I})^{-\frac{1}{2}}(\mathbf{Q}+\lambda'\mathbf{I})^{-\frac{1}{2}}\mathbf{Q}(a'_j)(\mathbf{Q}+\lambda'\mathbf{I})^{-\frac{1}{2}}]\|_{\mathrm{op}} \\
&= \|\mathbb{E}[(\mathbf{Q}+\lambda'\mathbf{I})^{-\frac{1}{2}}\mathbf{Q}(a'_j)(\mathbf{Q}+\lambda'\mathbf{I})^{-1}\mathbf{Q}(a'_j)(\mathbf{Q}+\lambda'\mathbf{I})^{-\frac{1}{2}}]\|_{\mathrm{op}} \\
&\overset{(i)}{\lesssim} \frac{\xi}{\lambda'}\|\mathbb{E}[(\mathbf{Q}+\lambda'\mathbf{I})^{-\frac{1}{2}}\mathbf{Q}(a'_j)^{\frac{1}{2}}\mathbf{Q}(a'_j)^{\frac{1}{2}}(\mathbf{Q}+\lambda'\mathbf{I})^{-\frac{1}{2}}]\|_{\mathrm{op}} \\
&\lesssim \frac{\xi}{\lambda'}\|\mathbb{E}[(\mathbf{Q}+\lambda'\mathbf{I})^{-\frac{1}{2}}\mathbf{Q}(a'_j)(\mathbf{Q}+\lambda'\mathbf{I})^{-\frac{1}{2}}]\|_{\mathrm{op}} \\
&\lesssim \frac{\xi}{\lambda'}\|(\mathbf{Q}+\lambda'\mathbf{I})^{-\frac{1}{2}}\mathbf{Q}(\mathbf{Q}+\lambda'\mathbf{I})^{-\frac{1}{2}}\|_{\mathrm{op}} \\
&\lesssim \frac{\xi}{\lambda'}.
\end{aligned}
$$

Hence $\|\sum_{j=1}^n \mathbb{E}[\mathbf{T}_j^2]\|_{\mathrm{op}} \lesssim n\xi/\lambda'$. In step (i), we use the fact that

$$\|\mathbf{Q}(a'_j)^{\frac{1}{2}}(\mathbf{Q}+\lambda'\mathbf{I})^{-1}\mathbf{Q}(a'_j)^{\frac{1}{2}}\|_{\mathrm{op}} \leq \frac{1}{\lambda'}\|\mathbf{Q}(a'_j)\|_{\mathrm{op}} \leq \frac{\xi}{\lambda'}.$$

Let

$$\mathbf{J} := (\mathbf{Q} + \lambda'\mathbf{I})^{-1}.$$

By cyclicity of the trace, for each $j$,

$$
\begin{aligned}
\mathrm{Tr}(\mathbf{T}_j^2) &= \mathrm{Tr}\left((\mathbf{Q}+\lambda'\mathbf{I})^{-1/2}\Delta_j(\mathbf{Q}+\lambda'\mathbf{I})^{-1}\Delta_j(\mathbf{Q}+\lambda'\mathbf{I})^{-1/2}\right) \\
&= \mathrm{Tr}\left(\Delta_j\,\mathbf{J}\,\Delta_j\,\mathbf{J}\right),
\end{aligned}
\tag{A.4}
$$

where $\Delta_j := \mathbf{Q}(a'_j) - \mathbf{Q}$. Since $\{a'_j\}_{j=1}^n$ are i.i.d., we have

$$\mathrm{Tr}\left(\sum_{j=1}^n \mathbb{E}[\mathbf{T}_j^2]\right) = n\,\mathbb{E}\,\mathrm{Tr}(\mathbf{T}_1^2) = n\,\mathbb{E}\,\mathrm{Tr}\left(\Delta\,\mathbf{J}\,\Delta\,\mathbf{J}\right), \tag{A.5}$$

with generic $\Delta := \mathbf{Q}(a') - \mathbf{Q}$ for $a' \sim \mathcal{P}_{\mathrm{samp}}$.

Next, using Hilbert–Schmidt norms,

$$\mathrm{Tr}(\Delta\mathbf{J}\Delta\mathbf{J}) = \|\mathbf{J}^{1/2}\Delta\mathbf{J}^{1/2}\|_{\mathrm{HS}}^2,$$

and the inequality $\|U - V\|_{\mathrm{HS}}^2 \leq 2\|U\|_{\mathrm{HS}}^2 + 2\|V\|_{\mathrm{HS}}^2$ with $U = \mathbf{J}^{1/2}\mathbf{Q}(a')\mathbf{J}^{1/2}$ and $V = \mathbf{J}^{1/2}\mathbf{Q}\mathbf{J}^{1/2}$, we obtain

$$\mathrm{Tr}(\Delta\mathbf{J}\Delta\mathbf{J}) \leq 2\,\mathrm{Tr}\left(\mathbf{Q}(a')\,\mathbf{J}\,\mathbf{Q}(a')\,\mathbf{J}\right) + 2\,\mathrm{Tr}\left(\mathbf{Q}\,\mathbf{J}\,\mathbf{Q}\,\mathbf{J}\right). \tag{A.6}$$

We now bound $\mathrm{Tr}(A\mathbf{J}A\mathbf{J})$ for a generic PSD operator $A \succeq 0$:

$$
\begin{aligned}
\mathrm{Tr}(A\mathbf{J}A\mathbf{J}) &= \mathrm{Tr}\left((\mathbf{J}^{1/2}A\mathbf{J}^{1/2})^2\right) \\
&\leq \|\mathbf{J}^{1/2}A\mathbf{J}^{1/2}\|_{\mathrm{op}}\,\mathrm{Tr}(\mathbf{J}^{1/2}A\mathbf{J}^{1/2}) \\
&= \|\mathbf{J}^{1/2}A\mathbf{J}^{1/2}\|_{\mathrm{op}}\,\mathrm{Tr}(A\mathbf{J}) \\
&\leq \|A\|_{\mathrm{op}}\,\|\mathbf{J}\|_{\mathrm{op}}\,\mathrm{Tr}(A\mathbf{J}) \\
&\leq \frac{\|A\|_{\mathrm{op}}}{\lambda'}\,\mathrm{Tr}\left(A(\mathbf{Q}+\lambda'\mathbf{I})^{-1}\right),
\end{aligned}
\tag{A.7}
$$

since $\|\mathbf{J}\|_{\mathrm{op}} = \|(\mathbf{Q} + \lambda'\mathbf{I})^{-1}\|_{\mathrm{op}} \leq 1/\lambda'$.

Under the bounded-feature assumption $\|\psi(x,a)\|_{\mathbb{F}}^2 \leq \xi$, we have $\|\mathbf{Q}(a)\|_{\mathrm{op}} \leq \xi$ and $\|\mathbf{Q}\|_{\mathrm{op}} \leq \xi$. Applying (A.7) with $A = \mathbf{Q}(a')$ and $A = \mathbf{Q}$, and then taking expectation over $a'$, yields

$$\mathbb{E} \operatorname{Tr}\left(\mathbf{Q}(a')\mathbf{J}\mathbf{Q}(a')\mathbf{J}\right) \leq \frac{\xi}{\lambda'} \mathbb{E} \operatorname{Tr}\left(\mathbf{Q}(a')(\mathbf{Q} + \lambda'\mathbf{I})^{-1}\right) = \frac{\xi}{\lambda'} \operatorname{Tr}\left(\mathbf{Q}(\mathbf{Q} + \lambda'\mathbf{I})^{-1}\right), \tag{A.8}$$

$$\operatorname{Tr}\left(\mathbf{Q}\mathbf{J}\mathbf{Q}\mathbf{J}\right) \leq \frac{\xi}{\lambda'} \operatorname{Tr}\left(\mathbf{Q}(\mathbf{Q} + \lambda'\mathbf{I})^{-1}\right). \tag{A.9}$$

Combining (A.5), (A.6), (A.8), and (A.9), we conclude that

$$\operatorname{Tr}\left(\sum_{j=1}^{n} \mathbb{E}[\mathbf{T}_j^2]\right) \leq \frac{4n\xi}{\lambda'} \operatorname{Tr}\left(\mathbf{Q}(\mathbf{Q} + \lambda'\mathbf{I})^{-1}\right) \tag{A.10}$$

Moreover, since $\operatorname{Tr}(\mathbf{Q}) = \mathbb{E}\|\psi\|_{\mathbb{F}}^2 \leq \xi$, we have $\operatorname{Tr}(\mathbf{Q})\|(\mathbf{Q} + \lambda'\mathbf{I})^{-1}\|_{\mathrm{op}} \leq \xi/\lambda'$, and thus the cruder bound

$$\operatorname{Tr}\left(\sum_{j=1}^{n} \mathbb{E}[\mathbf{T}_j^2]\right) \leq \frac{4n\xi^2}{\lambda'^2}. \tag{A.11}$$

By applying Lemma D.3 of Wang (2026), for any $\lambda' \geq c_1 \xi \frac{\log n}{n}$ for some constant $c_1 > 1$, we have the following with probability at least $1 - n^{-11}$:

$$\|(\mathbf{Q} + \lambda'\mathbf{I})^{-\frac{1}{2}}(\bar{\mathbf{Q}} - \mathbf{Q})(\mathbf{Q} + \lambda'\mathbf{I})^{-\frac{1}{2}}\|_{\mathrm{op}} \lesssim \frac{1}{2}.$$

Hence, $\bar{\mathbf{Q}} \preceq 2(\mathbf{Q} + \lambda'\mathbf{I})$, and thus

$$\bar{\mathbf{Q}} \preceq c\left(\mathbf{Q} + \frac{\log n}{n}\mathbf{I}\right) \tag{A.12}$$

holds for some absolute constant $c > 1$ with probability at least $1 - n^{-11}$. $\qquad\square$

**Definition A.3** (Good event $\mathscr{E}_2$). We define the good event $\mathscr{E}_2$ as the event where Equation (A.2) holds for all $j \in [n]$ and Equation (A.3) holds. By the above argument, we have $\mathbb{P}[\mathscr{E}_2] \geq 1 - 2n^{-11}$.

**Good Event 3: $\mathscr{E}_3$.** Using Lemma F.2, with probability at least $1 - \frac{1}{n^{11}}$, for some absolute constant $c > 1$, we have

$$\frac{1}{c}(\mathbf{A}^\top\mathbf{A} + \log n \cdot \mathbf{I}) \preceq n\mathbf{\Sigma}_{\mathrm{samp}} + \log n \cdot \mathbf{I} \preceq c(\mathbf{A}^\top\mathbf{A} + \log n \cdot \mathbf{I}). \tag{A.13}$$

Similarly, using Lemma F.2, with probability at least $1 - \frac{1}{n^{11}}$, we have

$$\frac{1}{c}(\mathbf{Z}^\top\mathbf{Z} + \log n \cdot \mathbf{I}) \preceq n\mathbb{E}_{(x,a)\sim\mathcal{P}_{X,A}}[\psi(x,a)\psi(x,a)^\top] + \log n \cdot \mathbf{I} \preceq c(\mathbf{Z}^\top\mathbf{Z} + \log n \cdot \mathbf{I}) \tag{A.14}$$

for some absolute constant $c > 1$.

**Definition A.4** (Good event $\mathscr{E}_3$). We define the good event $\mathscr{E}_3$ as the event where Equation (A.13) and Equation (A.14) both hold. By the previous argument, we have $\mathbb{P}[\mathscr{E}_3] \geq 1 - 2n^{-11}$.

**Final Good Event: $\mathscr{E}_{\mathrm{good}}$.** Finally, we define the final good event as the intersection of $\mathscr{E}_1$, $\mathscr{E}_2$, and $\mathscr{E}_3$.

**Definition A.5** (Final good event). We define $\mathscr{E}_{\mathrm{good}} := \mathscr{E}_1 \cap \mathscr{E}_2 \cap \mathscr{E}_3$. We emphasize that this event depends only on the randomness of $(x_i, a_i)$ for $i \in [n]$ and $a'_j$ for $j \in [n]$, not on the randomness of the noise variables.

**Lemma A.6** (Good event). *We have*

$$\mathbb{P}[\mathscr{E}_{\mathrm{good}}] \geq 1 - \frac{5}{n^{11}}.$$

*Proof.* This follows directly from the previous arguments via a union bound. $\qquad\square$

**Key Inequality Under Event $\mathscr{E}_{\text{good}}$.** Next, we present a key property of the design operator $\mathbf{W}$ under the final good event $\mathscr{E}_{\text{good}}$.

**Lemma A.7** (Upper bound of $\mathbf{W}^\top \mathbf{W}$). *Under the event $\mathscr{E}_{\text{good}}$, we have*

$$\frac{1}{n}\mathbf{W}^\top\mathbf{W} \preceq c\frac{1}{n\gamma}(\mathbf{Z}^\top\mathbf{Z} + \log n\mathbf{I})$$

*for some absolute constant $c > 1$.*

*Proof.* We first observe that by Jensen/Cauchy-Schwarz,

$$\frac{1}{n}\mathbf{W}^\top\mathbf{W} = \frac{1}{n}\sum_{j=1}^n \hat{\bar{\psi}}(a_j')\hat{\bar{\psi}}(a_j')^\top \preceq \frac{1}{n^2}\sum_{i\in[n],j\in[n]} \psi(x_i, a_j')\psi(x_i, a_j')^\top.$$

This holds because for any $u \in \mathbb{F}$, we have

$$u^\top\mathbf{W}^\top\mathbf{W}u = \sum_{j=1}^n\left(\frac{1}{n}\sum_{i=1}^n \psi(x_i, a_j')^\top u\right)^2$$

$$\leq \sum_{j=1}^n \frac{1}{n}\sum_{i=1}^n (\psi(x_i, a_j')^\top u)^2.$$

Using the definition of $\mathscr{E}_{\text{good}}$, there exist two absolute constants $c_1, c_2 > 1$ such that

$$\frac{1}{n^2}\sum_{i\in[n],j\in[n]} \psi(x_i, a_j')\psi(x_i, a_j')^\top = \frac{1}{n}\sum_{j=1}^n\frac{1}{n}\sum_{i\in[n]} \psi(x_i, a_j')\psi(x_i, a_j')^\top$$

$$:= \frac{1}{n}\sum_{j=1}^n \widehat{\mathbf{Q}}(a_j')$$

$$\preceq c_1\frac{1}{n}\sum_{j=1}^n\left(\mathbf{Q}(a_j') + \frac{\log n}{n}I\right) \quad \text{(by definition of } \mathscr{E}_{\text{good}})$$

$$\preceq c_1\frac{\log n}{n}\mathbf{I} + c_1\frac{1}{n}\sum_{j=1}^n \mathbf{Q}(a_j')$$

$$= c_1\frac{\log n}{n}\mathbf{I} + c_1\bar{\mathbf{Q}}$$

$$\preceq c_2\frac{\log n}{n}\mathbf{I} + c_2\mathbf{Q} \quad \text{(by definition of } \mathscr{E}_{\text{good}})$$

$$= c_2\frac{\log n}{n}\mathbf{I} + c_2\mathbb{E}_{\mathcal{P}_X\otimes\mathcal{P}_{\text{samp}}}[\psi(x, a)\psi(x, a)^\top]$$

$$\preceq c_2\frac{\log n}{n}\mathbf{I} + c_2\frac{1}{\gamma}\mathbb{E}_{(x,a)\sim\mathcal{P}_{X,A}}[\psi(x, a)\psi(x, a)^\top] \quad \text{(by Definition 4.1)}.$$

On the other hand, under the good event $\mathscr{E}_{\text{good}}$, we can lower bound the right-hand side as

$$c\left(\frac{1}{n}\mathbf{Z}^\top\mathbf{Z} + \frac{\log n}{n}\mathbf{I}\right) \succeq \mathbb{E}_{(x,a)\sim\mathcal{P}_{X,A}}[\psi(x, a)\psi(x, a)^\top] + \frac{\log n}{n}\mathbf{I}.$$

By combining these, we obtain the desired result. $\square$

*Remark* A.8 (A useful observation). For any two design operators $\mathbf{T}_1, \mathbf{T}_2$, suppose that

$$\mathbf{T}_1^\top\mathbf{T}_1 \preceq c_1(\mathbf{T}_2^\top\mathbf{T}_2 + c_2\log n\mathbf{I})$$

for some absolute constant $c_1 > 1, c_2 > 0$. Then, for any $c_3 > 0$, for $c = \max(1, c_2/c_3)$, we have

$$\mathbf{T}_1^\top\mathbf{T}_1 \preceq c(\mathbf{T}_2^\top\mathbf{T}_2 + c_3\log n\mathbf{I}).$$

**How We Use Remark A.8.** Second-moment concentration gives shifts of the form $+\log n\mathbf{I}$. When we need $+r\mathbf{I}$ instead (e.g., $r = n\lambda$, $n\lambda_0$, $n_1\lambda$, $n_2\tilde{\lambda}$), we apply Remark A.8 with $c_3 = r/\log n$. If $r \gtrsim \log n$, then $c_3$ is bounded below by an absolute constant, so the resulting multiplicative constant remains absolute.

### A.3. Error Decomposition of Pseudo-Outcomes

For any $a \in \mathcal{A}$, we decompose the estimation error of the pseudo-outcome $m(a)$:

$$m(a) - \phi(a)^\top \eta^\star = m(a) - h^\star(a) = m(a) - \mathbb{E}_{x \sim \mathcal{P}_X}[f^\star(x, a)] = m(a) - \mathbb{E}_{x \sim \mathcal{P}_X}[\psi(x, a)]^\top \theta^\star.$$

Moreover, we have

$$m(a) = \frac{1}{n} \sum_{i=1}^n \hat{f}(x_i, a) = \hat{\bar{\psi}}(a)^\top \hat{\theta}.$$

Thus,

$$
\begin{aligned}
m(a) - \phi(a)^\top \eta^\star &= \frac{1}{n} \sum_{i=1}^n \psi(x_i, a)^\top \hat{\theta} - \frac{1}{n} \sum_{i=1}^n \psi(x_i, a)^\top \theta^\star + \frac{1}{n} \sum_{i=1}^n \psi(x_i, a)^\top \theta^\star - \mathbb{E}_{x \sim \mathcal{P}_X}[\psi(x, a)]^\top \theta^\star \\
&= \hat{\bar{\psi}}(a)^\top \hat{\theta} - \hat{\bar{\psi}}(a)^\top \theta^\star + \underbrace{\hat{\bar{\psi}}(a)^\top \theta^\star - \bar{\psi}(a)^\top \theta^\star}_{:=b(a)} \\
&= \hat{\bar{\psi}}(a)^\top (\hat{\theta} - \theta^\star) + b(a)
\end{aligned}
\tag{A.15}
$$

We define $\boldsymbol{b}$ as the vector $(b(a_1'), \dots, b(a_n'))$ where

$$
\begin{aligned}
b(a_j') &= \hat{\bar{\psi}}(a_j')^\top \theta^\star - \bar{\psi}(a_j')^\top \theta^\star \\
&= \frac{1}{n} \sum_{i=1}^n f^\star(x_i, a_j') - \mathbb{E}_{x \sim \mathcal{P}_X}[f^\star(x, a_j')].
\end{aligned}
$$

Note that under the good event $\mathscr{E}_{\text{good}}$, for all $a_j' \in \{a_j'\}_{j=1}^n$ we have

$$|b(a_j')| \lesssim \frac{\sqrt{\log n}}{\sqrt{n}} \|\theta^\star\|_{\mathbb{F}}.$$

**Lemma A.9** (Error decomposition). *Let $\hat{\eta}_\lambda$ denote the corresponding Hilbertian element of $\hat{h}_\lambda$ in $\mathbb{H}$ (explicitly defined in Appendix A.1). Then we have the following decomposition:*

$$
\begin{aligned}
\hat{\eta}_\lambda - \eta^\star &= (\mathbf{A}^\top \mathbf{A} + n\lambda\mathbf{I})^{-1}\mathbf{A}^\top \boldsymbol{b} + (\mathbf{A}^\top \mathbf{A} + n\lambda\mathbf{I})^{-1}\mathbf{A}^\top \mathbf{W}(\hat{\theta} - \theta^\star) - n\lambda(\mathbf{A}^\top \mathbf{A} + n\lambda\mathbf{I})^{-1}\eta^\star \\
&:= \mathscr{B} + \mathscr{P} + \mathscr{C}.
\end{aligned}
$$

*Proof.* By the established decomposition (A.15),

$$m(a) - \phi(a)^\top \eta^\star = \hat{\bar{\psi}}(a)^\top (\hat{\theta} - \theta^\star) + b(a).$$

Hence we get

$$
\begin{aligned}
\hat{\eta}_\lambda - \eta^\star &= (\mathbf{A}^\top \mathbf{A} + n\lambda\mathbf{I})^{-1} \left( \sum_{j=1}^n \phi(a'_j)m(a'_j) - \mathbf{A}^\top \mathbf{A}\eta^\star - n\lambda\eta^\star \right) \\
&= (\mathbf{A}^\top \mathbf{A} + n\lambda\mathbf{I})^{-1} \left( \sum_{j=1}^n \phi(a'_j)(m(a'_j) - \phi(a'_j)^\top \eta^\star) - n\lambda\eta^\star \right) \\
&= (\mathbf{A}^\top \mathbf{A} + n\lambda\mathbf{I})^{-1} \left( \sum_{j=1}^n \phi(a'_j)(m(a'_j) - h^\star(a'_j)) - n\lambda\eta^\star \right) \\
&\overset{(i)}{=} (\mathbf{A}^\top \mathbf{A} + n\lambda\mathbf{I})^{-1} \sum_{j=1}^n \phi(a'_j)\big(b(a'_j) + \hat{\bar{\psi}}(a'_j)^\top(\hat{\theta} - \theta^\star)\big) - n\lambda(\mathbf{A}^\top \mathbf{A} + n\lambda\mathbf{I})^{-1}\eta^\star \\
&= \underbrace{(\mathbf{A}^\top \mathbf{A} + n\lambda\mathbf{I})^{-1}\mathbf{A}^\top \boldsymbol{b}}_{:=\mathscr{B}} + \underbrace{(\mathbf{A}^\top \mathbf{A} + n\lambda\mathbf{I})^{-1}\mathbf{A}^\top \mathbf{W}(\hat{\theta} - \theta^\star)}_{\mathscr{P}} + \underbrace{(-n\lambda(\mathbf{A}^\top \mathbf{A} + n\lambda\mathbf{I})^{-1}\eta^\star)}_{\mathscr{C}} \\
&:= \mathscr{B} + \mathscr{P} + \mathscr{C}.
\end{aligned}
$$

Step (i) holds by the established decomposition (A.15). $\qquad\square$

Recall that we defined

$$
\begin{aligned}
\mathscr{B} &:= (\mathbf{A}^\top \mathbf{A} + n\lambda\mathbf{I})^{-1}\mathbf{A}^\top \boldsymbol{b} \\
\mathscr{P} &:= (\mathbf{A}^\top \mathbf{A} + n\lambda\mathbf{I})^{-1}\mathbf{A}^\top \mathbf{W}(\hat{\theta} - \theta^\star) \\
\mathscr{C} &:= -n\lambda(\mathbf{A}^\top \mathbf{A} + n\lambda\mathbf{I})^{-1}\eta^\star
\end{aligned}
$$

Here, $\mathscr{P}$, the propagated error term, is our main challenge.

## A.4. Bounding MISE with Respect to $\boldsymbol{\Sigma}_{\mathrm{ref}}$

Recall that $\hat{\eta}_\lambda$ is the Hilbertian element of $\hat{h}_\lambda$, and we define $\mathcal{E}_{\mathbb{H}}(\cdot)$ as follows:

$$
\mathcal{E}(\hat{h}_\lambda) = \|\hat{\eta}_\lambda - \eta^\star\|^2_{\boldsymbol{\Sigma}_{\mathrm{ref}}} := \mathcal{E}_{\mathbb{H}}(\hat{\eta}_\lambda).
$$

Using the established decomposition, we get

$$
\mathcal{E}_{\mathbb{H}}(\hat{\eta}_\lambda) \lesssim \|\boldsymbol{\Sigma}_{\mathrm{ref}}^{\frac{1}{2}}(\mathscr{B} + \mathscr{C} + \mathscr{P})\|^2_{\mathbb{H}} \lesssim \|\boldsymbol{\Sigma}_{\mathrm{ref}}^{\frac{1}{2}}\mathscr{B}\|^2_{\mathbb{H}} + \|\boldsymbol{\Sigma}_{\mathrm{ref}}^{\frac{1}{2}}\mathscr{C}\|^2_{\mathbb{H}} + \|\boldsymbol{\Sigma}_{\mathrm{ref}}^{\frac{1}{2}}\mathscr{P}\|^2_{\mathbb{H}}.
$$

**Definition A.10.** For any $\lambda > 0$, we define

$$
\mathbf{S}_\lambda := \boldsymbol{\Sigma}_{\mathrm{ref}}^{\frac{1}{2}}(\boldsymbol{\Sigma}_{\mathrm{ref}} + \lambda\mathbf{I})^{-1}\boldsymbol{\Sigma}_{\mathrm{ref}}^{\frac{1}{2}}.
$$

**Some Useful Inequalities.** Recall the definition of the good event $\mathscr{E}_{\mathrm{good}}$. In this subsection, every invocation of Remark A.8 uses $r = n\lambda$, so we work in the regime $n\lambda \gtrsim \log n$. Under this event, Remark A.8 implies that for some $c > 1$,

$$
\frac{1}{c}(n\boldsymbol{\Sigma}_{\mathrm{samp}} + n\lambda\mathbf{I}) \preceq \mathbf{A}^\top \mathbf{A} + n\lambda\mathbf{I} \preceq c(n\boldsymbol{\Sigma}_{\mathrm{samp}} + n\lambda\mathbf{I}).
$$

Moreover, we observe that

$$
\begin{aligned}
\|(\frac{1}{n}\mathbf{A}^\top \mathbf{A} + \lambda\mathbf{I})^{-\frac{1}{2}}\boldsymbol{\Sigma}_{\mathrm{ref}}(\frac{1}{n}\mathbf{A}^\top \mathbf{A} + \lambda\mathbf{I})^{-\frac{1}{2}}\|_{\mathrm{op}} &= \|\boldsymbol{\Sigma}_{\mathrm{ref}}^{\frac{1}{2}}(\frac{1}{n}\mathbf{A}^\top \mathbf{A} + \lambda\mathbf{I})^{-1}\boldsymbol{\Sigma}_{\mathrm{ref}}^{\frac{1}{2}}\|_{\mathrm{op}} \\
&\lesssim \|\boldsymbol{\Sigma}_{\mathrm{ref}}^{\frac{1}{2}}(\boldsymbol{\Sigma}_{\mathrm{samp}} + \lambda\mathbf{I})^{-1}\boldsymbol{\Sigma}_{\mathrm{ref}}^{\frac{1}{2}}\|_{\mathrm{op}} \\
&= \|\boldsymbol{\Sigma}_{\mathrm{ref}}^{\frac{1}{2}}(\boldsymbol{\Sigma}_{\mathrm{ref}} + \lambda\mathbf{I})^{-1}\boldsymbol{\Sigma}_{\mathrm{ref}}^{\frac{1}{2}}\|_{\mathrm{op}}
\end{aligned}
$$

$$= \|\mathbf{S}_\lambda\|_{\text{op}}.$$

Hence,

$$(\frac{1}{n}\mathbf{A}^\top\mathbf{A} + \lambda\mathbf{I})^{-\frac{1}{2}}\mathbf{\Sigma}_{\text{ref}}(\frac{1}{n}\mathbf{A}^\top\mathbf{A} + \lambda\mathbf{I})^{-\frac{1}{2}} \preceq c\|\mathbf{S}_\lambda\|_{\text{op}}\mathbf{I}. \tag{A.16}$$

for some absolute constant $c > 1$.

Using the established decomposition, we bound each term in turn.

**Bounding Term $\mathscr{B}$.** We first bound the term $\mathscr{B}$. Observe that

$$
\begin{aligned}
\|\mathbf{\Sigma}_{\text{ref}}^{\frac{1}{2}}\mathscr{B}\|_{\mathbb{H}}^2 &\lesssim \boldsymbol{b}^\top\mathbf{A}(\mathbf{A}^\top\mathbf{A} + n\lambda\mathbf{I})^{-1}\mathbf{\Sigma}_{\text{ref}}(\mathbf{A}^\top\mathbf{A} + n\lambda\mathbf{I})^{-1}\mathbf{A}^\top\boldsymbol{b} \\
&\lesssim \frac{1}{n}\|\mathbf{S}_\lambda\|_{\text{op}} \cdot \boldsymbol{b}^\top\mathbf{A}(\mathbf{A}^\top\mathbf{A} + n\lambda\mathbf{I})^{-1}\mathbf{A}^\top\boldsymbol{b} \quad \text{(by Equation (A.16))} \\
&\lesssim \frac{1}{n}\|\mathbf{S}_\lambda\|_{\text{op}}\|\boldsymbol{b}\|_2^2 \\
&\lesssim \frac{\log n}{n}\|\mathbf{S}_\lambda\|_{\text{op}}\|\theta^\star\|_{\mathbb{F}}^2
\end{aligned}
$$

where the last inequality follows from the fact that under the event $\mathscr{E}_{\text{good}}$, we have $|b(a'_j)|^2 \lesssim \|\theta^\star\|_{\mathbb{F}}^2\frac{\log n}{n}$. Since $\gamma \leq 1$, this contribution is absorbed into the nuisance-dependent term in Equation (A.18).

**Bounding Term $\mathscr{C}$.** Observe that

$$
\begin{aligned}
\|\mathbf{\Sigma}_{\text{ref}}^{\frac{1}{2}}\mathscr{C}\|_{\mathbb{H}}^2 &= (n\lambda)^2\|(\mathbf{A}^\top\mathbf{A} + n\lambda\mathbf{I})^{-1}\eta^\star\|_{\mathbf{\Sigma}_{\text{ref}}}^2 \\
&= \lambda^2(\eta^\star)^\top(\frac{1}{n}\mathbf{A}^\top\mathbf{A} + \lambda\mathbf{I})^{-1}\mathbf{\Sigma}_{\text{ref}}(\frac{1}{n}\mathbf{A}^\top\mathbf{A} + \lambda\mathbf{I})^{-1}\eta^\star \\
&\lesssim \lambda^2\|\mathbf{S}_\lambda\|_{\text{op}}(\eta^\star)^\top(\frac{1}{n}\mathbf{A}^\top\mathbf{A} + \lambda\mathbf{I})^{-1}\eta^\star \quad \text{(by Equation (A.16))} \\
&\lesssim \lambda\|\eta^\star\|_{\mathbb{H}}^2\|\mathbf{S}_\lambda\|_{\text{op}}.
\end{aligned}
$$

We next aim to control the propagated error terms. Recall that we defined

$$\mathscr{P} := (\mathbf{A}^\top\mathbf{A} + n\lambda\mathbf{I})^{-1}\mathbf{A}^\top\mathbf{W}(\hat{\theta} - \theta^\star).$$

We decompose it into

$$
\begin{aligned}
\mathscr{P} &= (\mathbf{A}^\top\mathbf{A} + n\lambda\mathbf{I})^{-1}\mathbf{A}^\top\mathbf{W}(\mathbf{Z}^\top\mathbf{Z} + n\lambda_0\mathbf{I})^{-1}\mathbf{Z}^\top\boldsymbol{\varepsilon} - n\lambda_0(\mathbf{A}^\top\mathbf{A} + n\lambda\mathbf{I})^{-1}\mathbf{A}^\top\mathbf{W}(\mathbf{Z}^\top\mathbf{Z} + n\lambda_0\mathbf{I})^{-1}\theta^\star \\
&:= \mathscr{P}_v + \mathscr{P}_b
\end{aligned} \tag{A.17}
$$

for

$$
\begin{aligned}
\mathscr{P}_v &:= (\mathbf{A}^\top\mathbf{A} + n\lambda\mathbf{I})^{-1}\mathbf{A}^\top\mathbf{W}(\mathbf{Z}^\top\mathbf{Z} + n\lambda_0\mathbf{I})^{-1}\mathbf{Z}^\top\boldsymbol{\varepsilon} \\
\mathscr{P}_b &:= -n\lambda_0(\mathbf{A}^\top\mathbf{A} + n\lambda\mathbf{I})^{-1}\mathbf{A}^\top\mathbf{W}(\mathbf{Z}^\top\mathbf{Z} + n\lambda_0\mathbf{I})^{-1}\theta^\star.
\end{aligned}
$$

and we aim to bound each term.

**Bounding Term $\mathscr{P}_v$ (Propagated Variance).** Under the event $\mathscr{E}_{\text{good}}$, with probability at least $1 - n^{-11}$ (over the randomness of $\boldsymbol{\varepsilon}$), by the Hanson-Wright inequality (Wang, 2026), we get

$$
\begin{aligned}
&\|\mathbf{\Sigma}_{\text{ref}}^{\frac{1}{2}}\mathscr{P}_v\|_{\mathbb{H}}^2 \\
&= \|\mathbf{\Sigma}_{\text{ref}}^{\frac{1}{2}}(\mathbf{A}^\top\mathbf{A} + n\lambda\mathbf{I})^{-1}\mathbf{A}^\top\mathbf{W}(\mathbf{Z}^\top\mathbf{Z} + n\lambda_0\mathbf{I})^{-1}\mathbf{Z}^\top\boldsymbol{\varepsilon}\|_{\mathbb{H}}^2 \\
&\lesssim \text{Tr}\left(\mathbf{\Sigma}_{\text{ref}}^{\frac{1}{2}}(\mathbf{A}^\top\mathbf{A} + n\lambda\mathbf{I})^{-1}\mathbf{A}^\top\mathbf{W}(\mathbf{Z}^\top\mathbf{Z} + n\lambda_0\mathbf{I})^{-1}\mathbf{Z}^\top\mathbf{Z}(\mathbf{Z}^\top\mathbf{Z} + n\lambda_0\mathbf{I})^{-1}\mathbf{W}^\top\mathbf{A}(\mathbf{A}^\top\mathbf{A} + n\lambda\mathbf{I})^{-1}\mathbf{\Sigma}_{\text{ref}}^{\frac{1}{2}}\right)\log n
\end{aligned}
$$

$$\lesssim \mathrm{Tr}\left(\boldsymbol{\Sigma}_{\mathrm{ref}}^{\frac{1}{2}}(\mathbf{A}^\top\mathbf{A}+n\lambda\mathbf{I})^{-1}\mathbf{A}^\top\mathbf{W}(\mathbf{Z}^\top\mathbf{Z}+n\lambda_0\mathbf{I})^{-1}\mathbf{W}^\top\mathbf{A}(\mathbf{A}^\top\mathbf{A}+n\lambda\mathbf{I})^{-1}\boldsymbol{\Sigma}_{\mathrm{ref}}^{\frac{1}{2}}\right)\log n$$

$$\overset{(i)}{\lesssim}\frac{1}{\gamma}\mathrm{Tr}\left(\boldsymbol{\Sigma}_{\mathrm{ref}}^{\frac{1}{2}}(\mathbf{A}^\top\mathbf{A}+n\lambda\mathbf{I})^{-1}\mathbf{A}^\top\mathbf{A}(\mathbf{A}^\top\mathbf{A}+n\lambda\mathbf{I})^{-1}\boldsymbol{\Sigma}_{\mathrm{ref}}^{\frac{1}{2}}\right)\log n$$

$$\lesssim\frac{1}{\gamma}\mathrm{Tr}\left(\boldsymbol{\Sigma}_{\mathrm{ref}}(\mathbf{A}^\top\mathbf{A}+n\lambda\mathbf{I})^{-1}\right)\log n$$

$$\overset{(ii)}{\lesssim}\frac{\log n}{n\gamma}\mathrm{Tr}(\mathbf{S}_\lambda).$$

where in step (i), we use the result of Lemma A.7. For step (ii), we use Equation (A.13). We also repeatedly apply Remark A.8 at each step.

**Bounding Term $\mathscr{P}_b$ (Propagated Bias).**   Note that

$$\|\boldsymbol{\Sigma}_{\mathrm{ref}}^{\frac{1}{2}}\mathscr{P}_b\|_{\mathbb{H}}$$
$$\leq n\lambda_0\|\boldsymbol{\Sigma}_{\mathrm{ref}}^{\frac{1}{2}}(\mathbf{A}^\top\mathbf{A}+n\lambda\mathbf{I})^{-1}\mathbf{A}^\top\mathbf{W}(\mathbf{Z}^\top\mathbf{Z}+n\lambda_0\mathbf{I})^{-1}\theta^\star\|_{\mathbb{H}}$$
$$\lesssim n\lambda_0\|\boldsymbol{\Sigma}_{\mathrm{ref}}^{\frac{1}{2}}(\mathbf{A}^\top\mathbf{A}+n\lambda\mathbf{I})^{-\frac{1}{2}}\|_{\mathrm{op}}\cdot\|(\mathbf{A}^\top\mathbf{A}+n\lambda\mathbf{I})^{-\frac{1}{2}}\mathbf{A}^\top\|_{\mathrm{op}}\|\mathbf{W}(\mathbf{Z}^\top\mathbf{Z}+n\lambda_0\mathbf{I})^{-1}\theta^\star\|_2$$
$$\lesssim n\lambda_0\cdot\frac{1}{\sqrt{n}}\|\mathbf{S}_\lambda\|_{\mathrm{op}}^{\frac{1}{2}}\cdot 1\cdot\|\mathbf{W}(\mathbf{Z}^\top\mathbf{Z}+n\lambda_0\mathbf{I})^{-1}\theta^\star\|_2 \quad\text{(by Equation (A.16))}$$
$$\lesssim n\lambda_0\cdot\frac{1}{\sqrt{n}}\|\mathbf{S}_\lambda\|_{\mathrm{op}}^{\frac{1}{2}}\cdot\|\mathbf{W}(\mathbf{Z}^\top\mathbf{Z}+n\lambda_0\mathbf{I})^{-1/2}\|_{\mathrm{op}}\|(\mathbf{Z}^\top\mathbf{Z}+n\lambda_0\mathbf{I})^{-1/2}\theta^\star\|_{\mathbb{F}}$$
$$\lesssim\sqrt{n\lambda_0}\cdot\frac{1}{\sqrt{n}}\|\mathbf{S}_\lambda\|_{\mathrm{op}}^{\frac{1}{2}}\cdot\|\mathbf{W}(\mathbf{Z}^\top\mathbf{Z}+n\lambda_0\mathbf{I})^{-1/2}\|_{\mathrm{op}}\|\theta^\star\|_{\mathbb{F}}$$

By Lemma A.7 and Remark A.8, since $\lambda_0\asymp\frac{\log n}{n}$, we have

$$\|(\mathbf{Z}^\top\mathbf{Z}+n\lambda_0\mathbf{I})^{-1/2}\mathbf{W}^\top\mathbf{W}(\mathbf{Z}^\top\mathbf{Z}+n\lambda_0\mathbf{I})^{-1/2}\|_{\mathrm{op}}\lesssim\frac{1}{\gamma}.$$

Thus, we obtain

$$\|\boldsymbol{\Sigma}_{\mathrm{ref}}^{\frac{1}{2}}\mathscr{P}_b\|_{\mathbb{H}}^2\lesssim n\lambda_0\cdot\frac{1}{n}\|\mathbf{S}_\lambda\|_{\mathrm{op}}\cdot\frac{1}{\gamma}\|\theta^\star\|_{\mathbb{F}}^2$$
$$\lesssim\frac{\log n}{n\gamma}\|\mathbf{S}_\lambda\|_{\mathrm{op}}\|\theta^\star\|_{\mathbb{F}}^2 \quad(\text{since }\lambda_0\asymp\frac{\log n}{n}).$$

Collecting the bounds above, we obtain

$$\mathcal{E}_{\mathbb{H}}(\hat{\eta}_\lambda)\lesssim\lambda\|\mathbf{S}_\lambda\|_{\mathrm{op}}\|\eta^\star\|_{\mathbb{H}}^2+\frac{\log n}{n\gamma}\mathrm{Tr}(\mathbf{S}_\lambda)+\frac{\log n}{n\gamma}\|\mathbf{S}_\lambda\|_{\mathrm{op}}\|\theta^\star\|_{\mathbb{F}}^2$$
$$\lesssim\lambda\|\mathbf{S}_\lambda\|_{\mathrm{op}}\|h^\star\|_{\mathcal{H}}^2+\frac{\log n}{n\gamma}\mathrm{Tr}(\mathbf{S}_\lambda)+\frac{\log n}{n\gamma}\|\mathbf{S}_\lambda\|_{\mathrm{op}}\|f^\star\|_{\mathcal{F}}^2. \tag{A.18}$$

Note that

$$\|\mathbf{S}_\lambda\|_{\mathrm{op}}\leq 1$$

and

$$\mathrm{Tr}(\mathbf{S}_\lambda)=\Gamma(\lambda).$$

Next, under polynomial eigendecay, we obtain the following (Wang, 2026):

$$\Gamma(\lambda)\lesssim\lambda^{-\frac{1}{2\ell}}.$$

For exponential decay, we obtain

$$\Gamma(\lambda) \lesssim \log(1/\lambda).$$

Finally, under a finite spectrum, we have

$$\Gamma(\lambda) \lesssim D.$$

Combining this with the result of Equation (A.18) leads to the desired result.

## B. Proof of Theorem 4.5 (Model Selection)

### B.1. Two Key Propositions for Model Selection

Recall that when we run Algorithm 2, we set $\tilde{\lambda} \asymp \frac{\log n}{n}$. Under weak overlap, we define the term $\mathscr{O}$ as follows:

$$\mathscr{O} := \frac{\xi \log n}{n\gamma} \left(1 + \|\theta^\star\|_{\mathbb{F}}^2\right).$$

This term, $\mathscr{O}$, will serve as the price of model selection in our analysis.

For any estimator $\hat{h}$ and its corresponding Hilbertian element $\hat{\eta}$, we define the **in-sample MISE** as the empirical MSE over $\{\tilde{a}_j\}_{j=1}^n$, namely

$$\mathcal{E}_{\mathbb{H}}^{\text{in}}(\hat{\eta}) := \frac{1}{n} \sum_{j=1}^n |\phi(\tilde{a}_j)^\top (\hat{\eta} - \eta^\star)|^2.$$

We present two oracle inequalities. The first is an oracle inequality for the in-sample MISE defined above.

**Proposition B.1** (Oracle inequality for in-sample MISE). *With probability at least $1 - 2n^{-11}$, we have*

$$\mathcal{E}_{\mathbb{H}}^{\text{in}}(\hat{\eta}_{\hat{\lambda}}) \lesssim \min_{\lambda \in \Lambda} \mathcal{E}_{\mathbb{H}}^{\text{in}}(\hat{\eta}_\lambda) + \mathscr{O}.$$

This proposition provides a performance guarantee for our model selection procedure with respect to the in-sample MISE. Next, we provide a key inequality for connecting the in-sample MISE and the population MISE.

**Proposition B.2** (Connecting in-sample MISE and population MISE). *With probability at least $1 - 2n^{-11}$, the following holds uniformly for all $\lambda \in \Lambda$:*

$$\mathcal{E}_{\mathbb{H}}^{\text{in}}(\hat{\eta}_\lambda) \lesssim \mathcal{E}_{\mathbb{H}}(\hat{\eta}_\lambda) + \mathscr{O}$$
$$\mathcal{E}_{\mathbb{H}}(\hat{\eta}_\lambda) \lesssim \mathcal{E}_{\mathbb{H}}^{\text{in}}(\hat{\eta}_\lambda) + \mathscr{O}.$$

Theorem 4.5 follows by combining Proposition B.1 and Proposition B.2. We aim to apply Theorem 4.3. By our design of $\Lambda$, there exists $\lambda^\star \in \Lambda$ that achieves the nearly optimal result of Theorem 4.3. The additional model-selection price $\mathscr{O}$ contributes only

$$\widetilde{\mathcal{O}}\left(\frac{1 + \|\theta^\star\|_{\mathbb{F}}^2}{n\gamma}\right),$$

and is therefore absorbed into the rates in Theorem 4.5. Then, we obtain the desired result directly. Thus, the remaining sections aim to prove the two propositions.

### B.2. Notation and Hilbertian Formulation for Algorithm 2

We introduce notation for the proof in Table 4.

*Table 4.* Detailed Summary of Notations and Operators

| Notation | Definition / Description |
|---|---|
| $\mathcal{D}_{\text{train}}$ | $\{x_{1i}, a_{1i}, y_{1i}\}_{i=1}^{n_1}$ |
| $\mathcal{D}_{\text{val}}$ | $\{x_{2i}, a_{2i}, y_{2i}\}_{i=1}^{n_2}$ |
| $\mathbf{Z}_1$ | Design operator of $\{\psi(x_{1i}, a_{1i})\}_{i=1}^{n_1}$ in $\mathcal{D}_{\text{train}}$ |
| $\mathbf{Z}_2$ | Design operator of $\{\psi(x_{2i}, a_{2i})\}_{i=1}^{n_2}$ in $\mathcal{D}_{\text{val}}$ |
| $\{a'_{1j}\}_{j=1}^{n_1}$ | Sampled treatments from $\mathcal{P}_{\text{samp}}$ for estimating $\hat{h}_\lambda$ |
| $\{a'_{2j}\}_{j=1}^{n_2}$ | Sampled treatments from $\mathcal{P}_{\text{samp}}$ for estimating $\tilde{h}$ |
| $\mathbf{A}_1$ | Design operator of $\{\phi(a'_{1j})\}_{j=1}^{n_1}$ |
| $\mathbf{A}_2$ | Design operator of $\{\phi(a'_{2j})\}_{j=1}^{n_2}$ |
| $\mathbf{Y}_1$ | Vector version of $\{y_{1i}\}_{i=1}^{n_1}$ in $\mathcal{D}_{\text{train}}$ |
| $\mathbf{Y}_2$ | Vector version of $\{y_{2i}\}_{i=1}^{n_2}$ in $\mathcal{D}_{\text{val}}$ |
| $\bar{\psi}(a)$ | Feature mean defined as $\mathbb{E}_{x \sim \mathcal{P}_X}[\psi(x, a)]$ |
| $\hat{\bar{\psi}}_1(a)$ | Defined as $\frac{1}{n_1} \sum_{i=1}^{n_1} \psi(x_{1i}, a)$ |
| $\mathbf{W}_1$ | Design operator of $\{\hat{\bar{\psi}}_1(a'_{1j})\}_{j=1}^{n_1}$ in $\mathcal{D}_{\text{train}}$ |
| $\hat{\bar{\psi}}_2(a)$ | Defined as $\frac{1}{n_2} \sum_{i=1}^{n_2} \psi(x_{2i}, a)$ |
| $\mathbf{W}_2$ | Design operator of $\{\hat{\bar{\psi}}_2(a'_{2j})\}_{j=1}^{n_2}$ in $\mathcal{D}_{\text{val}}$ |
| $\mathbf{U}$ | Design operator of $\{\phi(\tilde{a}_j)\}_{j=1}^{n}$ |
| $\hat{\theta}$ | Hilbertian element corresponding to $\hat{f}$ (trained on $\mathcal{D}_{\text{train}}$) |
| $\hat{\eta}_\lambda$ | Hilbertian element corresponding to $\hat{h}_\lambda$ (trained on $\mathcal{D}_{\text{train}}$) |
| $\tilde{\eta}$ | Hilbertian element corresponding to $\tilde{h}$ (trained on $\mathcal{D}_{\text{val}}$) |
| $\mathbf{S}_\lambda$ | Regularized second-moment operator $\mathbf{\Sigma}_{\text{ref}}^{1/2}(\mathbf{\Sigma}_{\text{ref}} + \lambda \mathbf{I})^{-1}\mathbf{\Sigma}_{\text{ref}}^{1/2}$ |

**Hilbertian Formulation for $\tilde{h}$.** We also rewrite the regression used to obtain $\tilde{h}$ in terms of Hilbertian elements. To obtain $\tilde{h}$, we run Algorithm 1 with main regularizer $\tilde{\lambda} \asymp \log n/n$, using the dataset $\mathcal{D}_{\text{val}}$. Through the first nuisance estimation, we obtain $\tilde{f}$, and denote its Hilbertian element as $\tilde{\theta}$. Analogous to the analysis in Appendix A.1, $\tilde{\theta}$ admits the explicit representation

$$\tilde{\theta} := (\mathbf{Z}_2^\top \mathbf{Z}_2 + n_2 \lambda_0 \mathbf{I})^{-1} \mathbf{Z}_2^\top \mathbf{Y}_2.$$

Using this, the Hilbertian element $\tilde{\eta}$ of $\tilde{h}$ is obtained as follows:

$$\tilde{\eta} := (\mathbf{A}_2^\top \mathbf{A}_2 + n_2 \tilde{\lambda} \mathbf{I})^{-1} \mathbf{A}_2^\top \mathbf{W}_2 \tilde{\theta}.$$

**Hilbertian Formulation for $\hat{h}_\lambda$.** Analogous to the analysis in Appendix A.1, and allowing for a slight abuse of notation, we obtain the expression for $\hat{\eta}_\lambda$, the Hilbertian element corresponding to $\hat{h}_\lambda$:

$$\hat{\eta}_\lambda = (\mathbf{A}_1^\top \mathbf{A}_1 + n_1 \lambda \mathbf{I})^{-1} \mathbf{A}_1^\top \mathbf{W}_1 \hat{\theta},$$

where $\hat{\theta}$ (corresponding Hilbertian element of $\hat{f}$) is given by

$$\hat{\theta} = (\mathbf{Z}_1^\top \mathbf{Z}_1 + n_1 \lambda_0 \mathbf{I})^{-1} \mathbf{Z}_1^\top \mathbf{Y}_1.$$

The definitions of the second moments $\mathbf{\Sigma}_{\text{ref}}$ and $\mathbf{\Sigma}_{\text{samp}}$ are the same as in Appendix A.1.

## B.3. Good Events for Model Selection

We previously defined a good event for the dataset $\mathcal{D}$ in Appendix A.2. By applying the same argument, we define a good event for the model selection proof. Recall that $n_1 = n_2 = \frac{n}{2}$. By the same logic as in Appendix A.2, the following bounds hold, each with probability at least $1 - n^{-11}$, for all $\{a'_{1j}\}_{j=1}^{n_1}$ and $\{a'_{2j}\}_{j=1}^{n_2}$:

$$\left| \frac{1}{n_1} \sum_{i=1}^{n_1} f^\star(x_{1i}, a'_{1j}) - \mathbb{E}_{x \sim \mathcal{P}_X}[f^\star(x, a'_{1j})] \right| \lesssim \frac{\sqrt{\xi} \|\theta^\star\|_{\mathbb{F}} \sqrt{\log n}}{\sqrt{n}}$$

$$\left| \frac{1}{n_2} \sum_{i=1}^{n_2} f^\star(x_{2i}, a'_{2j}) - \mathbb{E}_{x \sim \mathcal{P}_X}[f^\star(x, a'_{2j})] \right| \lesssim \frac{\sqrt{\xi} \|\theta^\star\|_{\mathbb{F}} \sqrt{\log n}}{\sqrt{n}}$$

$$(\text{B.1})$$

Again, by the same logic and Lemma F.2, applied separately to the sample sizes $n_1$ and $n_2$, we first obtain shifts of the form $+ \log(n_k/\delta)\mathbf{I}$ (for $k \in \{1, 2\}$). Taking $\delta = n^{-11}$, we have

$$\log(n_k/\delta) = \log((n/2)n^{11}) = 12 \log n - \log 2 \asymp \log n \asymp \log n_k.$$

Hence, after absorbing absolute constants (equivalently via Remark A.8), we may write the shifts as $+ \log n_k \mathbf{I}$. Therefore, the following inequalities hold for some absolute constant $c > 1$, each with probability at least $1 - n^{-11}$:

$$\frac{1}{c}(\mathbf{A}_1^\top \mathbf{A}_1 + \log n_1 \mathbf{I}) \preceq n_1 \mathbf{\Sigma}_{\text{samp}} + \log n_1 \mathbf{I} \preceq c(\mathbf{A}_1^\top \mathbf{A}_1 + \log n_1 \mathbf{I})$$

$$\frac{1}{c}(\mathbf{A}_2^\top \mathbf{A}_2 + \log n_2 \mathbf{I}) \preceq n_2 \mathbf{\Sigma}_{\text{samp}} + \log n_2 \mathbf{I} \preceq c(\mathbf{A}_2^\top \mathbf{A}_2 + \log n_2 \mathbf{I})$$

$$\frac{1}{c}(\mathbf{Z}_1^\top \mathbf{Z}_1 + \log n_1 \mathbf{I}) \preceq n_1 \mathbb{E}_{(x,a) \sim \mathcal{P}_{X,A}}[\psi(x,a)\psi(x,a)^\top] + \log n_1 \mathbf{I} \preceq c(\mathbf{Z}_1^\top \mathbf{Z}_1 + \log n_1 \mathbf{I})$$

$$\frac{1}{c}(\mathbf{Z}_2^\top \mathbf{Z}_2 + \log n_2 \mathbf{I}) \preceq n_2 \mathbb{E}_{(x,a) \sim \mathcal{P}_{X,A}}[\psi(x,a)\psi(x,a)^\top] + \log n_2 \mathbf{I} \preceq c(\mathbf{Z}_2^\top \mathbf{Z}_2 + \log n_2 \mathbf{I}).$$

$$(\text{B.2})$$

In Appendix A, we defined $\mathbf{Q}(a)$ as

$$\mathbf{Q}(a) = \mathbb{E}_{x \sim \mathcal{P}_X}[\psi(x,a)\psi(x,a)^\top].$$

We also define $\bar{\mathbf{Q}}_1$ and $\bar{\mathbf{Q}}_2$ as follows:

$$\bar{\mathbf{Q}}_1 := \frac{1}{n_1} \sum_{j=1}^{n_1} \mathbf{Q}(a'_{1j}), \quad \bar{\mathbf{Q}}_2 := \frac{1}{n_2} \sum_{j=1}^{n_2} \mathbf{Q}(a'_{2j})$$

Then, by the same argument as in Appendix A.1, for some constant $c > 1$, the following holds with probability at least $1 - n^{-11}$:

$$\bar{\mathbf{Q}}_1 \preceq c(\mathbf{Q} + \frac{\log n_1}{n_1}\mathbf{I}), \quad \bar{\mathbf{Q}}_2 \preceq c(\mathbf{Q} + \frac{\log n_2}{n_2}\mathbf{I}).$$

On the event where the two inequalities above hold, applying the same argument as in Lemma A.7 yields

$$\frac{1}{n_1}\mathbf{W}_1^\top \mathbf{W}_1 \preceq \frac{c}{\gamma} \times \frac{1}{n_1}(\mathbf{Z}_1^\top \mathbf{Z}_1 + \log n_1 \mathbf{I})$$

$$\frac{1}{n_2}\mathbf{W}_2^\top \mathbf{W}_2 \preceq \frac{c}{\gamma} \times \frac{1}{n_2}(\mathbf{Z}_2^\top \mathbf{Z}_2 + \log n_2 \mathbf{I})$$

$$(\text{B.3})$$

Finally, we present concentration bounds for $\mathbf{U}^\top \mathbf{U}$. Since $\mathbf{\Sigma}_{\mathrm{ref}} = \mathbb{E}_{a \sim \mathcal{P}_{\mathrm{ref}}}[\phi(a)\phi(a)^\top]$, by applying Lemma F.2, with probability at least $1 - n^{-11}$, we have

$$\frac{1}{c}(\mathbf{U}^\top \mathbf{U} + \log n\mathbf{I}) \preceq n\mathbf{\Sigma}_{\mathrm{ref}} + \log n\mathbf{I} \preceq c(\mathbf{U}^\top \mathbf{U} + \log n\mathbf{I}). \tag{B.4}$$

Here, the shift remains $+\log n\mathbf{I}$ because $\mathbf{U}$ is built from $n$ pseudo-test samples $\{\tilde{a}_j\}_{j=1}^n$. In subsequent bounds, whenever needed, we use the scale relations above and Remark A.8 to replace the $+\log n_1\mathbf{I}$ and $+\log n_2\mathbf{I}$ shifts by $+n_1\lambda\mathbf{I}$, $+n_1\lambda_0\mathbf{I}$, $+n_2\lambda_0\mathbf{I}$, or $+n_2\tilde{\lambda}\mathbf{I}$. Since $n_1 = n_2 = n/2$, we also have $\log n_1 \asymp \log n_2 \asymp \log n$, so these shifts are interchangeable with $+\log n\mathbf{I}$ up to absolute constants.

**Definition B.3** (Good event). We define the good event $\mathscr{F}_{\mathrm{good}}$ as the event where Equations (B.1) to (B.4) hold.

Thus, the following lemma follows directly.

**Lemma B.4.** *We have*

$$\mathbb{P}[\mathscr{F}_{\mathrm{good}}] \geq 1 - \frac{1}{n^{10}}.$$

*Proof.* The proof is straightforward from the previous observations by taking a union bound. $\square$

### B.4. Proof of Proposition B.1

Our goal is to apply Lemma B.7. We analyze the quality of the pseudo-test outcomes defined as

$$\tilde{\mathbf{m}} := (\tilde{h}(\tilde{a}_j))_{j=1}^n = \mathbf{U}\tilde{\eta} = \mathbf{U}(\mathbf{A}_2^\top \mathbf{A}_2 + n_2\tilde{\lambda}\mathbf{I})^{-1}\mathbf{A}_2^\top \mathbf{W}_2\tilde{\theta}.$$

Recall that we set

$$\tilde{\theta} := (\mathbf{Z}_2^\top \mathbf{Z}_2 + n_2\lambda_0\mathbf{I})^{-1}\mathbf{Z}_2^\top \mathbf{Y}_2.$$

We also define

$$\mathbf{m}^\star := (h^\star(\tilde{a}_j))_{j=1}^n = \mathbf{U}\eta^\star.$$

By the same argument as in Appendix A.3, the estimation error of $\tilde{\eta}$ admits the decomposition

$$\tilde{\eta} - \eta^\star = (\mathbf{A}_2^\top \mathbf{A}_2 + n_2\tilde{\lambda}\mathbf{I})^{-1}\mathbf{A}_2^\top \boldsymbol{b}_2 + (\mathbf{A}_2^\top \mathbf{A}_2 + n_2\tilde{\lambda}\mathbf{I})^{-1}\mathbf{A}_2^\top \mathbf{W}_2(\tilde{\theta} - \theta^\star) - n_2\tilde{\lambda} \cdot (\mathbf{A}_2^\top \mathbf{A}_2 + n_2\tilde{\lambda}\mathbf{I})^{-1}\eta^\star$$
$$:= \tilde{\mathscr{B}} + \tilde{\mathscr{P}} + \mathscr{C}$$

and this leads to

$$\tilde{\mathbf{m}} - \mathbf{m}^\star = \mathbf{U}(\tilde{\eta} - \eta^\star) = \mathbf{U}\tilde{\mathscr{B}} + \mathbf{U}\tilde{\mathscr{P}} + \mathbf{U}\mathscr{C}. \tag{B.5}$$

**Lemma B.5.** *Under the event $\mathscr{F}_{\mathrm{good}}$,*

$$\|\tilde{\mathbf{m}} - \mathbb{E}_{\boldsymbol{\varepsilon}_2}[\tilde{\mathbf{m}}]\|_{\psi_2}^2 \lesssim \frac{1}{\gamma}\sigma^2.$$

*Here and throughout this proof, $\mathbb{E}_{\boldsymbol{\varepsilon}_2}$ denotes conditional expectation with respect to the validation noise vector $\boldsymbol{\varepsilon}_2$, holding all design variables and query points fixed.*

*Proof.* We have

$$\|\tilde{\mathbf{m}} - \mathbb{E}_{\boldsymbol{\varepsilon}_2}[\tilde{\mathbf{m}}]\|_{\psi_2} \lesssim \|\mathbf{U}(\mathbf{A}_2^\top \mathbf{A}_2 + n_2\tilde{\lambda}\mathbf{I})^{-1}\mathbf{A}_2^\top \mathbf{W}_2(\tilde{\theta} - \mathbb{E}_{\boldsymbol{\varepsilon}_2}[\tilde{\theta}])\|_{\psi_2}$$
$$\lesssim \|\mathbf{U}(\mathbf{A}_2^\top \mathbf{A}_2 + n_2\tilde{\lambda}\mathbf{I})^{-1}\mathbf{A}_2^\top \mathbf{W}_2(\mathbf{Z}_2^\top \mathbf{Z}_2 + n_2\lambda_0\mathbf{I})^{-1}\mathbf{Z}_2^\top \boldsymbol{\varepsilon}_2\|_{\psi_2}$$
$$\lesssim \sigma\|\mathbf{U}(\mathbf{A}_2^\top \mathbf{A}_2 + n_2\tilde{\lambda}\mathbf{I})^{-1}\mathbf{A}_2^\top \mathbf{W}_2(\mathbf{Z}_2^\top \mathbf{Z}_2 + n_2\lambda_0\mathbf{I})^{-1}\mathbf{Z}_2^\top\|_{\mathrm{op}}$$
$$\lesssim \sigma\|\mathbf{U}(\mathbf{A}_2^\top \mathbf{A}_2 + n_2\tilde{\lambda}\mathbf{I})^{-1}\mathbf{A}_2^\top \mathbf{W}_2(\mathbf{Z}_2^\top \mathbf{Z}_2 + n_2\lambda_0\mathbf{I})^{-\frac{1}{2}}\|_{\mathrm{op}}$$

$$\lesssim \sigma \|\mathbf{U}(\mathbf{A}_2^\top \mathbf{A}_2 + n_2\tilde{\lambda}\mathbf{I})^{-1}\mathbf{A}_2^\top\|_{\mathrm{op}} \cdot \|\mathbf{W}_2(\mathbf{Z}_2^\top \mathbf{Z}_2 + n_2\lambda_0\mathbf{I})^{-\frac{1}{2}}\|_{\mathrm{op}}$$

$$\overset{(i)}{\lesssim} \sigma\sqrt{\frac{1}{\gamma}}\|\mathbf{U}(\mathbf{A}_2^\top \mathbf{A}_2 + n_2\tilde{\lambda}\mathbf{I})^{-1}\mathbf{A}_2^\top\|_{\mathrm{op}}$$

$$\lesssim \sigma\sqrt{\frac{1}{\gamma}}\|\mathbf{U}(\mathbf{A}_2^\top \mathbf{A}_2 + n_2\tilde{\lambda}\mathbf{I})^{-\frac{1}{2}}\|_{\mathrm{op}}$$

$$\overset{(ii)}{\lesssim} \sigma\sqrt{\frac{1}{\gamma}}$$

where step $(i)$ holds by Equation (B.3) and step $(ii)$ holds by the relation

$$\frac{1}{c}(n_2\mathbf{\Sigma}_{\mathrm{samp}} + n_2\tilde{\lambda}\mathbf{I}) \preceq \mathbf{A}_2^\top \mathbf{A}_2 + n_2\tilde{\lambda}\mathbf{I}$$

and

$$\frac{1}{n}\mathbf{U}^\top\mathbf{U} \preceq c(\mathbf{\Sigma}_{\mathrm{ref}} + \tilde{\lambda}\mathbf{I})$$

for some absolute constant $c > 1$ under the good event $\mathscr{F}_{\mathrm{good}}$. We also repeatedly apply Remark A.8 at each step. □

**Lemma B.6.** *Under the good event $\mathscr{F}_{\mathrm{good}}$, we have*

$$\|\mathbb{E}_{\boldsymbol{\varepsilon}_2}[\tilde{\mathbf{m}}] - \mathbf{m}^\star\|_2^2 \lesssim (\frac{1}{\gamma}\|\theta^\star\|_{\mathbb{F}}^2 + \|\eta^\star\|_{\mathbb{H}}^2)\log n.$$

*Proof.* Recall that $\tilde{\mathbf{m}} = \mathbf{U}\tilde{\eta}$ and we defined

$$\tilde{\mathscr{P}} := (\mathbf{A}_2^\top \mathbf{A}_2 + n_2\tilde{\lambda}\mathbf{I})^{-1}\mathbf{A}_2^\top \mathbf{W}_2(\tilde{\theta} - \theta^\star)$$

in the previous decomposition Equation (B.5). We decompose it again as

$$\tilde{\mathscr{P}} = (\mathbf{A}_2^\top \mathbf{A}_2 + n_2\tilde{\lambda}\mathbf{I})^{-1}\mathbf{A}_2^\top \mathbf{W}_2(\mathbf{Z}_2^\top \mathbf{Z}_2 + n_2\lambda_0\mathbf{I})^{-1}\mathbf{Z}_2^\top \boldsymbol{\varepsilon}_2$$
$$+ (\mathbf{A}_2^\top \mathbf{A}_2 + n_2\tilde{\lambda}\mathbf{I})^{-1}\mathbf{A}_2^\top \mathbf{W}_2(-n_2\lambda_0)(\mathbf{Z}_2^\top \mathbf{Z}_2 + n_2\lambda_0\mathbf{I})^{-1}\theta^\star$$
$$:= \tilde{\mathscr{P}}_v + \tilde{\mathscr{P}}_b.$$

where

$$\tilde{\mathscr{P}}_v := (\mathbf{A}_2^\top \mathbf{A}_2 + n_2\tilde{\lambda}\mathbf{I})^{-1}\mathbf{A}_2^\top \mathbf{W}_2(\mathbf{Z}_2^\top \mathbf{Z}_2 + n_2\lambda_0\mathbf{I})^{-1}\mathbf{Z}_2^\top \boldsymbol{\varepsilon}_2$$
$$\tilde{\mathscr{P}}_b := -n_2\lambda_0(\mathbf{A}_2^\top \mathbf{A}_2 + n_2\tilde{\lambda}\mathbf{I})^{-1}\mathbf{A}_2^\top \mathbf{W}_2(\mathbf{Z}_2^\top \mathbf{Z}_2 + n_2\lambda_0\mathbf{I})^{-1}\theta^\star.$$

Using the previous decomposition Equation (B.5), we obtain

$$\|\mathbb{E}_{\boldsymbol{\varepsilon}_2}[\tilde{\mathbf{m}}] - \mathbf{m}^\star\|_2 \leq \|\mathbf{U}\tilde{\mathscr{C}}\|_2 + \|\mathbf{U}\tilde{\mathscr{P}}_b\|_2 + \|\mathbf{U}\tilde{\mathscr{B}}\|_2.$$

First, we have

$$\|\mathbf{U}\tilde{\mathscr{B}}\|_2^2 \lesssim \boldsymbol{b}_2^\top \mathbf{A}_2(\mathbf{A}_2^\top \mathbf{A}_2 + n_2\tilde{\lambda}\mathbf{I})^{-1}\mathbf{U}^\top\mathbf{U}(\mathbf{A}_2^\top \mathbf{A}_2 + n_2\tilde{\lambda}\mathbf{I})^{-1}\mathbf{A}_2^\top \boldsymbol{b}_2$$

$$\overset{(i)}{\lesssim} \boldsymbol{b}_2^\top \mathbf{A}_2(\mathbf{A}_2^\top \mathbf{A}_2 + n_2\tilde{\lambda}\mathbf{I})^{-1}(n\mathbf{\Sigma}_{\mathrm{ref}} + \log n\mathbf{I})(\mathbf{A}_2^\top \mathbf{A}_2 + n_2\tilde{\lambda}\mathbf{I})^{-1}\mathbf{A}_2^\top \boldsymbol{b}_2$$

$$\overset{(ii)}{\lesssim} \boldsymbol{b}_2^\top \mathbf{A}_2(\mathbf{A}_2^\top \mathbf{A}_2 + n_2\tilde{\lambda}\mathbf{I})^{-1}\mathbf{A}_2^\top \boldsymbol{b}_2 \quad \text{(By Equation (B.2))}$$

$$\lesssim \|\boldsymbol{b}_2\|_2^2$$

$$\overset{(iii)}{\lesssim} \|\theta^\star\|_{\mathbb{F}}^2 \log n.$$

where step (i) holds by the definition of the good event $\mathscr{F}_{\mathrm{good}}$, especially Equation (B.4) and step (ii) holds by Equation (B.2). Also, step (iii) holds by Equation (B.1). We also repeatedly apply Remark A.8 at each step.

Second, we bound

$$\|\mathbf{U}\check{\mathscr{C}}\|_2^2 \lesssim (n_2\tilde{\lambda})^2 (\eta^\star)^\top (\mathbf{A}_2^\top \mathbf{A}_2 + n_2\tilde{\lambda}\mathbf{I})^{-1} \mathbf{U}^\top \mathbf{U} (\mathbf{A}_2^\top \mathbf{A}_2 + n_2\tilde{\lambda}\mathbf{I})^{-1} \eta^\star$$

$$\overset{(i)}{\lesssim} (n_2\tilde{\lambda})^2 (\eta^\star)^\top (\mathbf{A}_2^\top \mathbf{A}_2 + n_2\tilde{\lambda}\mathbf{I})^{-1} (n\mathbf{\Sigma}_{\mathrm{ref}} + \log n\mathbf{I}) (\mathbf{A}_2^\top \mathbf{A}_2 + n_2\tilde{\lambda}\mathbf{I})^{-1} \eta^\star$$

$$\overset{(ii)}{\lesssim} (n_2\tilde{\lambda})^2 (\eta^\star)^\top (\mathbf{A}_2^\top \mathbf{A}_2 + n_2\tilde{\lambda}\mathbf{I})^{-1} \eta^\star$$

$$\lesssim (n_2\tilde{\lambda})\|\eta^\star\|_{\mathbb{H}}^2$$

$$\lesssim \log n\|\eta^\star\|_{\mathbb{H}}^2.$$

where step (i) holds by Equation (B.4) and step (ii) holds by Equation (B.2).

Finally, we bound

$$\|\mathbf{U}\tilde{\mathscr{P}}_b\|_2^2$$

$$\leq (n_2\lambda_0)^2 \|\mathbf{U}(\mathbf{A}_2^\top \mathbf{A}_2 + n_2\tilde{\lambda}\mathbf{I})^{-1} \mathbf{A}_2^\top \mathbf{W}_2 (\mathbf{Z}_2^\top \mathbf{Z}_2 + n_2\lambda_0\mathbf{I})^{-1}\theta^\star\|_2^2$$

$$\lesssim (n_2\lambda_0)^2 \|\mathbf{U}(\mathbf{A}_2^\top \mathbf{A}_2 + n_2\tilde{\lambda}\mathbf{I})^{-\frac{1}{2}}\|_{\mathrm{op}}^2 \cdot \|(\mathbf{A}_2^\top \mathbf{A}_2 + n_2\tilde{\lambda}\mathbf{I})^{-\frac{1}{2}} \mathbf{A}_2^\top\|_{\mathrm{op}}^2 \|\mathbf{W}_2 (\mathbf{Z}_2^\top \mathbf{Z}_2 + n_2\lambda_0\mathbf{I})^{-1}\theta^\star\|_2^2$$

$$\lesssim (n_2\lambda_0)^2 \|\mathbf{U}(\mathbf{A}_2^\top \mathbf{A}_2 + n_2\tilde{\lambda}\mathbf{I})^{-\frac{1}{2}}\|_{\mathrm{op}}^2 \cdot \|\mathbf{W}_2 (\mathbf{Z}_2^\top \mathbf{Z}_2 + n_2\lambda_0\mathbf{I})^{-1}\theta^\star\|_2^2$$

$$\lesssim (n_2\lambda_0)^2 \|\mathbf{U}(\mathbf{A}_2^\top \mathbf{A}_2 + n_2\tilde{\lambda}\mathbf{I})^{-\frac{1}{2}}\|_{\mathrm{op}}^2 \cdot \|\mathbf{W}_2 (\mathbf{Z}_2^\top \mathbf{Z}_2 + n_2\lambda_0\mathbf{I})^{-\frac{1}{2}}\|_{\mathrm{op}}^2 \cdot \frac{1}{n_2\lambda_0}\|\theta^\star\|_{\mathbb{F}}^2$$

$$\overset{(i)}{\lesssim} n_2\lambda_0 \|\mathbf{U}(\mathbf{A}_2^\top \mathbf{A}_2 + n_2\tilde{\lambda}\mathbf{I})^{-\frac{1}{2}}\|_{\mathrm{op}}^2 \cdot \frac{1}{\gamma} \cdot \|\theta^\star\|_{\mathbb{F}}^2$$

$$\overset{(ii)}{\lesssim} \frac{1}{\gamma} n_2\lambda_0 \|\theta^\star\|_{\mathbb{F}}^2$$

$$\lesssim \frac{1}{\gamma} \log n\|\theta^\star\|_{\mathbb{F}}^2$$

where step (i) holds by Equation (B.3) and step (ii) holds by Equation (B.4). □

**Final Step of the Proof.** We now apply Lemma B.7. Condition on $\mathcal{D}_{\mathrm{train}}$, the training query points $\{a'_{1j}\}_{j=1}^{n_1}$, the test points $\{\tilde{a}_j\}_{j=1}^{n}$, the validation design variables $\{(x_{2i}, a_{2i})\}_{i=1}^{n_2}$, the validation query points $\{a'_{2j}\}_{j=1}^{n_2}$, and the event $\mathscr{F}_{\mathrm{good}}$. Under this conditioning, $\{\hat{h}_\lambda\}_{\lambda \in \Lambda}$ are deterministic candidates, while the linear maps used to construct $\tilde{h}$ are fixed; hence $\tilde{h}$ is random only through the validation noise vector $\varepsilon_2$. We have

$$\mathcal{E}_{\mathbb{H}}^{\mathrm{in}}(\hat{\eta}_{\hat{\lambda}}) \lesssim \min_{\lambda \in \Lambda} \mathcal{E}_{\mathbb{H}}^{\mathrm{in}}(\hat{\eta}_\lambda) + \frac{1}{n}\|\mathbf{U}(\mathbb{E}_{\varepsilon_2}[\tilde{\eta}] - \eta^\star)\|_2^2 + \frac{\log(Ln)}{n}\|\mathbf{U}(\tilde{\eta} - \mathbb{E}_{\varepsilon_2}[\tilde{\eta}])\|_{\psi_2}^2$$

$$\lesssim \min_{\lambda \in \Lambda} \mathcal{E}_{\mathbb{H}}^{\mathrm{in}}(\hat{\eta}_\lambda) + \mathscr{O}$$

holds with conditional probability at least $1 - n^{-11}$. Since $\mathbb{P}[\mathscr{F}_{\mathrm{good}}] \geq 1 - n^{-11}$, a union bound gives unconditional probability at least $1 - 2n^{-11}$, completing the proof.

**Lemma B.7** (Theorem 5.2 from Wang 2026). *Let $\{z_i\}_{i=1}^{n}$ be deterministic elements in a set $\mathcal{Z}$; $g^\star$ and $\{g_j\}_{j=1}^{m}$ be deterministic functions on $\mathcal{Z}$; $\tilde{g}$ be a random function on $\mathcal{Z}$. Define*

$$\mathcal{L}(g) = \frac{1}{n}\sum_{i=1}^{n} |g(z_i) - g^\star(z_i)|^2$$

*for any function $g$ on $\mathcal{Z}$. Assume that the random vector $\tilde{y} = (\tilde{g}(z_1), \tilde{g}(z_2), \cdots, \tilde{g}(z_n))^\top$ satisfies $\|\tilde{y} - \mathbb{E}\tilde{y}\|_{\psi_2} \leq V < \infty$. Choose any*

$$\hat{j} \in \underset{j \in [m]}{\mathrm{argmin}} \left\{ \frac{1}{n}\sum_{i=1}^{n} |g_j(z_i) - \tilde{g}(z_i)|^2 \right\}.$$

*There exists a universal constant C such that for any $\delta \in (0, 1]$, with probability at least $1 - \delta$ we have*

$$\mathcal{L}\left(g_{\hat{\jmath}}\right) \leq \inf_{\gamma > 0} \left\{ (1 + \gamma) \min_{j \in [m]} \mathcal{L}\left(g_j\right) + C\left(1 + \gamma^{-1}\right) \left( \mathcal{L}(\mathbb{E}\widetilde{g}) + \frac{V^2 \log(m/\delta)}{n} \right) \right\}.$$

*Consequently,*

$$\mathbb{E}\mathcal{L}\left(g_{\hat{\jmath}}\right) \leq \inf_{\gamma > 0} \left\{ (1 + \gamma) \min_{j \in [m]} \mathcal{L}\left(g_j\right) + C\left(1 + \gamma^{-1}\right) \left( \mathcal{L}(\mathbb{E}\widetilde{g}) + \frac{V^2 (1 + \log m)}{n} \right) \right\}.$$

## B.5. Proof of Proposition B.2

We aim to apply Lemma B.10. We then analyze the behavior of $\hat{h}_\lambda$, which is obtained by running Algorithm 1 using the dataset $\mathcal{D}_{\text{train}}$. We use the decomposition established in Appendix A.3:

$$\hat{\eta}_\lambda - \eta^\star = (\mathbf{A}_1^\top \mathbf{A}_1 + n_1 \lambda \mathbf{I})^{-1} \mathbf{A}_1^\top \boldsymbol{b}_1 + (\mathbf{A}_1^\top \mathbf{A}_1 + n_1 \lambda \mathbf{I})^{-1} \mathbf{A}_1^\top \mathbf{W}_1 (\hat{\theta} - \theta^\star) - n_1 \lambda (\mathbf{A}_1^\top \mathbf{A}_1 + n_1 \lambda \mathbf{I})^{-1} \eta^\star$$
$$:= \mathscr{B}_1 + \mathscr{P}_1 + \mathscr{C}_1.$$

**Lemma B.8.** *Under the event $\mathscr{F}_{\text{good}}$, $\|\hat{\eta}_\lambda - \mathbb{E}_{\boldsymbol{\varepsilon}_1}[\hat{\eta}_\lambda]\|_{\psi_2}^2 \lesssim \frac{1}{\gamma} \sigma^2$.*

*Proof.* Using the established error decomposition, we get

$$\begin{aligned}
\|\hat{\eta}_\lambda - \mathbb{E}_{\boldsymbol{\varepsilon}_1}[\hat{\eta}_\lambda]\|_{\psi_2} &\lesssim \|(\mathbf{A}_1^\top \mathbf{A}_1 + n_1 \lambda \mathbf{I})^{-1} \mathbf{A}_1^\top \mathbf{W}_1 (\hat{\theta} - \mathbb{E}_{\boldsymbol{\varepsilon}_1}[\hat{\theta}])\|_{\psi_2} \\
&\lesssim \|(\mathbf{A}_1^\top \mathbf{A}_1 + n_1 \lambda \mathbf{I})^{-1} \mathbf{A}_1^\top \mathbf{W}_1 (\mathbf{Z}_1^\top \mathbf{Z}_1 + n_1 \lambda_0 \mathbf{I})^{-1} \mathbf{Z}_1^\top \boldsymbol{\varepsilon}_1\|_{\psi_2} \\
&\leq \sigma \|(\mathbf{A}_1^\top \mathbf{A}_1 + n_1 \lambda \mathbf{I})^{-1} \mathbf{A}_1^\top \mathbf{W}_1 (\mathbf{Z}_1^\top \mathbf{Z}_1 + n_1 \lambda_0 \mathbf{I})^{-1} \mathbf{Z}_1^\top\|_{\text{op}} \\
&\leq \sigma \|(\mathbf{A}_1^\top \mathbf{A}_1 + n_1 \lambda \mathbf{I})^{-1} \mathbf{A}_1^\top \mathbf{W}_1 (\mathbf{Z}_1^\top \mathbf{Z}_1 + n_1 \lambda_0 \mathbf{I})^{-\frac{1}{2}}\|_{\text{op}} \\
&\overset{(i)}{\lesssim} \sigma \sqrt{\frac{1}{\gamma}} \|(\mathbf{A}_1^\top \mathbf{A}_1 + n_1 \lambda \mathbf{I})^{-1} \mathbf{A}_1^\top\|_{\text{op}} \\
&\lesssim \frac{\sigma}{\sqrt{n_1 \lambda \gamma}} \\
&\lesssim \sigma \sqrt{\frac{1}{\gamma}}
\end{aligned}$$

where (i) holds by Equation (B.3) and Remark A.8. $\qquad\square$

**Lemma B.9.** *Under the event $\mathscr{F}_{\text{good}}$, we have*

$$\|\mathbb{E}_{\boldsymbol{\varepsilon}_1}[\hat{\eta}_\lambda] - \eta^\star\|_{\mathbb{H}}^2 \lesssim \|\eta^\star\|_{\mathbb{H}}^2 + \frac{1}{\gamma} \|\theta^\star\|_{\mathbb{F}}^2.$$

*Proof.* Using the same argument as in Appendix A.3, we get

$$\begin{aligned}
&\|\mathbb{E}_{\boldsymbol{\varepsilon}_1}[\hat{\eta}_\lambda] - \eta^\star\|_{\mathbb{H}} \\
&= \|(\mathbf{A}_1^\top \mathbf{A}_1 + n_1 \lambda \mathbf{I})^{-1} \mathbf{A}_1^\top \boldsymbol{b}_1 + (\mathbf{A}_1^\top \mathbf{A}_1 + n_1 \lambda \mathbf{I})^{-1} \mathbf{A}_1^\top \mathbf{W}_1 (\mathbb{E}_{\boldsymbol{\varepsilon}_1}[\hat{\theta}] - \theta^\star) - n_1 \lambda (\mathbf{A}_1^\top \mathbf{A}_1 + n_1 \lambda \mathbf{I})^{-1} \eta^\star\|_{\mathbb{H}} \\
&\leq \|(\mathbf{A}_1^\top \mathbf{A}_1 + n_1 \lambda \mathbf{I})^{-1} \mathbf{A}_1^\top \boldsymbol{b}_1\|_{\mathbb{H}} + \|(\mathbf{A}_1^\top \mathbf{A}_1 + n_1 \lambda \mathbf{I})^{-1} \mathbf{A}_1^\top \mathbf{W}_1 (\mathbb{E}_{\boldsymbol{\varepsilon}_1}[\hat{\theta}] - \theta^\star)\|_{\mathbb{H}} \\
&\quad + \|n_1 \lambda (\mathbf{A}_1^\top \mathbf{A}_1 + n_1 \lambda \mathbf{I})^{-1} \eta^\star\|_{\mathbb{H}}.
\end{aligned}$$

We bound each term. First, we see that

$$\begin{aligned}
\|(\mathbf{A}_1^\top \mathbf{A}_1 + n_1 \lambda \mathbf{I})^{-1} \mathbf{A}_1^\top \boldsymbol{b}_1\|_{\mathbb{H}}^2 &\lesssim \frac{1}{n_1 \lambda} \|(\mathbf{A}_1^\top \mathbf{A}_1 + n_1 \lambda \mathbf{I})^{-1/2} \mathbf{A}_1^\top \boldsymbol{b}_1\|_{\mathbb{H}}^2 \\
&\lesssim \frac{1}{n_1 \lambda} \|(\mathbf{A}_1^\top \mathbf{A}_1 + n_1 \lambda \mathbf{I})^{-1/2} \mathbf{A}_1^\top\|_{\text{op}}^2 \|\boldsymbol{b}_1\|_2^2
\end{aligned}$$

$$\lesssim \frac{1}{n_1 \lambda} \log n \|\theta^\star\|_{\mathbb{F}}^2 \quad \text{(by definition of } \mathscr{F}_{\text{good}})$$
$$\lesssim \|\theta^\star\|_{\mathbb{F}}^2.$$

Next, we see that

$$\|(\mathbf{A}_1^\top \mathbf{A}_1 + n_1 \lambda \mathbf{I})^{-1} \mathbf{A}_1^\top \mathbf{W}_1 (\mathbb{E}_{\boldsymbol{\varepsilon}_1}[\hat{\theta}] - \theta^\star)\|_{\mathbb{H}}$$
$$\lesssim \|(\mathbf{A}_1^\top \mathbf{A}_1 + n_1 \lambda \mathbf{I})^{-1} \mathbf{A}_1^\top \mathbf{W}_1 (\mathbf{Z}_1^\top \mathbf{Z}_1 + n_1 \lambda_0 \mathbf{I})^{-1} n_1 \lambda_0 \theta^\star\|_{\mathbb{H}}$$
$$\lesssim \frac{1}{\sqrt{n_1 \lambda}} \|(\mathbf{A}_1^\top \mathbf{A}_1 + n_1 \lambda \mathbf{I})^{-\frac{1}{2}} \mathbf{A}_1^\top \mathbf{W}_1 (\mathbf{Z}_1^\top \mathbf{Z}_1 + n_1 \lambda_0 \mathbf{I})^{-1} n_1 \lambda_0 \theta^\star\|_{\mathbb{H}}$$
$$\lesssim \frac{1}{\sqrt{n_1 \lambda}} \|\mathbf{W}_1 (\mathbf{Z}_1^\top \mathbf{Z}_1 + n_1 \lambda_0 \mathbf{I})^{-1} n_1 \lambda_0 \theta^\star\|_2$$
$$\lesssim \frac{1}{\sqrt{n_1 \lambda}} \|\mathbf{W}_1 (\mathbf{Z}_1^\top \mathbf{Z}_1 + n_1 \lambda_0 \mathbf{I})^{-\frac{1}{2}}\|_{\text{op}} \|(\mathbf{Z}_1^\top \mathbf{Z}_1 + n_1 \lambda_0 \mathbf{I})^{-\frac{1}{2}} n_1 \lambda_0 \theta^\star\|_{\mathbb{F}}$$
$$\lesssim \frac{\sqrt{n_1 \lambda_0}}{\sqrt{n_1 \lambda}} \|\theta^\star\|_{\mathbb{F}} \|\mathbf{W}_1 (\mathbf{Z}_1^\top \mathbf{Z}_1 + n_1 \lambda_0 \mathbf{I})^{-\frac{1}{2}}\|_{\text{op}}$$
$$\lesssim \sqrt{\frac{1}{\gamma}} \|\theta^\star\|_{\mathbb{F}} \quad \text{(by } \lambda \gtrsim \lambda_0 \text{ and Equation (B.3))}.$$

Lastly,

$$\|n_1 \lambda (\mathbf{A}_1^\top \mathbf{A}_1 + n_1 \lambda \mathbf{I})^{-1} \eta^\star\|_{\mathbb{H}}^2 \lesssim \|\eta^\star\|_{\mathbb{H}}^2.$$

Combining the obtained bounds, we finally prove the lemma. □

**Final Step: Applying Lemma B.10.** Fix $\lambda \in \Lambda$ and define the random function

$$f_\lambda(a) := \langle \phi(a), \hat{\eta}_\lambda - \eta^\star \rangle_{\mathbb{H}}, \qquad a \in \mathcal{A}.$$

Then

$$\|f_\lambda\|_n^2 = \mathcal{E}_{\mathbb{H}}^{\text{in}}(\hat{\eta}_\lambda), \qquad \|f_\lambda\|_{L^2(\mathcal{P}_{\text{ref}})}^2 = \mathcal{E}_{\mathbb{H}}(\hat{\eta}_\lambda).$$

We will apply Lemma B.10 to $f_\lambda$. For this final comparison step, we do not condition on the pseudo-test points $\{\tilde{a}_i\}_{i=1}^n$, nor on the pseudo-test covariance event in Equation (B.4). The preceding bounds for $\hat{\eta}_\lambda$ use only the training split and the training query points, so after conditioning on those training-side design/query quantities, $f_\lambda$ may still be random through the training noise but remains independent of the i.i.d. pseudo-test points $\{\tilde{a}_i\}_{i=1}^n$.

First, by kernel boundedness and Lemma B.8,

$$\|\langle \phi(\tilde{a}_1), \hat{\eta}_\lambda - \mathbb{E}_{\boldsymbol{\varepsilon}_1}[\hat{\eta}_\lambda] \rangle_{\mathbb{H}}\|_{\psi_2} \lesssim \sqrt{\xi} \|\hat{\eta}_\lambda - \mathbb{E}_{\boldsymbol{\varepsilon}_1}[\hat{\eta}_\lambda]\|_{\psi_2} \lesssim \sqrt{\frac{\xi}{\gamma}} \sigma.$$

Also, by kernel boundedness and Lemma B.9,

$$|\langle \phi(\tilde{a}_1), \mathbb{E}_{\boldsymbol{\varepsilon}_1}[\hat{\eta}_\lambda] - \eta^\star \rangle_{\mathbb{H}}| \leq \|\phi(\tilde{a}_1)\|_{\mathbb{H}} \|\mathbb{E}_{\boldsymbol{\varepsilon}_1}[\hat{\eta}_\lambda] - \eta^\star\|_{\mathbb{H}} \lesssim \sqrt{\xi} \left( \|\eta^\star\|_{\mathbb{H}} + \frac{1}{\sqrt{\gamma}} \|\theta^\star\|_{\mathbb{F}} \right).$$

Hence, for any $\varepsilon \in (0, 1/2]$, there exists a constant $c_1 > 1$ such that

$$\mathbb{P}\left( |f_\lambda(\tilde{a}_1)| > c_1 \sqrt{\xi} \left( \frac{\sigma}{\sqrt{\gamma}} \sqrt{\log(1/\varepsilon)} + \|\eta^\star\|_{\mathbb{H}} + \frac{1}{\sqrt{\gamma}} \|\theta^\star\|_{\mathbb{F}} \right) \right) \leq \varepsilon.$$

Next, note that the sub-Gaussian variable here is the centered term

$$\langle \phi(\tilde{a}_1), \hat{\eta}_\lambda - \mathbb{E}_{\boldsymbol{\varepsilon}_1}[\hat{\eta}_\lambda] \rangle_{\mathbb{H}},$$

conditional on $\tilde{a}_1$. For each fixed $\tilde{a}_1$, the randomness comes only from $\hat{\eta}_\lambda$ through $\varepsilon_1$, and the resulting $\psi_2$ bound is uniform in $\tilde{a}_1$ because $\|\phi(\tilde{a}_1)\|_{\mathbb{H}}^2 = K_{\mathbb{H}}(a, a) \leq \xi$; a final application of the tower property then yields the same order for the unconditional fourth moment. Using this observation together with $(x + y)^4 \leq 8(x^4 + y^4)$ and the standard fact that $\mathbb{E}|X|^4 \lesssim \|X\|_{\psi_2}^4$ for sub-Gaussian $X$, we obtain

$$\mathbb{E}|f_\lambda(\tilde{a}_1)|^4 \lesssim \mathbb{E}\big|\langle\phi(\tilde{a}_1), \hat{\eta}_\lambda - \mathbb{E}_{\varepsilon_1}[\hat{\eta}_\lambda]\rangle_{\mathbb{H}}\big|^4 + \big|\langle\phi(\tilde{a}_1), \mathbb{E}_{\varepsilon_1}[\hat{\eta}_\lambda] - \eta^\star\rangle_{\mathbb{H}}\big|^4$$

$$\lesssim \xi^2\left(\frac{\sigma}{\sqrt{\gamma}} + \|\eta^\star\|_{\mathbb{H}} + \frac{1}{\sqrt{\gamma}}\|\theta^\star\|_{\mathbb{F}}\right)^4.$$

Therefore, for sufficiently large constants $c_2, c_3 > 1$, define

$$r_n := c_2\sqrt{\xi}\left(\frac{\sigma}{\sqrt{\gamma}}\sqrt{\log n} + \|\eta^\star\|_{\mathbb{H}} + \frac{1}{\sqrt{\gamma}}\|\theta^\star\|_{\mathbb{F}}\right), \qquad U_n := c_3\sqrt{\xi}\left(\frac{\sigma}{\sqrt{\gamma}} + \|\eta^\star\|_{\mathbb{H}} + \frac{1}{\sqrt{\gamma}}\|\theta^\star\|_{\mathbb{F}}\right).$$

Then

$$\mathbb{P}\big(|f_\lambda(\tilde{a}_1)| > r_n\big) \leq n^{-16}, \qquad \mathbb{E}|f_\lambda(\tilde{a}_1)|^4 \leq U_n^4.$$

**Lemma B.10** (Part 1 of Lemma D.5 from Wang 2026). *Let $\{z_i\}_{i=1}^n$ be i.i.d. samples in a space $\mathcal{Z}$. For any $g : \mathcal{Z} \to \mathbb{R}$, define*

$$\|g\|_n := \left(\frac{1}{n}\sum_{i=1}^n g^2(z_i)\right)^{1/2}, \qquad \|g\|_{L^2} := \sqrt{\mathbb{E}[g^2(z_1)]}.$$

*Let $f : \mathcal{Z} \to \mathbb{R}$ be a random function independent of $\{z_i\}_{i=1}^n$. Suppose that*

$$\mathbb{P}\big(|f(z_1)| > r\big) \leq \varepsilon \qquad \text{and} \qquad \mathbb{E}|f(z_1)|^4 \leq U^4$$

*for some deterministic $r, U \geq 0$ and $\varepsilon \in (0, 1)$. Then, for any $\delta \in (0, 1)$,*

$$\mathbb{P}\left(\big|\|f\|_n - \|f\|_{L^2}\big| \leq r\sqrt{\frac{7\log(2/\delta)}{n}} + U\varepsilon^{1/4}\right) \geq 1 - n\varepsilon - \delta.$$

Applying Lemma B.10 with $z_i = \tilde{a}_i$, $f = f_\lambda$, $\varepsilon = n^{-16}$, and $\delta = n^{-13}$, we get for each fixed $\lambda \in \Lambda$,

$$\Big|\sqrt{\mathcal{E}_{\mathbb{H}}^{\text{in}}(\hat{\eta}_\lambda)} - \sqrt{\mathcal{E}_{\mathbb{H}}(\hat{\eta}_\lambda)}\Big| \lesssim r_n\sqrt{\frac{\log n}{n}} + U_n n^{-6}$$

with probability at least $1 - n^{-15} - n^{-13}$. Using $(x + y)^2 \leq 2x^2 + 2y^2$, this implies

$$\mathcal{E}_{\mathbb{H}}^{\text{in}}(\hat{\eta}_\lambda) \lesssim \mathcal{E}_{\mathbb{H}}(\hat{\eta}_\lambda) + \frac{r_n^2\log n}{n} + U_n^2 n^{-12},$$

$$\mathcal{E}_{\mathbb{H}}(\hat{\eta}_\lambda) \lesssim \mathcal{E}_{\mathbb{H}}^{\text{in}}(\hat{\eta}_\lambda) + \frac{r_n^2\log n}{n} + U_n^2 n^{-12}.$$

By definition of $r_n, U_n$, the last two terms are $\lesssim \mathcal{O}$ up to logarithmic factors, and $U_n^2 n^{-12}$ is negligible. All bounds above depend on $\lambda$ only through the condition $\lambda \gtrsim \lambda_0$, hence they hold uniformly over $\lambda \in \Lambda$. Since $|\Lambda| \leq n$, a union bound over $\lambda \in \Lambda$, together with the high-probability training-side events used in the preceding lemmas, yields overall probability at least $1 - 2n^{-11}$, which proves Proposition B.2.

## C. Proof of Theorem 4.6 (Lower Bound)

The lower bound is easy to obtain on simpler subclasses, for example by taking $f^\star(x, a)$ to depend only on $a$, or by considering settings where $X$ and $A$ are independent, in which case the problem reduces immediately to ordinary one-dimensional nonparametric regression. However, those subclasses do not reflect the main challenge of the paper, namely confounding under weak overlap. For completeness, we therefore present a genuinely confounded hard instance and show that the same minimax rate already persists there.

**Confounded Hard Instance.** We prove the lower bound by restricting the minimax problem to a confounded hard subclass. The case $d = 1$ is even simpler, since there are no covariates and the argument reduces directly to standard random-design regression. We therefore present the confounded construction for $d \geq 2$.

Let $\mathcal{P}_{\text{ref}} = \mathcal{U}([0,1])$, and write $x = (x_1, \ldots, x_{d-1}) \in [0,1]^{d-1}$. Define

$$u(x) := 2x_1 - 1, \qquad v(a) := 2a - 1, \qquad p_\gamma := \min\{1, 2\gamma\}, \qquad c_\gamma := 1 - \frac{\gamma}{p_\gamma}.$$

Then $p_\gamma \in [\gamma, 2\gamma]$ and $c_\gamma \in [0, 1/2]$. Choose a constant $\tau > 0$ sufficiently small so that, viewing $u$ as a function on $[0,1]^d$ that does not depend on $a$,

$$\|\tau u\|_{H^\beta([0,1]^d)} \leq \frac{1}{2}.$$

Next choose $c_0 \in (0,1]$ sufficiently small, depending only on $(\beta, d, \tau)$, such that for every $h \in H^\beta([0,1])$ with

$$\|h\|_{H^\beta([0,1])} \leq 1 \qquad \text{and} \qquad h(0) = 0,$$

the lifted function $(x, a) \mapsto c_0 h(a)$ has $H^\beta([0,1]^d)$-norm at most $1/2$. This is possible because the map $h \mapsto h(\cdot)$, viewed as a function on $[0,1]^{d-1} \times [0,1]$, is continuous from $H^\beta([0,1])$ into $H^\beta([0,1]^d)$. For such $h$, define

$$f_h^\star(x, a) := c_0 h(a) + \tau u(x).$$

Then $f_h^\star \in H^\beta([0,1]^d)$ and $\|f_h^\star\|_{H^\beta([0,1]^d)} \leq 1$.

For each such $h$, consider the data-generating process:

$$X \sim \mathcal{U}([0,1]^{d-1}),$$
$$A \mid X = x \sim (1 - p_\gamma)\delta_0 + p_\gamma q_x(a)\, da, \qquad q_x(a) := 1 + c_\gamma u(x)v(a),$$
$$Y = c_0 h(A) + \tau u(X) + \varepsilon, \qquad \varepsilon \sim \mathcal{N}(0,1),$$

with $\varepsilon$ independent of $(X, A)$. Since $c_\gamma \leq 1/2$, we have $q_x(a) \in [1 - c_\gamma, 1 + c_\gamma] \subset [1/2, 3/2]$, so $q_x$ is a valid density on $[0,1]$. Moreover,

$$p_\gamma q_x(a) \geq p_\gamma(1 - c_\gamma) = \gamma \qquad \text{for all } (x, a) \in [0,1]^{d-1} \times [0,1].$$

Hence

$$\frac{d\mathcal{P}_{\text{ref}}}{d\mathcal{P}_{A|X=x}}(a) \leq \frac{1}{\gamma} \qquad \text{for } \mathcal{P}_{\text{ref}}\text{-a.e. } a,$$

so Relative Overlap of degree $\gamma$ holds.

The TEF under this construction is

$$h_h^\star(a) := \mathbb{E}[f_h^\star(X, a)] = c_0 h(a),$$

because $\mathbb{E}[u(X)] = 0$. The model is genuinely confounded whenever $c_\gamma > 0$: by Bayes' rule,

$$\mathbb{E}[u(X) \mid A = a] = \frac{\mathbb{E}[u(X)(1 + c_\gamma u(X)v(a))]}{\mathbb{E}[1 + c_\gamma u(X)v(a)]} = c_\gamma v(a)\mathbb{E}[u(X)^2] = \frac{c_\gamma}{3}(2a - 1),$$

since $u(X) \sim \mathcal{U}([-1,1])$ has mean 0 and second moment $1/3$. Therefore

$$\mathbb{E}[Y \mid A = a] = c_0 h(a) + \frac{\tau c_\gamma}{3}(2a - 1),$$

which differs from the TEF unless $c_\gamma = 0$.

Now define the transformed response

$$Z_i := Y_i - \tau u(X_i) = c_0 h(A_i) + \varepsilon_i.$$

The map $(x, a, y) \mapsto (x, a, z = y - \tau u(x))$ is one-to-one, so it is equivalent to work with the sample $(X_i, A_i, Z_i)_{i=1}^n$. For one observation, the joint density under parameter $h$ is

$$q_h(x, a, z) = p_{X,A}(x, a)\, \varphi(z - c_0 h(a)),$$

where $\varphi$ is the standard normal density and $p_{X,A}$ does not depend on $h$. Hence, for the full sample,

$$q_h^{(n)}(x_{1:n}, a_{1:n}, z_{1:n}) = \underbrace{\prod_{i=1}^n p_{X,A}(x_i, a_i)}_{\text{free of } h} \cdot \underbrace{\prod_{i=1}^n \varphi(z_i - c_0 h(a_i))}_{\text{depends on the data only through } T^n},$$

where

$$T^n := \big((A_i, Z_i)\big)_{i=1}^n.$$

Therefore $T^n$ is sufficient for $h$ by the factorization theorem.

Let $\hat{h} = \delta(X_{1:n}, A_{1:n}, Y_{1:n})$ be any estimator with values in $L^2(\mathcal{P}_{\text{ref}})$, and define its Rao–Blackwellization

$$\tilde{h}(T^n) := \mathbb{E}_h[\hat{h} \mid T^n].$$

Since the conditional law of the full data given $T^n$ is parameter-free, $\tilde{h}$ is again a valid estimator. By conditional Jensen's inequality,

$$\mathbb{E}_h\left[\|\tilde{h} - h_h^\star\|_{L^2(\mathcal{P}_{\text{ref}})}^2\right] \leq \mathbb{E}_h\left[\|\hat{h} - h_h^\star\|_{L^2(\mathcal{P}_{\text{ref}})}^2\right].$$

Thus it suffices to lower bound the reduced experiment $(A_i, Z_i)_{i=1}^n$.

Because $\mathbb{E}[u(X)] = 0$, the perturbation averages out in the marginal law of $A$, and we obtain

$$A \sim (1 - p_\gamma)\delta_0 + p_\gamma \mathcal{U}([0,1]).$$

Let

$$N := \sum_{i=1}^n \mathbf{1}\{A_i \neq 0\} \sim \text{Binomial}(n, p_\gamma).$$

Conditional on $N = m$, the $m$ informative observations are i.i.d.

$$(U_j, c_0 h(U_j) + \varepsilon_j), \qquad U_j \sim \mathcal{U}([0,1]),$$

while the remaining observations have $A_i = 0$ and $Z_i = \varepsilon_i$, which are uninformative because $h(0) = 0$. Hence, conditional on $N = m$, the reduced experiment is exactly one-dimensional random-design nonparametric regression on the fixed-radius Sobolev class

$$\mathcal{H}_0 := \{c_0 h : h \in H^\beta([0,1]), \|h\|_{H^\beta([0,1])} \leq 1, h(0) = 0\}.$$

By the classical minimax lower bound for Sobolev regression (e.g. via Fano or Assouad; see Tsybakov 2009), there exists a constant $c_\beta > 0$, depending only on $\beta$ and $c_0$, such that for every $m \geq 1$,

$$\inf_{\bar{h}} \sup_{r \in \mathcal{H}_0} \mathbb{E}_r\left[\|\bar{h} - r\|_{L^2(\mathcal{P}_{\text{ref}})}^2 \mid N = m\right] \geq c_\beta \, m^{-\frac{2\beta}{2\beta+1}}.$$

The restriction $r(0) = 0$ does not change the rate, since the standard packing can be chosen with support away from $0$.

Now define the event

$$\mathcal{G} := \{N \leq 2p_\gamma n\}.$$

If $p_\gamma \geq 1/2$, then $\mathcal{G}$ holds automatically because $N \leq n \leq 2p_\gamma n$. If $p_\gamma < 1/2$, a binomial upper-tail bound yields

$$\mathbb{P}(\mathcal{G}) \geq 1 - \exp(-c \, p_\gamma n)$$

for a universal constant $c > 0$. In particular, $\mathbb{P}(\mathcal{G}) \geq 1/2$ whenever $p_\gamma n$ is larger than a sufficiently large absolute constant. Using the conditional lower bound on $\mathcal{G}$, we obtain

$$\inf_{\bar{h}} \sup_{r \in \mathcal{H}_0} \mathbb{E}_r\left[\|\bar{h} - r\|_{L^2(\mathcal{P}_{\text{ref}})}^2\right] \geq \mathbb{P}(\mathcal{G}) \cdot c_\beta (2p_\gamma n)^{-\frac{2\beta}{2\beta+1}}$$

$$\gtrsim (p_\gamma n)^{-\frac{2\beta}{2\beta+1}}$$

$$\gtrsim (\gamma n)^{-\frac{2\beta}{2\beta+1}},$$

where the last step uses $p_\gamma \leq 2\gamma$. Thus the desired lower bound holds for all sufficiently large $n$, which is the regime relevant for the minimax-rate statement. By sufficiency and Rao–Blackwell, the same lower bound holds for estimators based on the original sample $(X_i, A_i, Y_i)_{i=1}^n$. This proves Theorem 4.6.

# D. Proof of Remark 4.4

We prove the claim on the same design event $\mathscr{E}_{\mathrm{good}}$ introduced in Section A.2. Recall from Lemma A.6 that

$$\mathbb{P}(\mathscr{E}_{\mathrm{good}}) \geq 1 - 5n^{-11}.$$

Under the assumptions of Theorem 4.3, we have $\boldsymbol{\Sigma}_{\mathrm{samp}} = \boldsymbol{\Sigma}_{\mathrm{ref}}$. For brevity, write

$$\boldsymbol{\Sigma} := \boldsymbol{\Sigma}_{\mathrm{ref}} = \boldsymbol{\Sigma}_{\mathrm{samp}}.$$

Because $\lambda \gtrsim \log n / n$, Equation (A.13) together with Remark A.8 implies that on $\mathscr{E}_{\mathrm{good}}$,

$$\frac{1}{c}\left(\frac{1}{n}\mathbf{A}^\top\mathbf{A} + \lambda\mathbf{I}\right) \preceq \boldsymbol{\Sigma} + \lambda\mathbf{I} \preceq c\left(\frac{1}{n}\mathbf{A}^\top\mathbf{A} + \lambda\mathbf{I}\right) \tag{D.1}$$

for some absolute constant $c > 1$.

Let $\hat{\eta}_\lambda$ and $\eta^\star$ denote the Hilbertian elements corresponding to $\hat{h}_\lambda$ and $h^\star$, respectively. By the reproducing property and bounded kernel,

$$\|\hat{h}_\lambda - h^\star\|_{L^\infty(\mathcal{A})} \leq \sup_{a\in\mathcal{A}} \|\phi(a)\|_{\mathbb{H}} \|\hat{\eta}_\lambda - \eta^\star\|_{\mathbb{H}} \lesssim \|\hat{\eta}_\lambda - \eta^\star\|_{\mathbb{H}}.$$

Thus it suffices to control the $\mathbb{H}$-norm error.

Using the decomposition from Appendix A.3,

$$\hat{\eta}_\lambda - \eta^\star = \mathscr{B} + \mathscr{P}_v + \mathscr{P}_b + \mathscr{C}.$$

We bound each term on $\mathscr{E}_{\mathrm{good}}$.

**Step 1: Empirical-Centering Term $\mathscr{B}$.** We have

$$\begin{aligned}
\|\mathscr{B}\|_{\mathbb{H}}^2 &= \left\|(\mathbf{A}^\top\mathbf{A} + n\lambda\mathbf{I})^{-1}\mathbf{A}^\top\boldsymbol{b}\right\|_{\mathbb{H}}^2 \\
&= \boldsymbol{b}^\top\mathbf{A}(\mathbf{A}^\top\mathbf{A} + n\lambda\mathbf{I})^{-2}\mathbf{A}^\top\boldsymbol{b} \\
&\leq \frac{1}{n\lambda}\boldsymbol{b}^\top\mathbf{A}(\mathbf{A}^\top\mathbf{A} + n\lambda\mathbf{I})^{-1}\mathbf{A}^\top\boldsymbol{b} \\
&\leq \frac{1}{n\lambda}\|\boldsymbol{b}\|_2^2.
\end{aligned}$$

Under $\mathscr{E}_{\mathrm{good}}$, $|b(a_j')|^2 \lesssim \log n / n$ for all $j \in [n]$, so $\|\boldsymbol{b}\|_2^2 \lesssim \log n$. Therefore,

$$\|\mathscr{B}\|_{\mathbb{H}}^2 \lesssim \frac{\log n}{n\lambda}. \tag{D.2}$$

**Step 2: Propagated-Variance Term $\mathscr{P}_v$.** Recall that

$$\mathscr{P}_v = (\mathbf{A}^\top\mathbf{A} + n\lambda\mathbf{I})^{-1}\mathbf{A}^\top\mathbf{W}(\mathbf{Z}^\top\mathbf{Z} + n\lambda_0\mathbf{I})^{-1}\mathbf{Z}^\top\boldsymbol{\varepsilon}.$$

Conditional on $\mathscr{E}_{\mathrm{good}}$, Hanson–Wright yields an event $\mathscr{E}_v$ satisfying

$$\mathbb{P}(\mathscr{E}_v \mid \mathscr{E}_{\mathrm{good}}) \geq 1 - n^{-11}$$

on which

$$\begin{aligned}
\|\mathscr{P}_v\|_{\mathbb{H}}^2 &\lesssim \log n \operatorname{Tr}\left((\mathbf{A}^\top\mathbf{A} + n\lambda\mathbf{I})^{-1}\mathbf{A}^\top\mathbf{W}(\mathbf{Z}^\top\mathbf{Z} + n\lambda_0\mathbf{I})^{-1}\mathbf{Z}^\top\mathbf{Z}(\mathbf{Z}^\top\mathbf{Z} + n\lambda_0\mathbf{I})^{-1}\mathbf{W}^\top\mathbf{A}(\mathbf{A}^\top\mathbf{A} + n\lambda\mathbf{I})^{-1}\right) \\
&\leq \log n \operatorname{Tr}\left((\mathbf{A}^\top\mathbf{A} + n\lambda\mathbf{I})^{-1}\mathbf{A}^\top\mathbf{W}(\mathbf{Z}^\top\mathbf{Z} + n\lambda_0\mathbf{I})^{-1}\mathbf{W}^\top\mathbf{A}(\mathbf{A}^\top\mathbf{A} + n\lambda\mathbf{I})^{-1}\right).
\end{aligned}$$

By Lemma A.7 and $\lambda_0 \asymp \log n / n$,

$$\mathbf{W}(\mathbf{Z}^\top\mathbf{Z} + n\lambda_0\mathbf{I})^{-1}\mathbf{W}^\top \lesssim \frac{1}{\gamma}\mathbf{I}_n \qquad \text{on } \mathscr{E}_{\mathrm{good}}.$$

Hence, on $\mathscr{E}_{\text{good}} \cap \mathscr{E}_v$,

$$\|\mathscr{P}_v\|_{\mathbb{H}}^2 \lesssim \frac{\log n}{\gamma} \operatorname{Tr}\left(\mathbf{A}(\mathbf{A}^\top \mathbf{A} + n\lambda\mathbf{I})^{-2}\mathbf{A}^\top\right)$$

$$= \frac{\log n}{n\gamma} \operatorname{Tr}\left[\left(\frac{1}{n}\mathbf{A}^\top \mathbf{A}\right)\left(\frac{1}{n}\mathbf{A}^\top \mathbf{A} + \lambda\mathbf{I}\right)^{-2}\right].$$

**A Useful Empirical-Effective-Dimension Bound.** Define

$$\widehat{\Gamma}(\lambda) := \operatorname{Tr}\left[\left(\frac{1}{n}\mathbf{A}^\top \mathbf{A}\right)\left(\frac{1}{n}\mathbf{A}^\top \mathbf{A} + \lambda\mathbf{I}\right)^{-1}\right].$$

Since $t/(t+\lambda)^2 \leq \lambda^{-1} t/(t+\lambda)$ for every $t \geq 0$, the previous display implies

$$\operatorname{Tr}\left[\left(\frac{1}{n}\mathbf{A}^\top \mathbf{A}\right)\left(\frac{1}{n}\mathbf{A}^\top \mathbf{A} + \lambda\mathbf{I}\right)^{-2}\right] \leq \frac{\widehat{\Gamma}(\lambda)}{\lambda}.$$

Next, Equation (D.1) implies

$$\left(\frac{1}{n}\mathbf{A}^\top \mathbf{A} + \lambda\mathbf{I}\right)^{-1} \preceq c(\mathbf{\Sigma} + \lambda\mathbf{I})^{-1} \qquad \text{on } \mathscr{E}_{\text{good}}.$$

Hence, on $\mathscr{E}_{\text{good}}$,

$$\widehat{\Gamma}(\lambda) \lesssim \operatorname{Tr}\left[\left(\frac{1}{n}\mathbf{A}^\top \mathbf{A}\right)(\mathbf{\Sigma} + \lambda\mathbf{I})^{-1}\right]$$

$$= \frac{1}{n}\sum_{j=1}^n \left\langle \phi(a_j'), (\mathbf{\Sigma} + \lambda\mathbf{I})^{-1}\phi(a_j')\right\rangle_{\mathbb{H}}.$$

Define

$$U_j := \left\langle \phi(a_j'), (\mathbf{\Sigma} + \lambda\mathbf{I})^{-1}\phi(a_j')\right\rangle_{\mathbb{H}}, \qquad j \in [n].$$

Then $U_j \geq 0$ and

$$\mathbb{E}[U_j] = \operatorname{Tr}\left[(\mathbf{\Sigma} + \lambda\mathbf{I})^{-1}\mathbf{\Sigma}\right] = \Gamma(\lambda).$$

Also, by boundedness of the kernel,

$$0 \leq U_j \leq \frac{\|\phi(a_j')\|_{\mathbb{H}}^2}{\lambda} \lesssim \frac{1}{\lambda}.$$

Consequently,

$$\operatorname{Var}(U_j) \leq \mathbb{E}[U_j^2] \lesssim \frac{\mathbb{E}[U_j]}{\lambda} = \frac{\Gamma(\lambda)}{\lambda}.$$

Therefore, by Bernstein's inequality, there exists an event $\mathscr{E}_\Gamma$ with

$$\mathbb{P}(\mathscr{E}_\Gamma) \geq 1 - n^{-11}$$

such that on $\mathscr{E}_\Gamma$,

$$\frac{1}{n}\sum_{j=1}^n U_j \lesssim \Gamma(\lambda) + \sqrt{\frac{\Gamma(\lambda)\log n}{n\lambda}} + \frac{\log n}{n\lambda}$$

$$\lesssim \Gamma(\lambda) + \frac{\log n}{n\lambda},$$

where the second line uses $2\sqrt{ab} \leq a + b$. Combining the displays above, on $\mathscr{E}_{\text{good}} \cap \mathscr{E}_v \cap \mathscr{E}_\Gamma$,

$$\operatorname{Tr}\left[\left(\frac{1}{n}\mathbf{A}^\top \mathbf{A}\right)\left(\frac{1}{n}\mathbf{A}^\top \mathbf{A} + \lambda\mathbf{I}\right)^{-2}\right] \lesssim \frac{1}{\lambda}\left(\Gamma(\lambda) + \frac{\log n}{n\lambda}\right).$$

Therefore,

$$\|\mathscr{P}_v\|_{\mathbb{H}}^2 \lesssim \frac{\log n}{n\gamma\lambda}\left(\Gamma(\lambda) + \frac{\log n}{n\lambda}\right) \qquad \text{on } \mathscr{E}_{\text{good}} \cap \mathscr{E}_v \cap \mathscr{E}_\Gamma. \tag{D.3}$$

**Step 3: Propagated-Bias Term $\mathscr{P}_b$.** Recall that

$$\mathscr{P}_b = -n\lambda_0(\mathbf{A}^\top\mathbf{A} + n\lambda\mathbf{I})^{-1}\mathbf{A}^\top\mathbf{W}(\mathbf{Z}^\top\mathbf{Z} + n\lambda_0\mathbf{I})^{-1}\theta^\star.$$

Using

$$\|(\mathbf{A}^\top\mathbf{A} + n\lambda\mathbf{I})^{-1}\mathbf{A}^\top\|_{\text{op}}^2 \le \frac{1}{n\lambda},$$

we obtain

$$\|\mathscr{P}_b\|_{\mathbb{H}} \le \frac{n\lambda_0}{\sqrt{n\lambda}}\|\mathbf{W}(\mathbf{Z}^\top\mathbf{Z} + n\lambda_0\mathbf{I})^{-1}\theta^\star\|_2.$$

Moreover, by Lemma A.7

$$\begin{aligned}
\|\mathbf{W}(\mathbf{Z}^\top\mathbf{Z} + n\lambda_0\mathbf{I})^{-1}\theta^\star\|_2^2 &= \left\langle\theta^\star, (\mathbf{Z}^\top\mathbf{Z} + n\lambda_0\mathbf{I})^{-1}\mathbf{W}^\top\mathbf{W}(\mathbf{Z}^\top\mathbf{Z} + n\lambda_0\mathbf{I})^{-1}\theta^\star\right\rangle \\
&\lesssim \frac{1}{\gamma}\left\langle\theta^\star, (\mathbf{Z}^\top\mathbf{Z} + n\lambda_0\mathbf{I})^{-1}\theta^\star\right\rangle \\
&\le \frac{1}{\gamma n\lambda_0}\|\theta^\star\|_{\mathbb{F}}^2.
\end{aligned}$$

Combining the displays above and using $\lambda_0 \asymp \log n / n$, we get

$$\|\mathscr{P}_b\|_{\mathbb{H}}^2 \lesssim \frac{\log n}{n\gamma\lambda}\|\theta^\star\|_{\mathbb{F}}^2 \lesssim \frac{\log n}{n\gamma\lambda}\|f^\star\|_{\mathcal{F}}^2 \qquad \text{on } \mathscr{E}_{\text{good}}. \tag{D.4}$$

**Step 4: Regularization-Bias Term $\mathscr{C}$.** Assume $h^\star \in \mathcal{H}^s$ for some $s \in (1, 2]$. By the standard spectral characterization of $\mathcal{H}^s$, this means

$$\eta^\star = \mathbf{\Sigma}^{(s-1)/2}g^\star \qquad \text{for some } g^\star \in \mathbb{H}.$$

Using Equation (D.1),

$$\begin{aligned}
\|\mathscr{C}\|_{\mathbb{H}}^2 = \lambda^2\left\|\left(\frac{1}{n}\mathbf{A}^\top\mathbf{A} + \lambda\mathbf{I}\right)^{-1}\eta^\star\right\|_{\mathbb{H}}^2 &\lesssim \lambda\left\langle\eta^\star, \left(\frac{1}{n}\mathbf{A}^\top\mathbf{A} + \lambda\mathbf{I}\right)^{-1}\eta^\star\right\rangle \\
&\lesssim \lambda\left\langle\eta^\star, (\mathbf{\Sigma} + \lambda\mathbf{I})^{-1}\eta^\star\right\rangle \\
&= \lambda\left\langle\mathbf{\Sigma}^{(s-1)/2}g^\star, (\mathbf{\Sigma} + \lambda\mathbf{I})^{-1}\mathbf{\Sigma}^{(s-1)/2}g^\star\right\rangle.
\end{aligned}$$

Let $\{(\rho_j, e_j)\}_{j\ge1}$ be the eigensystem of $\mathbf{\Sigma}$. Expanding in this basis gives

$$\begin{aligned}
\lambda\left\langle\mathbf{\Sigma}^{(s-1)/2}g^\star, (\mathbf{\Sigma} + \lambda\mathbf{I})^{-1}\mathbf{\Sigma}^{(s-1)/2}g^\star\right\rangle &= \lambda\sum_{j\ge1}\frac{\rho_j^{s-1}}{\rho_j + \lambda}\langle g^\star, e_j\rangle_{\mathbb{H}}^2 \\
&\le \lambda^{s-1}\sum_{j\ge1}\langle g^\star, e_j\rangle_{\mathbb{H}}^2 \\
&= \lambda^{s-1}\|g^\star\|_{\mathbb{H}}^2,
\end{aligned}$$

where we used $\lambda\rho^{s-1}/(\rho + \lambda) \le \lambda^{s-1}$ for every $\rho \ge 0$ and every $s \in (1, 2]$. Therefore,

$$\|\mathscr{C}\|_{\mathbb{H}} \lesssim \lambda^{\frac{s-1}{2}}. \tag{D.5}$$

**Step 5: Combine the Bounds.** On the event $\mathscr{E}_{\text{good}} \cap \mathscr{E}_v \cap \mathscr{E}_\Gamma$, Equations (D.2) to (D.5) imply

$$\|\hat{h}_\lambda - h^\star\|_{L^\infty(\mathcal{A})} \lesssim \lambda^{\frac{s-1}{2}} + \sqrt{\frac{\log n}{n\lambda}} + \sqrt{\frac{\log n}{n\gamma\lambda}\left(\Gamma(\lambda) + \frac{\log n}{n\lambda}\right)} + \frac{\|f^\star\|_{\mathcal{F}}\sqrt{\log n}}{\sqrt{n\gamma\lambda}}. \tag{D.6}$$

Under polynomial eigendecay $\rho_j \lesssim j^{-2\ell}$, we have

$$\Gamma(\lambda) \lesssim \lambda^{-\frac{1}{2\ell}}.$$

Since $n_{\text{eff}} = \gamma n$ and $\gamma \leq 1$,

$$\sqrt{\frac{\log n}{n\lambda}} \leq \sqrt{\frac{\log n}{n_{\text{eff}}\lambda}}.$$

Also, because $\lambda \gtrsim \log n / n$, we have

$$\frac{\log n}{n\lambda} \lesssim 1.$$

Thus Equation (D.6) becomes

$$\|\hat{h}_\lambda - h^\star\|_{L^\infty(\mathcal{A})} \lesssim \widetilde{\mathcal{O}}\left(\lambda^{\frac{s-1}{2}} + n_{\text{eff}}^{-1/2}\lambda^{-\frac{1}{2}-\frac{1}{4\ell}} + (1 + \|f^\star\|_{\mathcal{F}})\, n_{\text{eff}}^{-1/2}\lambda^{-\frac{1}{2}}\right).$$

Now choose

$$\lambda \asymp n_{\text{eff}}^{-1/(s+1/(2\ell))}.$$

With this choice,

$$\lambda^{\frac{s-1}{2}} \asymp n_{\text{eff}}^{-1/2}\lambda^{-\frac{1}{2}-\frac{1}{4\ell}} \asymp n_{\text{eff}}^{-\frac{s-1}{2(s+\frac{1}{2\ell})}}.$$

The remaining term satisfies

$$n_{\text{eff}}^{-1/2}\lambda^{-1/2} = n_{\text{eff}}^{-\frac{s-1+\frac{1}{2\ell}}{2(s+\frac{1}{2\ell})}},$$

which is of strictly smaller order because $\ell > 1/2$. Hence, on $\mathscr{E}_{\text{good}} \cap \mathscr{E}_v \cap \mathscr{E}_\Gamma$,

$$\|\hat{h}_\lambda - h^\star\|_{L^\infty(\mathcal{A})} = \widetilde{\mathcal{O}}\left(n_{\text{eff}}^{-\frac{s-1}{2(s+\frac{1}{2\ell})}}\right).$$

Finally,

$$\mathbb{P}(\mathscr{E}_{\text{good}} \cap \mathscr{E}_v \cap \mathscr{E}_\Gamma) \geq 1 - 5n^{-11} - n^{-11} - n^{-11} \geq 1 - n^{-10}$$

for all $n$ sufficiently large. This proves Remark 4.4.

## E. Additional Experimental Details

### E.1. Synthetic Data

For *our proposed method* (Algorithm 2), we partition the dataset into a training set $\mathcal{D}_{\text{train}}$ and a validation set $\mathcal{D}_{\text{val}}$, with sizes $n_1 = |\mathcal{D}_{\text{train}}|$ and $n_2 = |\mathcal{D}_{\text{val}}|$, respectively. The regularization grid is defined as $\Lambda = \{c\frac{2^{k-1}}{n_1} : k = 1, \ldots, 9\}$ with $c = 0.1$, and the proxy regularizer is fixed at $\tilde{\lambda} = c/n_2$. For pseudo-outcome construction, we set $\mathcal{P}_{\text{samp}}$ to the uniform distribution on the treatment domain. Regarding kernel specification, we employ a Matérn kernel with smoothness parameter $\nu = 1.5$ for the first-stage nuisance estimation ($f^\star$), and a smoother Matérn kernel with $\nu = 2.5$ for the second-stage target estimation ($h^\star$).

For the *Plug-in KRR* baseline, we adopt the same data-splitting strategy and regularization grid $\Lambda$. Candidate models $\{\hat{f}_\lambda\}_{\lambda \in \Lambda}$ are trained on $\mathcal{D}_{\text{train}}$ using KRR with regularizer $\lambda$, and the final regularizer is selected by minimizing the Mean Squared Error (MSE) on $\mathcal{D}_{\text{val}}$. We use a Matérn kernel with smoothness parameter $\nu = 1.5$.

The *Direct Regression* baseline is similarly implemented using KRR with a Matérn kernel ($\nu = 1.5$). We also apply the same hold-out validation strategy: candidate estimators are trained on $\mathcal{D}_{\text{train}}$ using the regularization grid $\Lambda$, and the final regularizer is selected by MSE on $\mathcal{D}_{\text{val}}$.

For all methods, we construct a length-scale grid by matching the kernel correlation at the median pairwise distance and cross-validating over this grid, following the widely used median-heuristic for kernel bandwidth selection (Garreau et al., 2017; Gretton et al., 2012). Specifically, let $r_{\mathrm{med}}$ denote the median pairwise distance. For each correlation level $\rho$ in a chosen set within $[0.15, 0.85]$, we define the candidate length-scale $\ell_\rho$ by solving $K(r_{\mathrm{med}}; \ell_\rho)/K(0; \ell_\rho) = \rho$. We then cross-validate over the resulting grid $\{\ell_\rho\}$ to select the final length-scale $\ell$. In our proposed framework, the length-scale of the second-stage kernel $K_{\mathcal{H}}$ is set equal to that of $K_{\mathcal{F}}$. In this experiment, the selected length-scale was $\ell = 3$ for our method and Plug-in KRR, and $\ell = 2$ for Direct Regression.

### E.2. Semi-real Data

We employ a tensor-product Laplace kernel (Matérn kernel with smoothness $\nu = 0.5$) for nuisance estimation (for both our method and Plug-in KRR) and a Matérn kernel with smoothness $\nu = 1.5$ for the second-stage KRR. For Direct Regression, we use a Laplace kernel. The length-scale parameters for the nuisance kernel $K_{\mathcal{F}}$ were selected via cross-validation on a grid constructed around the median heuristic (similar to the synthetic experiments), resulting in $\ell_x = 13$ (covariates) and $\ell_a = 6000$ (treatment). These length-scale values were used for both our method and Plug-in KRR. For Direct Regression, we selected the length-scale using the same procedure, which yielded $\ell = 3000$.

For our method, we set the second-stage kernel length-scale of $K_{\mathcal{H}}$ equal to the treatment length-scale of $K_{\mathcal{F}}$ (i.e., $\ell = 6000$). We denote the sizes of the training and validation sets as $n_1 = |\mathcal{D}_{\mathrm{train}}|$ and $n_2 = |\mathcal{D}_{\mathrm{val}}|$, respectively. Consistent with the synthetic experiments, we define the regularization grid as $\Lambda = \{c\frac{2^{k-1}}{n_1} : k = 1, \ldots, 9\}$ with $c = 0.1$, and set the proxy regularizer to $\tilde{\lambda} = c/n_2$. The second-stage regression requires a sampling distribution $\mathcal{P}_{\mathrm{samp}}$; since approximately 94% of the observed treatments in the Job Corps data lie within $[0, 3000]$, we sample $a$ uniformly from this range.

For Plug-in KRR (LOOCV), to ensure computational efficiency during LOOCV, we employ the Nyström approximation with $m = 700$ landmark points.

For our method and the Plug-in baseline, covariates are normalized to $[0, 1]^d$. For DML baselines, to be consistent with Colangelo & Lee (2026), categorical covariates are one-hot encoded, and the DML baselines use the same standardization procedures as in the original study.

## F. Technical Lemmas

We first present key trace-class lemmas from Wang (2026).

**Lemma F.1** (Lemma E.1 from Wang 2026). *Suppose that $\boldsymbol{x} \in \mathbb{R}^d$ is a zero-mean random vector with $\|\boldsymbol{x}\|_{\psi_2} \leq 1$. There exists a universal constant $C > 0$ such that for any symmetric and positive semi-definite matrix $\boldsymbol{\Sigma} \in \mathbb{R}^{d \times d}$,*

$$\mathbb{P}\left(\boldsymbol{x}^\top \boldsymbol{\Sigma} \boldsymbol{x} \leq C \operatorname{Tr}(\boldsymbol{\Sigma}) t\right) \geq 1 - e^{-r(\boldsymbol{\Sigma})t}, \quad \forall t \geq 1.$$

*Here $r(\boldsymbol{\Sigma}) = \operatorname{Tr}(\boldsymbol{\Sigma})/\|\boldsymbol{\Sigma}\|_2$ is the effective rank of $\boldsymbol{\Sigma}$.*

**Lemma F.2** (Corollary E.1 from Wang 2026). *Let $\{\boldsymbol{x}_i\}_{i=1}^n$ be i.i.d. random elements in a separable Hilbert space $\mathbb{H}$ with $\boldsymbol{\Sigma} = \mathbb{E}(\boldsymbol{x}_i \otimes \boldsymbol{x}_i)$ being trace class. Define $\hat{\boldsymbol{\Sigma}} = \frac{1}{n}\sum_{i=1}^n \boldsymbol{x}_i \otimes \boldsymbol{x}_i$. Choose any constant $\gamma \in (0, 1)$ and define an event $\mathcal{A} = \{(1 - \gamma)(\boldsymbol{\Sigma} + \lambda \boldsymbol{I}) \preceq \hat{\boldsymbol{\Sigma}} + \lambda \boldsymbol{I} \preceq (1 + \gamma)(\boldsymbol{\Sigma} + \lambda \boldsymbol{I})\}$. If $\|\boldsymbol{x}_i\|_{\mathbb{H}} \leq \xi$ holds almost surely for some constant $\xi$, then there exists a constant $C \geq 1$ determined by $\gamma$ such that $\mathbb{P}(\mathcal{A}) \geq 1 - \delta$ holds so long as $\delta \in (0, 1/14]$ and $\lambda \geq \frac{C\xi \log(n/\delta)}{n}$. Hence with probability at least $1 - \delta$, we have*

$$\frac{1}{C\xi}(\widehat{\boldsymbol{\Sigma}} + \frac{\log(n/\delta)}{n}\mathbf{I}) \preceq \boldsymbol{\Sigma} + \frac{\log(n/\delta)}{n}\mathbf{I} \preceq C\xi(\widehat{\boldsymbol{\Sigma}} + \frac{\log(n/\delta)}{n}\mathbf{I}).$$

