# OpenReview forum: "Estimating Continuous Treatment Effects with Two-Stage Kernel Ridge Regression"
_ICML.cc/2026/Conference — ICML 2026 regular_

### Official Review · Reviewer_i1N8 · 2026-02-23

**Soundness:** 4
**Presentation:** 3
**Significance:** 3
**Originality:** 3
**Overall Recommendation:** 4
**Confidence:** 5

**Summary:**

The paper proposes a novel method for estimating continuos treatment effect with kernel regression. Although the exsiting method just takes the avearge of psudo-outcomes of a fixed treatment, the paper proposes to conduct another regression on psudo-outcomes with multiple treatments. By doing this, we can explicitly control the smoothness of the structral function. The theory suggests this is a minimax optimial, and the proposed method outperforms existing methods in an empirical evaluation.

**Compliance With Llm Reviewing Policy:**

Affirmed.

**Final Justification:**

After the rebuttal, I have raised the score

**Key Questions For Authors:**

My main questions are:
- When the practioner cares more about $L_2$ norm rather than $L_\infty$?
- What is the connection to tensor-product formulation?
- The paper states that the derivation "relaxes the requirement that the bound does not assume $f^*$ must lie strictly within a fixed ball of the RKHS", but I do not see what technical difficulty prevents us from assuming the RKHS norm of true function grows with the data size. All we have to do is to replace the constant upper-bound of the norm to some data size dependent variable, right? I would like to see whether we need true innovation here.
- To me it is unclear whether there is any theoretical background in Algorithm 2, or is it just heuristics?

**Limitations:**

yes

**Strengths And Weaknesses:**

**Soundness**: The technical discussion seems reasonable to me, though I have not checked the proof thoroughly.

**Presentation**: The paper is well-presented

**Significance**: It would be worth noting that metrics are somewhat different from the existing continuous treatment effect estimators. In the exising work (e.g. Singh et al. 2024), we care the $L_\infty$ norm i.e. $\max_a |\hat{h}(a) - h(a)|$, meaning that we want all poential outcomes to be estimated well. It is natural extension from semi-parametric case $A\in\{0, 1\}$, in which we are interested in estimating both treated/untreated potential outcomes. On the other hand, the paper considers the $L_2$  norm on $P_{ref}$ as shown in Eq 2.1. We might need more careful discusstion around this, as different metrics means different optimality.

**Originality** : Although the idea of conducting another regression to control the smoothness of structrual function is novel, there might be alternative way to achieve this; One is to use tensor-product kernel (Singh et al. 2024) which defines the kernel $k_{\mathcal{F}}(z, z') = k_{\mathcal{H}}(a,a')k_x(x, x')$. Then, the estimated structural function can be written as $\hat{h}(\cdot) = \sum_{ij} \alpha_i k_{\mathcal{H}}(\cdot, a_i) k_x(x_i, x_j)$, which lies in the RKHS $\mathbb{H}$. I would expect more detailed discussion with this.

---

> ### Author Rebuttal · Authors · 2026-03-29
>
> We sincerely thank you for the careful review and constructive feedback. Below we address each of your questions and respectfully ask you to reconsider the submission in light of these clarifications.
>
> **Significance and Q1. $L^2$ and $L^\\infty$.**
>
> In nonparametric function estimation, MISE is a standard performance metric; see, for example, [1]. It captures the overall quality of the estimated treatment-effect function.
> This metric is also used in continuous treatment effect estimation. For example, [2,5] use integrated-error criteria for dose-response estimation.
> The experiments in Kennedy et al. (2017) likewise assess the quality of the estimated curve through *integrated error*.
> $L^2$-type criteria are also standard in binary-treatment CATE estimation; see, for example, Nie and Wager (2021) and [4].
>
> For RKHS-based methods, $L^\\infty$ guarantees typically require additional structure. In Singh et al. (2024), for instance, uniform control is obtained under a source condition and a tensor-product RKHS. **This creates a practical tradeoff: $L^\\infty$ guarantees often require stronger modeling assumptions, and these can be nontrivial burdens for practitioners.** In practice, a user may not wish to impose a tensor-product kernel. For that reason, we focus on the globally meaningful $L^2$ criterion in the main text, while noting that an $L^\\infty$ extension is available under an additional source condition.
>
> *$L^\\infty$ Results under a source condition.* If one imposes a Singh-style source condition directly on the *target* RKHS $\\mathcal{H}$, our estimator also admits an $L^\\infty$ guarantee. The key advantage is that both the source condition and the eigendecay assumption are imposed only on $\\mathcal{H}$, whereas in Singh et al. the analogous assumptions are imposed on the substantially more complex joint nuisance space $\\mathcal{F}$. Since $\\mathcal{H}$ is a treatment-only space, it is often expected to exhibit faster eigendecay than $\\mathcal{F}$, making the resulting $L^\\infty$ guarantee strictly stronger than the corresponding Singh et al. result.
> We will add this as a remark in the main text and provide the full argument in the appendix.
>
> **Remark ($L^\\infty$ bound under a source condition).** Assume $h^\\star \\in \\mathcal{H}^s$ for some $s\\in(1,2]$, and suppose the target kernel $K_{\\mathcal{H}}$ has polynomial eigendecay $\\rho_j\\lesssim j^{-2\\ell}$. Then, under the assumptions of Theorem 4.3, we obtain the $L^\\infty$ rate $n_{\\mathrm{eff}}^{-(s-1)/(2(s+1/(2\\ell)))}$ if $\\lambda \\asymp n_{\\mathrm{eff}}^{-1/(s+1/(2\\ell))}$.
>
> **Q2 and Originality. Connection to the tensor-product kernel.**
>
> Singh et al.'s tensor-product estimator can indeed be represented as a function in the treatment RKHS $\\mathcal{H}$ after marginalization, but their theory remains governed by $\\mathcal{F}$, with no guarantee of adaptation to $\\mathcal{H}$. We will make this connection explicit in the revision.
>
> **Q3. About $\\|f^\\star\\|_\\mathcal{F}$.**
>
> In standard nonparametric analysis, once the RKHS norm grows beyond a constant scale, the rate itself typically deteriorates.
> By contrast, our result can still achieve the optimal learning bound even when $\\|f^\\star\\|_\\mathcal{F}$ is larger than $\\mathcal{O}(1)$.
>
> For example, [3] show that over Sobolev-type RKHS classes $H^\\beta(\\mathcal{Z})$, $\\mathcal{Z} \\subset \\mathbb{R}^d$, the minimax regression rate with $n$ samples scales as $n^{-2\\beta/(d+2\\beta)} \\| f^\\star\\|_\\mathcal{F}^{2d/(d+2\\beta)}$. So once $\\|f^\\star\\|_F$ is allowed to grow, the rate necessarily deteriorates.
>
> Our result is stronger in this respect. For example, if $h^\\star\\in H^1(\\mathcal{A})$ has bounded norm, then the rate $n^{-2/3}$ is still achieved when $\\|f^\\star\\|_{\\mathcal{F}}^2\\lesssim n^{1/3}$. This statement does not depend on any special choice of kernel for $\\mathcal{F}$; for instance, it can accommodate NTK-type kernels as well. This is the sense in which our result relaxes the usual fixed-radius RKHS-ball assumption, and we will clarify this point explicitly in the revision.
>
> **Q4. Theoretical Background in Algorithm 2.**
>
> Algorithm 2 is not heuristic: it is supported by a strong theoretical guarantee, namely the adaptivity result in Theorem 4.4, which is one of the main contributions of the paper.
>
> **References.**
>
> **[1]** Tsybakov (2009). *Introduction to Nonparametric Estimation*.
>
> **[2]** Schwab et al. (2020). *Learning Counterfactual Representations for Estimating Individual Dose-Response Curves*.
>
> **[3]** Green et al. (2021). *Minimax Optimal Regression over Sobolev Spaces via Laplacian Regularization on Neighborhood Graphs*.
>
> **[4]** Foster and Syrgkanis (2023). *Orthogonal Statistical Learning*.
>
> **[5]** Huang and Chan (2018). *Joint sufficient dimension reduction for estimating continuous treatment effect functions*.

---

> > ### Author Rebuttal · Reviewer_i1N8 · 2026-04-02
> >
> > The rebuttal adequately addresed my concerns. I have raised my score.

---

> > > ### Author Response · Authors · 2026-04-03
> > >
> > > We are delighted that your concerns have been addressed. Thank you very much for your time and for favorably re-assessing our work.

---

### Official Review · Reviewer_CvHX · 2026-03-11

**Soundness:** 4
**Presentation:** 3
**Significance:** 3
**Originality:** 3
**Overall Recommendation:** 5
**Confidence:** 3

**Summary:**

This submission studies the estimation of the causal effect, more precisely, of the so-called treatment effect function (TEF), in a model with continuous treatments, under the assumption of a distribution shift (confounding). A two-step estimation procedure is proposed with the first step being the estimation of the so-called conditional mean function, resulting in so-called pseudo-outcomes. These pseudo-outcomes are the used in the second step to estimate the TEF by simply averaging the estimated conditional mean function over the observed covariates (marginalization). Both steps employ kernel ridge regression. The second step involves estimating a function that at least as smooth, if not smoother, then the (nuisance) target in step 1, and thus step 1 is a more difficult problem. It is shown that it is the simpler problem that determines the final convergence rate. In a special case, when the reference measure is uniform, the derived rates are shown to be minimax optimal in respective smoothness classes. All this is done using an RKHS framework, and it seems that in the theoretical work, the goal was to push towards weak assumptions. The provided numerical studies indicate a strong performance compared to competitors.

**Compliance With Llm Reviewing Policy:**

Affirmed.

**Final Justification:**

The rebuttal has addressed all my questions. I like this work and keep my score.

**Key Questions For Authors:**

1) I was surprised that there was no discussion about the relation to the work by Wang (2023) [It seems that this work recently has been published in the Annals of Statistics, by the way.] This clearly seems to be related work, and a discussion is certainly warranted, because appreciating the differences to this work seems important for the assessment of the novelty of the contribution of the submission. To me the response to this question is important and it will likely impact my recommendation.

2) page 5: I found the $\mu_{\rm ref}$ assumption interesting. I was wondering, though, how much more generality this assumption adds (as compared to a uniform lower and upper bound, say, on the conditional densities. I was also wondering if $\mu_{\rm ref}$ could also be discrete? If yes, would your result also apply to a the case of discrete treatments? Can you perhaps comment on that?

3) page 3: In the definition of the RKHS, why do you use the TWO Hilbert spaces $\mathbb H$ and $\mathcal H$?

4) page 3, Assumption 2.2 and Assumption 2.3: Both assumptions say that something is bounded by an absolute constant ($\sigma$ in assumption 2.2 and $||h^{\star}||$ in assumption 2.3). Thinking about this assumption, and I can imagine that $\mathcal H$ is a subset of ${\mathbb R}^p$, and $p$ is growing with sample size, for instance. Then you are saying that $||h^*||_{\mathcal H}$ is bounded by a constant not depending on the sample size. How stringent is this assumptions? Thinking about the bound for $\sigma$, I was a little more confused, because this is an assumption on a marginal (one-dimensional) distribution. Do you mean that the error distribution might depend on the sample size, and then it should be in such a way that $\sigma$ is uniformly bounded? If yes, do you have a real example in mind, or is this simply for mathematical generality? I am asking this mostly out of curiosity…

5) Numerical results: I did see your comment about addressing misspecification in future work. I still was wondering whether you have some experience in how robust the presented results are to the choice of the function spaces (i.e. the kernels)? Did you also try a misspecified model where the target function does not lie in the model class?

**Limitations:**

Yes.

**Strengths And Weaknesses:**

PRESENTATION:

The manuscript is competently written, although it reads somewhat dense. One of my main open questions (see also my question to the author(s) below) is about the relation to the work by Wang (2023). I also have some more detailed questions/comments on the presentation:

Some more detailed comments on presentation:

- page 1, last paragraph (Theory) provides an informal description of the contribution. When I read this for the first time, it was stumbling over the phrase “…the intrinsic overlap of the data…” Perhaps this can be rephrased?

- page 3, Assumption 2.2: It seems that the purpose of the literature given right below the assumptions is to convince the reader that these assumptions are common in the literature. However, I somehow was unable to find the sub-Gaussian assumptions in the cited references. Maybe I did not look close enough, but if I did, please provide the appropriate references.

- page 5: Theorem 4.3 mentions $P_{\rm ref}$. I did not see this defined.

- References: Both Ghorbandi et al. (2021) and Liang and Rakhlin (2020) are entirely capitalized. Please check. In other references (such as in Künzel et al.  2019, and Robins, 2000) the outlets are not using capital letters. This is in contrast to the other references. Please check

ORIGINALITY AND SIGNIFICANCE: The contributions appear to be novel, and I like the fact that the strong theoretical work is leading to a methodology that appears to outperform competitors. However, the final assessment of significance and originality  somehow hinges on a better understanding of the relation to Wang (2026) that I was asking about (see my question below).

SOUNDNESS: I did not check all the technical details. The parts that I considered more closely appear to be technically sound, and overall my impression is the the author(s) have a good grasp of the underlying technical tools.

---

> ### Author Rebuttal · Authors · 2026-03-29
>
> Thank you for recognizing the importance of our work and for the positive evaluation.
> We believe our method is especially attractive under weak overlap, poorly behaved covariate distributions, or unstable conditional-density estimation. We also clarify the distinction from Wang (2023) in Q1 below.
>
> **C1. Overlap terminology.**
>
> We would revise the sentence as follows:
>
> *"We prove that our method adapts to unknown structural parameters, specifically the degree of treatment overlap in the data and the spectral decay of the kernel."*
>
> **C2. Sub-Gaussian references.**
>
> Kennedy et al. (2017) effectively assume bounded pseudo-outcomes, which amounts to bounded $Y$, and Bonvini and Kennedy (2022) also work under boundedness-type conditions. Since boundedness is stronger than sub-Gaussianity, these references therefore impose stronger assumptions than ours, not weaker ones.
> We will revise the sentence accordingly and also mention binary-treatment references such as Nie and Wager (2021), where sub-Gaussian or closely related regular-noise assumptions are standard.
>
> **C3. Definition of the reference distribution.**
>
> In our notation, the MISE is defined with respect to a reference measure, denoted by $\\mu_{\\mathrm{ref}}$. We intend $\\mu_{\\mathrm{ref}}$ to be a probability measure, and we use $P_{\\mathrm{ref}}$ for the corresponding distribution.
> If the reference measure is not normalized, one can equivalently work with its normalized version.
> This was not stated clearly enough, and we will make it explicit in the revision to avoid confusion.
>
> **C4. Reference formatting.**
>
> We will correct it in the revision.
>
> **Q1. Relation to Wang (2023).**
>
> We will expand the discussion of Wang (2023) and make the methodological and analytical distinctions explicit.
>
> - We analyze a *two-stage KRR procedure with two different kernels* for continuous treatment effect estimation. The main proof-level novelty is that we treat the first-stage nuisance estimator as a random element rather than replacing it by a deterministic error bound. For this reason, our work differs from Wang (2023), DML-style analyses, and Singh et al. (2024) not only in setting, but also in methodology, analysis, and guarantees.
> - Wang (2023) studies model selection for single-stage KRR under covariate shift. In contrast, one of our main contributions is a finite-sample error analysis for the two-stage KRR estimator itself. Although we borrow technical lemmas from Wang (2023), the setting is substantially different: we perform model selection for a two-stage KRR procedure, and the causal estimand is defined across two different RKHSs, $\\mathcal{F}$ and $\\mathcal{H}$.
>
> **Q2. Discrete reference measure.**
>
> Yes, the reference measure can be discrete, as long as the relative-overlap condition continues to hold.
> In that case, the same theory applies, and the learning bound is governed by the eigenvalues of the kernel integral operator defined with respect to the discrete reference measure $\\mu_{\\text{ref}}$.
> This further illustrates the flexibility of our framework.
>
> **Q3. $\\mathcal{H}$ and $\\mathbb{H}$.**
>
> We use $\\mathcal{H}$ to denote the function class and $\\mathbb{H}$ to denote the Hilbert space itself. This distinction makes the later arguments both cleaner and more rigorous. The relation between the two is described around line 122 of the submission.
>
> **Q4. About $\\|h^\\star\\|_{\\mathcal{H}}$ and $\\sigma$.**
>
> First, $p$ is the treatment dimension. In many applications it is small, and in dose-response problems we typically have $p=1$. In such settings, assuming that $\\|h^\\star\\|_{\\mathcal{H}}$ is not excessively large is quite reasonable, so we view this as a fairly mild assumption in many applications of interest.
> Regarding the second part of the question, $\\sigma$ denotes a uniform bound on the noise level. This is satisfied, for example, by bounded responses or Gaussian noise with fixed variance, and assumptions of this kind are standard in the statistical literature. As you point out, such a parameter could in principle vary with the sample size, but in this paper we adopt the standard convention that it is uniformly bounded over $n$.
>
> **Q5. Empirical robustness to misspecification.**
>
> In fact, our semi-real experiment already contains a meaningful form of misspecification: the nuisance response is estimated using Generalized Random Forests, whereas our algorithm models it with a kernel method. In that sense, this experiment already reflects a form of misspecification of $\\mathcal{F}$.
> Nevertheless, our method still outperforms the GRF-based DML baseline.
> One likely reason is weak overlap: DML estimators contain an IPW-type component, so when overlap is weak (that is, when $\\gamma$ is small), the error can be substantially inflated.
> By contrast, our method is optimal with respect to the degree of overlap $\\gamma$.

---

> > ### Author Rebuttal · Reviewer_CvHX · 2026-04-02
> >
> > Thanks for your response to my questions. In particular, it clarified the connection to related prior work. I do like this work and I maintain my positive rating.

---

> > > ### Author Response · Authors · 2026-04-02
> > >
> > > Thank you for supporting our work and recognizing the value.

---

### Official Review · Reviewer_QzPS · 2026-03-11

**Soundness:** 2
**Presentation:** 3
**Significance:** 2
**Originality:** 1
**Overall Recommendation:** 3
**Confidence:** 5

**Summary:**

This paper studies nonparametric estimation of a continuous-treatment effect curve using a two-stage kernel ridge regression procedure. The paper also proposes a proxy-validation model-selection rule and proves upper bounds that scale with an effective sample size determined by overlap, together with a lower bound in a Sobolev setting. I think the paper is technically competent and addresses an important problem. The exposition is generally clear, and the structure-adaptation perspective, namely exploiting the fact that the marginal treatment-effect curve may be simpler than the full response surface, is a useful lens. However, I am not convinced the paper clears the bar for ICML. The main issue is novelty: the methodological idea and much of the positioning feel too close to existing continuous-treatment and kernel-based causal estimation work, and the paper does not make a sufficiently sharp technical leap beyond that literature. The theory is interesting but somewhat narrow, and the empirical section is not strong enough to compensate.

**Compliance With Llm Reviewing Policy:**

Affirmed.

**Key Questions For Authors:**

All the comments in the weakness section should have been better addressed in the main text.

**Limitations:**

Limitations are partially addressed in the conclusion section.

**Strengths And Weaknesses:**

Strengths

- The core observation, i.e., the TEF, being a marginal average of f^*, lives in a simpler function space than the full nuisance, is well-articulated and theoretically grounded. The decoupling of nuisance complexity from target complexity (Theorem 4.3) is the paper's most compelling contribution.

- Unlike standard doubly robust approaches that require estimating the generalized propensity score (conditional treatment density), this method sidesteps that entirely by relying on well-specification of the outcome model. This is a genuine practical advantage in high-dimensional settings where density estimation is fragile.

- The paper provides upper bounds (Theorem 4.3), a data-driven selection procedure with adaptivity guarantees (Theorem 4.4), and matching minimax lower bounds (Theorem 4.5). This constitutes a thorough theoretical treatment.

Weakness

- The novelty over prior two-stage regression work is not sufficiently clear. The core two-stage idea, first estimating the outcome surface and then marginalizing over covariates, is not new. Kennedy et al. (2017) already proposed pseudo-outcome regression for continuous treatments, and Singh (2022) and Singh et al. (2024) study kernel-based estimators in this setting. The paper argues that its contribution is adaptation to the complexity of the treatment-effect curve, but this distinction is not made sharply enough. In particular, the comparison to Singh et al. (2024) remains qualitative; the paper should provide a direct side-by-side comparison of the achieved rates and assumptions. Relatedly, the discussion of Liu et al. (2023) and Xia and Wainwright (2024) is too dismissive given the strong algorithmic similarity. A clearer account of what is genuinely new, especially relative to prior kernel ridge regression analyses such as Wang (2023), would substantially strengthen the paper.

- The well-specification assumption is strong and somewhat underemphasized. The theory relies on the assumption that the true outcome surface lies in the chosen reproducing kernel Hilbert space. This is a strong requirement, and considerably stronger than the robustness guarantees offered by doubly robust or debiased approaches, which remain consistent under weaker model conditions. The paper highlights avoidance of density estimation as an advantage, but this comes with a clear tradeoff: the method offers no protection against outcome-model misspecification. That tradeoff should be stated more explicitly, especially since the DML-style baselines used for comparison do not require exact well-specification of the outcome model.

- The term $|f^\star|{\mathcal F}^2 / (\gamma n)$ is potentially problematic and deserves more discussion. In Theorem 4.3, this term appears explicitly, and the paper allows $|f^\star|{\mathcal F}$ to grow with $n$. If this norm grows even moderately fast, the term can dominate the nominal convergence rate. The examples seem to implicitly assume $|f^\star|_{\mathcal F}=O(1)$, but this is not obviously innocuous in settings such as neural tangent kernels, where realistic functions may have large reproducing kernel Hilbert space norms. The paper should either state concrete conditions under which this term remains controlled, or explain more clearly how the guarantees weaken when it is not.

- The minimax lower bound addresses only a narrow special case. The lower bound is proved only in a highly restricted setting: a Sobolev class on [0,1], a treatment distribution with a particular point-mass-plus-uniform construction, and an outcome surface of the form $f^\star(x,a)=h^\star(a)$, so there is no meaningful covariate dependence. While this does validate the effective-sample-size dependence in a simplified regime, it does not capture the main challenge of the problem, namely the presence of nuisance covariates and confounding structure. As a result, the lower bound provides only limited support for the broader optimality claims in the paper.

- The proof strategy appears to rely heavily on existing technical machinery. A substantial portion of the analysis seems to be assembled from prior tools, especially Wang (2023), including the model-selection argument, the connecting lemma, and several key concentration steps. The remaining ingredients, such as the bound on W^\top W, appear to follow from standard operator concentration combined with the relative-overlap assumption. There is still value in adapting these tools to the causal setting, but the paper should be more transparent about which parts of the proof are technically new and which are direct applications of existing machinery. As written, the technical leap feels more incremental than the presentation suggests.

---

> ### Author Rebuttal · Authors · 2026-03-29
>
> We sincerely thank you for the careful review and constructive feedback. Below we address each concern and respectfully ask you to reconsider the submission in light of these clarifications.
>
> We respectfully disagree with the assessment that the technical novelty is limited. The key novelty is that we treat the nuisance estimator $\\hat f$ as a random object and explicitly exploit its fluctuation in the analysis.
>
> **W1. Novelty.**
>
> The main novelty lies not in using two stages per se, but in *how the first-stage nuisance error is analyzed*. Standard DML-type analyses treat nuisance estimation as an approximately fixed perturbation summarized by deterministic error bounds, so the final rate is driven by products of nuisance-estimation errors.
>
> Our analysis instead keeps the first-stage KRR estimator random and tracks how its fluctuation propagates through the pseudo-outcomes and is regularized at the second stage. This proof-level difference explains why first-stage undersmoothing matters and why the leading term can be governed by the simpler target RKHS $\\mathcal{H}$. To the best of our knowledge, this is the first analysis of a two-stage KRR procedure for continuous treatments that fully exploits first-stage randomness.
>
> Singh et al. (2024) is also kernel-based and studies causal functions in a unified framework, but for the comparison relevant here it remains **essentially plug-in**: the final learning bound is governed by the complexity of $\\mathcal{F}$, with no guarantee that it adapts to the simpler target space $\\mathcal{H}$.
>
> Wang (2023) studies model selection for single-stage KRR under covariate shift. In contrast, one of our main contributions is a finite-sample error analysis for the two-stage KRR estimator itself. Although we borrow technical lemmas from Wang (2023), the setting is substantially different: we perform model selection for a two-stage KRR procedure, and the causal estimand is defined across two different RKHSs, $\\mathcal{F}$ and $\\mathcal{H}$.
>
> Liu et al. (2023) and Xia and Wainwright (2024) also differ from our work in both setup and analysis: they study transfer learning and semi-supervised learning, not continuous treatment estimation, and treat the first stage through deterministic pseudo-response or nuisance-error terms.
>
> We will include the above discussions in the revision.
>
> **W2. Well-specification assumption.**
>
> There is a real tradeoff here. Beyond avoiding conditional-density estimation, our guarantees remain valid even when the covariate distribution is poorly behaved, for example discrete or lacking a regular density. We are also optimal in the weak-overlap regime, where $\\gamma$ may be small and serves as the key difficulty parameter, whereas DML is particularly fragile in such settings.
>
> | **Method** | **Pros** | **Cons** |
> | --- | --- | --- |
> | **DML** | Some robustness to misspecification | Needs strong overlap and conditional-density estimation; relies on regular covariate distributions. |
> | **Singh et al.** | Unified plug-in based framework for several causal functions | Error bound is governed by $\\mathcal{F}$, with no guarantee of adaptation to $\\mathcal{H}$. |
> | **Ours** | No conditional-density estimation; no covariate-distribution regularity requirement; optimal under weak overlap; adapts to the simpler target space $\\mathcal{H}$ | Requires well-specification of $f^\\star$ |
>
> No method dominates uniformly across all settings, but ours is especially attractive under weak overlap, poorly behaved covariate distributions, or unstable conditional-density estimation.
>
> We will include the above discussions in the revision.
>
> **W3. About $\\|f^\\star\\|_{\\mathcal{F}}^2$.**
>
> Due to the character limit here, please refer to our response to Reviewer i1N8's Q3, where we explain why allowing $\\|f^\\star\\|_{\\mathcal{F}}$ to grow beyond $\\mathcal{O}(1)$ is nontrivial and when the optimal learning rate is still preserved.
>
> **W4. Lower Bound.**
>
> The lower bound in the appendix is mathematically valid because minimax lower bounds only require a hard subclass, but we agree that the current subclass is too simple relative to the confounded setting emphasized in the paper. In response, we have now proved the same lower bound for a genuinely confounded subclass, and we will add this stronger statement in the revision. Specifically, for any $h \\in H^\\beta([0,1])$, one can construct an instance with $f^\\star(x,a)=h(a)+\\tau x$ and Relative Overlap of degree $\\gamma$ by placing a $1-2\\gamma$ fraction of the samples at $a=0$. After setting $Z_i:=Y_i-\\tau X_i$, the model reduces to $Z_i=h(A_i)+\\varepsilon_i$, and since the samples with $A_i=0$ are non-informative, the effective sample size is only of order $n\\gamma$, yielding the same lower bound $(\\gamma n)^{-2\\beta/(2\\beta+1)}$.
>
> **W5. Proof Novelty.**
>
> Please see W1 above for the main proof-level novelty, and our response to Reviewer CvHX's Q1 for further discussion of the relation to Wang (2023).

---

> > ### Author Rebuttal · Reviewer_QzPS · 2026-04-01
> >
> > I have a few follow-up questions:
> >
> > 1. You argue the main novelty is keeping the first stage estimator random rather than treating it through deterministic nuisance error bounds. Can you state precisely which theorem or rate in Singh et al. (2024) fails to recover your guarantee, and identify the exact term in your bound that is strictly improved by this random first stage analysis?
> >
> > 2. Your rebuttal claims the method is especially attractive under weak overlap and irregular covariate distributions, but the price is exact well specification of f^*. Can you give a concrete regime in which your method is provably preferable to doubly robust approaches after accounting for this stronger modeling assumption, rather than only qualitatively comparing pros and cons?
> >
> > 3. On the lower bound, you now say you can prove the same rate for a genuinely confounded subclass. What is the precise subclass, and in what sense does that lower bound match the full upper bound, beyond reproducing only the effective sample size dependence?

---

> > > ### Author Response · Authors · 2026-04-03
> > >
> > > We sincerely thank the reviewer for the thoughtful follow-up questions.
> > >
> > > **Q1**
> > >
> > > Singh et al. (2024, Theorem 2(1)) give the sup-norm rate $n^{-\\frac{c-1}{2(c+1/b)}}$, where $b$ is the polynomial eigendecay exponent of the *joint* kernel operator on $\\mathcal{F}$ and $c\\in(1,2]$ is the source-condition exponent. Since both are defined on the joint nuisance space $\\mathcal{F}$ over $(X,A)$, their guarantee is tied to ambient nuisance complexity rather than the simpler $\\mathcal{H}$.
> > >
> > > One concrete example is the Sobolev setting $\\mathcal{F}=H^\\beta(\\mathcal{Z})$, $\\mathcal{H}=H^\\beta(\\mathcal{A})$, with $\\mathcal{Z}\\subset\\mathbb{R}^d$ and $\\mathcal{A}\\subset\\mathbb{R}$. In this case, Singh et al.'s rate is $n^{-2\\beta/(4\\beta+d)}$ (with $b=2\\beta/d$ and $c=2$), which fails to recover our rate $n^{-2\\beta/(2\\beta+1)}$.
> > >
> > > In the appendix error analysis, the nuisance-propagated term $\\mathscr{P}$ (line 938) splits into variance $\\mathscr{P}_v$ and bias $\\mathscr{P}_b$ (line 1008). One gain from keeping the first stage random is that $\\mathscr{P}_v$ is absorbed into the $\\Gamma(\\lambda)/(\\gamma n)$ term in Theorem 4.3 through second-stage shrinkage $\\lambda$. Another gain is that first-stage undersmoothing (small $\\lambda_0 \\asymp 1/n$) makes $\\mathscr{P}_b$ only the mild overhead $\\|f^\\star\\|^2/n$ (line 1056). Thus $\\mathcal{F}$-complexity enters only through a mild overhead in Theorem 4.3, a key advantage of our method over prior work: we analyze bias and variance separately rather than controlling the overall nuisance error.
> > >
> > > **Q2**
> > >
> > > DML methods require estimating the conditional density $p_{A\\mid X=x}(a)$, denoted by $\\pi(a\\mid x)$. Let $\\inf_{a,x}\\pi(a\\mid x) \\gtrsim \\gamma$. Observe that
> > > $$
> > > \\left|\\frac{1}{\\hat\\pi}-\\frac{1}{\\pi}\\right|
> > > \\lesssim \\frac{1}{\\gamma^2}\\,|\\hat\\pi-\\pi|.
> > > $$
> > > In general, the error bound takes the form of the oracle regression rate in $\\mathcal{H}$ plus the second-order overhead
> > > $$
> > > \\frac{1}{\\gamma^2}\\,\\|(\\hat f-f^\\star)(\\hat \\pi-\\pi)\\|_{\\mathbb{X}}^2.
> > > $$
> > > Here $\\mathbb{X}$ denotes the norm appearing in the corresponding DML remainder bound. In Kennedy et al. and Colangelo and Lee, $\\mathbb{X}=L^\\infty$. In Bonvini and Kennedy, the remainder has a more complex form: it is controlled in an $L^2$ norm under the conditional measure $X\\mid A=a$. Either way, these requirements are substantially more demanding than a standard product of $L^2$ errors. A $\\gamma^{-2}$ factor also appears in binary ATE analyses.
> > >
> > > **Attractive regime:** $f^\\star \\in \\mathcal{F}$; overlap is weak (small $\\gamma$), **or** $\\|(\\hat f-f^\\star)(\\hat \\pi-\\pi)\\|_{\\mathbb{X}}^2$ is not small enough.
> > >
> > > **Concrete examples:** Consider a setting where $\\mathcal{F}$ is an NTK-RKHS, $\\mathcal{H}=H^2([0,1])$, and $f^\\star \\in \\mathcal{F}$.
> > >
> > > - (Case 1) $\\gamma \\asymp 1$, $\\|(\\hat f-f^\\star)(\\hat \\pi-\\pi)\\|_{\\mathbb{X}}^2 \\asymp n^{-2/3}$.
> > > - (Case 2) $\\gamma \\asymp n^{-1/3}$, $\\|(\\hat f-f^\\star)(\\hat \\pi-\\pi)\\|_{\\mathbb{X}}^2 \\asymp n^{-1}$.
> > >
> > > In Case 1, our bound retains the oracle rate $O(n^{-4/5})$, whereas the DML bound is only $O(n^{-2/3})$. In Case 2, our bound is $O(n^{-8/15})$, whereas the DML bound is only $O(n^{-1/3})$. Case 1 captures irregular covariates that make nuisance estimation slow, while Case 2 captures weak overlap despite fast nuisance rates.
> > >
> > > **Q3**
> > >
> > > The confounded hard subclass is the following. Set $\\mathcal{F} = H^\\beta([-1,1] \\times [0,1])$ and $\\mathcal{H} = H^\\beta([0,1])$. For simplicity, set $\\beta=2$. Fix $\\gamma<1/2$ and $\\tau>0$, and let $h$ vary over $\\mathcal{H}$ with $h(0)=0$. Draw $X\\sim \\text{Unif}([-1,1])$, and let
> > > $$
> > > p_x(a):=1+x\\Bigl(a-\\frac12\\Bigr).
> > > $$
> > > Then for $a\\in[0,1]$, set
> > > $$
> > > A\\mid X=x \\sim (1-2\\gamma)\\delta_0 + 2\\gamma\\,p_x(a)\\,da
> > > $$
> > > and let
> > > $$
> > > Y(a)=h(a)+\\tau X+\\varepsilon.
> > > $$
> > > Then
> > > $$
> > > f^\\star(x,a)=h(a)+\\tau x,
> > > \\qquad
> > > h^\\star(a)=\\mathbb{E}[Y(a)]=h(a),
> > > $$
> > > and Relative Overlap of degree $\\gamma$ holds.
> > >
> > > Now set $Z_i:=Y_i-\\tau X_i$. Then $Z_i=h(A_i)+\\varepsilon_i$, so the problem reduces to one-dimensional nonparametric regression (by a short Rao-Blackwell/sufficiency argument). Because $h(0)=0$, only $N\\asymp \\gamma n$ observations are informative on this subclass, which yields the minimax lower bound
> > > $$
> > > (\\gamma n)^{-\\frac{4}{5}}.
> > > $$
> > > Since one does not know the exact form of $f^\\star$, fitting it in the ambient class $\\mathcal{F}$ yields the slower MISE bound $n^{-\\frac{2}{3}}$. This helps explain why the lower bound is governed by the $\\mathcal{H}$-side barrier.
> > >
> > > Specializing Theorem 4.3(a) to $\\ell=\\beta=2$ matches this lower bound; see also Example 4.1. Thus the match is twofold: the fundamental limit is the TEF class $\\mathcal{H}$, not $\\mathcal{F}$, and weak overlap enters through the necessary effective sample size $n_{\\mathrm{eff}}=\\gamma n$.
> > >
> > > We will add these examples and clarifications in the revision.

---

### Official Review · Reviewer_SNs9 · 2026-03-13

**Soundness:** 3
**Presentation:** 3
**Significance:** 3
**Originality:** 3
**Overall Recommendation:** 5
**Confidence:** 1

**Summary:**

The paper proposes a method for estimating continuous treatment effects with a two-stage kernel ridge regression (KRR) process that learns the response function and then constructs pseudo-outcomes to train a second KRR model in a simpler space. The experiments show superior performance on a synthetic and a semi-synthetic dataset.

**Compliance With Llm Reviewing Policy:**

Affirmed.

**Final Justification:**

The authors addressed my questions, and the score remains positive.

**Key Questions For Authors:**

-

**Limitations:**

yes

**Strengths And Weaknesses:**

Strengths
- Extensive theoretical analysis of the proposed method.
- Superior mean integrated squared error values for experiments on synthetic and semi-synthetic datasets including the Job Corps dataset which is a popular benchmark for the continuous treatment effect estimation task.
- Clearly written text.

Weaknesses
- Most of the text lies in the appendices, which may indicate that a conference paper format is not the best fit for this method.
- Reference "Singh et al. (2024)" has only one author.

---

> ### Author Rebuttal · Authors · 2026-03-29
>
> We sincerely thank you for recognizing the value of our work and for the positive evaluation. Compared with prior work, our method has substantial advantages when the covariate distribution is poorly behaved, when conditional-density estimation is difficult, and when the problem lies in a weak-overlap regime. We hope the clarifications below make the contribution and impact of our paper even clearer. If any further questions about the paper arise, we would be more than happy to clarify them.
>
> **W1. Appendix-heavy presentation.**
>
> The main methodology, the core theoretical statements, and the numerical results are all in the main paper; the appendix is primarily devoted to proofs. In the revision, we would add a short proof sketch and more proof intuition to the main text so that the presentation is more self-contained and easier to follow in conference format.
>
> **W2. Reference formatting.**
>
> In the current submission, Singh (2022) refers to a single-author paper, while Singh et al. (2024) refers to the three-author Biometrika paper by Singh, Xu, and Gretton. We checked the manuscript and reference list, and the 2024 citation is correctly associated with the multi-author paper.

---

> > ### Author Rebuttal · Reviewer_SNs9 · 2026-04-04
> >
> > Thank you for your clarifications.

---

### Decision · Program_Chairs · 2026-04-30

**Decision:**

Accept (regular)

**Comment:**

The reviewers engaged in a constructive and detailed discussion about the work. There were two important/recurring comments that several of the reviewers raised:

1. the novelty of the work in light of other closely related papers (Singh and Wang). The authors responded that unlike their approach, Singh gives a plug-in estimator which leads to less favorable theoretical properties (estimation bottleneck is the complexity of $\mathcal{F}$) while Wang studies model selection for single-stage kernel ridge regression (KRR) under covariate shift.
2. the requirement of a well-specified $f^*$: The discussion between the reviewers and the authors implies that this assumption is not trivial to relax -- which I don't see as a particularly negative thing given that the authors are up front about the necessary assumptions. Future work might enable partial relaxations of this assumption. The authors clearly explained in the rebuttal what that assumption "buys" us: valid estimation under weak overlap, poorly behaved covariate distributions, or unstable conditional-density estimation.


Note: I've downweighted the review by SNs9 due to brevity and a lack of engagement in the discussion. The other 3 reviewers provided ample feedback ensuring fairness in the review process